# BAGEL-WORLD 🌍: TOWARDS HIGH-QUALITY VISUAL QUESTION-VISUAL ANSWERING

## ABSTRACT

This paper studies Visual Question–Visual Answering (VQ-VA): generating an image, rather than text, in response to a user's visual question—an ability that has recently emerged in proprietary systems such as NanoBanana and GPT-Image. To also bring this capability to open-source models, we introduce BAGEL-World, a data-centric framework built around an agentic pipeline for large-scale, targeted data construction. Leveraging web-scale deployment, this pipeline crawls a massive amount of ∼1.8M high-quality, interleaved image–text samples for model training. For evaluation, we further release IntelligentBench, a human-curated benchmark that systematically assesses VQ-VA along the aspects of world knowledge, design knowledge and reasoning. Training with BAGEL-World yields strong empirical gains: it helps LightBAGEL attain 45.0 on IntelligentBench, substantially surpassing the best prior open-source baselines (*i.e.*, 6.81@Light-BAGEL, 1.94@UniWorld-V1), and significantly narrowing the gap toward leading proprietary systems (*e.g.*, 81.67@NanoBanana, 82.64@GPT- Image). By releasing the full suite of model weights, datasets, and pipelines, we hope it will facilitate future research on VQVA.

## 1 INTRODUCTION

Driven by rapid advances in large multimodal generative models, frontier systems such as GPT-Image-1 (OpenAI, 2025) and NanoBanana (Nano Banana AI, 2025) now demonstrate exceptionally strong image generation and editing capabilities, showing reliable instruction following, high-fidelity synthesis, and improved consistency. Beyond these strengths, they also begin to exhibit an emergent ability we term Visual Question-Visual Answering (VQ-VA), *i.e.*, responding to a visual question with an image. As illustrated in Figure 1, when given a photo of a broken window and asked to speculate about what might be on the ground, NanoBanana generates an image depicting shards of glass; when shown an illustration of the stock market with a bull and asked "What is the contrasting trend?", NanoBanana creates an image of a bear to represent a bearish market. Producing such visual answers requires conditioning on the input image and instruction and, more critically, leveraging internalized world knowledge and multi-step reasoning to yield contextually coherent outputs.

Despite this progress, VQ-VA remains largely restricted to proprietary systems such as GPT-Image-1 and NanoBanana. As evident in Figure 1, current open-source models consistently underperform on these tasks: they often misinterpret the question or lack the world knowledge needed to synthesize an appropriate visual answer. We hypothesize that the primary bottleneck is data scarcity—open-source solutions are predominantly trained on standard image-editing datasets that emphasize predefined operations (*e.g.*, object addition, removal, replacement, style transfer), while underrepresenting free-form visual generation that demands knowledge and multi-step reasoning.

In this paper, we present BAGEL-World, a data-driven framework to bridge this gap. At its core is an agentic data-construction pipeline with five modules: (1) Retriever—identifies semantically and knowledge-driven image pairs from web-interleaved documents; (2) Instruction Generator—produces free-form questions that require knowledge and reasoning, conditioned on the first image and using the second image as the answer; (3) Filter—automatically removes low-quality questions

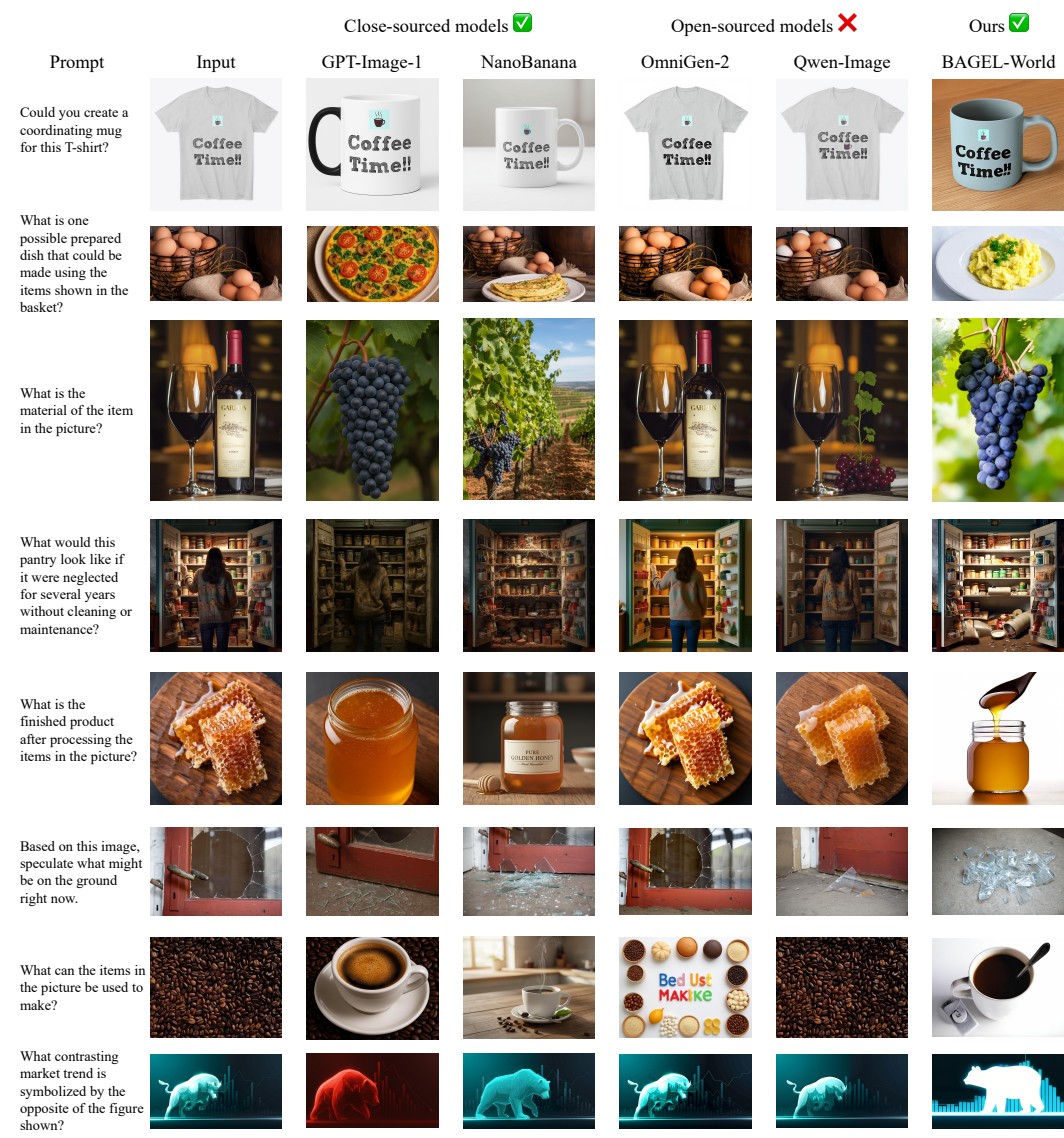

Figure 1: Examples of Visual Question–Visual Answering (VQ-VA), highlighting the substantial gap between existing closed-source models and open-weight models. The rightmost column further shows that a model trained with BAGEL-World significantly improves VQ-VA performance.

or pairs; (4) Rewriter—rephrases questions to enhance linguistic diversity; and (5) Reasoner—generates a natural-language reasoning trace that explains how to approach the question, what knowledge is required, and the detailed transformation from the source image to the target image.

Deployed at web scale, this pipeline successfully curates 1.8M high-quality, interleaved image–text training samples across three subdomains: world knowledge (covering scientific, spatial, temporal, and other real-world domains), design knowledge, and reasoning. Moreover, to systematically assess models' VQ-VA capability, we introduce IntelligentBench, a human-curated benchmark sourced from real-world, web-interleaved documents. Each item is designed to probe specific knowledge and reasoning demands in VQ-VA. Additionally, we leverage VLMs (*e.g.*, GPT-4o (OpenAI, 2025) and Gemini-2.5-Flash (Comanici et al., 2025) as automatic judges to facilitate large-scale evaluation and also compare their evaluation against human judgment.

To evaluate the effectiveness of BAGEL-World, we fine-tune LightBAGEL (Anonymous, 2025) (a fully open-source model, details in the supplementary files) on the 1.8M curated training samples and evaluate on the IntelligentBench. The results are exciting: while the prior open-source models only attain trivial performance (*e.g.*, 6.81@LightBagel, 1.94@UniWorld-V1), our BAGEL-World

Table 1: Comparison of major image-to-image datasets. QA indicates whether the dataset instructions are in question format rather than direct prompts. Knowledge-centric denotes whether the instructions require world knowledge. Real image is true only when both the input and output images are real. Concepts refers to the number of distinct words in the instructions. Note: For SEED-Data-Edit, only a subset (0.073M out of 3.7M) consists of real images.

| Dataset (image-to-image) | #Size | Freeform | QA | Knowledge Centric | Real Image | Concepts |
|---|---|---|---|---|---|---|
| MagicBrush (Zhang et al., 2023) | 10K | ✗ | ✗ | ✗ | ✓ | 2K |
| InstructPix2Pix (Brooks et al., 2023) | 313K | ✗ | ✗ | ✗ | ✗ | 11.6K |
| HQ-Edit (Hui et al., 2024) | 197K | ✗ | ✗ | ✗ | ✗ | 3.7K |
| SEED-Data-Edit (Ge et al., 2024) | 3.7M | ✗ | ✗ | ✗ | ✗ | 29.2K |
| UltraEdit (Zhao et al., 2024) | 4M | ✗ | ✗ | ✗ | ✗ | 3.7K |
| AnyEdit (Yu et al., 2025) | 2.5M | ✗ | ✗ | ✗ | ✗ | 6.4K |
| ImgEdit (Ye et al., 2025) | 1.2M | ✗ | ✗ | ✗ | ✗ | - |
| MetaQuery (Pan et al., 2025) | 2.4M | ✓ | ✗ | ✗ | ✓ | - |
| **Ours** | 1.8M | ✓ | ✓ | ✓ | ✓ | 87.9K |

substantially lifts the performance to 45.0, as shown in Table 2. Similar improvements can also be observed when evaluating on other VQ-VA-related benchmarks like RISEBench (Zhao et al., 2025) and KRIS-Bench (Wu et al., 2025c), where the full results are illustrated in Table 3 and Table 4. Moreover, our results demonstrate a substantial narrowing of the gap with leading proprietary systems such as Gemini (Google, 2024; Comanici et al., 2025) and GPT-4o (OpenAI, 2025).

To summarize, our contributions are as follows: (1) BAGEL-World, an agentic framework for curating free-form image manipulation data; (2) BAGEL-World 1.8M, a large-scale open-source dataset targeting knowledge- and reasoning-centric manipulations; (3) IntelligentBench, a human-curated benchmark for evaluating such abilities; and (4) a new model trained on BAGEL-World that surpasses all fully open-source models on multiple benchmarks. We will release the weights, datasets, pipelines, and benchmark to facilitate future research in Visual Question–Visual Answering.

## 2 RELATED WORK

**I2I Models.** Existing I2I models can be broadly categorized into three types: (1) single I2I models, (2) unified understanding-and-generation (U&G) models, and (3) leading proprietary models. For single I2I models, InstructPix2Pix (Brooks et al., 2023) leverages synthetic data generated by GPT-3 (Brown et al., 2020) and Stable Diffusion (Rombach et al., 2022) to train a conditional diffusion model capable of following human-written editing instructions. Emu Edit (Sheynin et al., 2024) is also diffusion-based, but it is trained on a diverse spectrum of editing tasks, including region-based I2I, free-form editing, and traditional computer vision tasks. Modern single I2I models such as Step1X-Edit (Liu et al., 2025), FLUX.1-Kontext (Labs et al., 2025), and Qwen-Image (Wu et al., 2025a) have substantially improved editing performance through both data scaling and model scaling. In parallel, U&G Chameleon-Team (2024); Zhou et al. (2024); Pan et al. (2025); Deng et al. (2025); Lin et al. (2025); Chen et al. (2025) models have gained popularity, benefiting from strong performance and cross-task learning advantages by combining understanding and generation. As for proprietary models, NanoBanana (Nano Banana AI, 2025) and GPT-Image-1 (OpenAI, 2025) still exhibit a noticeable advantage over all other models, particularly showing emerging abilities on I2I tasks that require world knowledge and reasoning. The main motivation of our work is to narrow this gap in this specific domain for the open-source community.

**Public I2I datasets.** MagicBrush (Zhang et al., 2023) introduces a manually annotated dataset containing 10k triplets, covering four types: single-turn, multi-turn, mask-provided, and mask-free editing. HQ-Edit (Hui et al., 2024) builds a scalable data collection pipeline leveraging GPT-4V (Achiam et al., 2023) and DALL·E 3 (Betker et al., 2023), resulting in around 200k editing samples. UltraEdit (Zhao et al., 2024) employs an automatic pipeline that integrates an LLM and SDXL (Podell et al., 2023), presenting a 4M-scale dataset consisting of real input images and synthetic edited images. SEED-Data-Edit (Ge et al., 2024) proposes a hybrid dataset constructed from both human annotation and automatic pipelines, and further introduces specifically designed high-quality multi-turn image-editing data. OmniEdit-1.2M (Wei et al., 2024) is built using seven different spe-

cialist models and employs an importance sampling strategy to improve data quality. ImgEdit (Ye et al., 2025) and AnyEdit2.5 (Yu et al., 2025) expand the coverage of editing types to 13 and 25, respectively, thereby enhancing the instruction diversity of image-editing datasets. Motivated by the strong performance of GPT-Image-1 (OpenAI, 2025) in generation tasks, GPT-IMAGE-EDIT-1.5M (Wang et al., 2025c) relabels previous OmniEdit, HQ-Edit, and UltraEdit datasets using GPT-Image-1, further improving the quality of open-source image-editing resources. Despite these advances, existing I2I datasets are primarily designed for standard image editing tasks. In contrast, ours targets the Visual Question–Visual Answering task, with a stronger emphasis on knowledge and reasoning.

**I2I benchmarks.** EmuEdit Benchmark (Sheynin et al., 2024) covers 7 fixed editing types and adopts L1, CLIP-I, and DINO as scoring metrics to evaluate editing ability. MagicBrushEdit Benchmark (Zhang et al., 2023) extends this to 9 predefined tasks and provides two modes: mask-free and mask-provided. ImageEdit (Ye et al., 2025) further expands to 14 tasks, introduces VLM-based scoring, and supports multi-turn editing with varying difficulty levels. OMNI-EDIT-Bench (Wei et al., 2024) is a high-resolution, multi-aspect-ratio, multi-task benchmark comprising 434 edits derived from 62 images, evaluated with both VLM scorers and human judgments. GEdit-Bench (Liu et al., 2025) contains 606 real-world user editing cases, filtered by humans and scored with VLMs. These benchmarks are designed for standard image editing evaluation, whereas ours targets the VQVA setting. We also highlight two concurrent reasoning- and knowledge-based image editing benchmarks: RISEBench (Zhao et al., 2025) and KRIS-Bench (Wu et al., 2025c). To our knowledge, some cases in RISEBench can be regarded as VQVA instances, and certain domains in KRIS-Bench also overlap with this setting. Our benchmark, IntelligentBench, differs in two key aspects: (1) RISEBench and KRIS-Bench are primarily designed for pixel-level alignment editing, whereas IntelligentBench includes many cases that require semantic-level reasoning beyond pixel alignment, as illustrated in Fig. 1; and (2) rather than relying on synthetic images, IntelligentBench is curated from real-world web content, with each case manually verified and paired with a real reference answer image.

## 3 METHODS

In this section, we elaborate on the details of the BAGEL-World data framework and Intelligent-Bench.

### 3.1 BAGEL-WORLD DATA FRAMEWORK

**Motivation.** The BAGEL-World framework tackles two key challenges: identifying suitable data for VQVA and designing a scalable pipeline for its construction. We target image pairs whose transformations (Image1 $\Leftrightarrow$ Image2) inherently require knowledge or reasoning—for example, (car wheel $\Leftrightarrow$ car), (mathematical equation $\Leftrightarrow$ its graph), or (window of a house $\Leftrightarrow$ broken glass on the ground). Such transformations capture semantic-level connections rather than superficial pixel-level alterations. By providing an image and formulating transformation-related questions whose answers require generating their corresponding counterparts, models can be trained to acquire knowledge-related VQVA ability. The subsequent step is to identify data sources rich in such pairs and to develop automated pipelines for large-scale collection and refinement. Inspired by the data used in large language model pretraining, we regard web-interleaved documents as a particularly promising candidate, since they naturally contain extensive world knowledge alongside closely associated images and text. And develop a pipeline to generate VQVA data from web interleave documents.

The BAGEL-World framework, as shown in Fig. 2, consists of two stages: preprocessing and an agentic pipeline for VQVA data construction. In the preprocessing stage, noisy web-interleaved documents are processed and assigned semantic labels, with only those belonging to the knowledge and design categories retained. The agentic pipeline then transforms the filtered documents into high-quality visual question–visual answering samples. Using this framework, we construct the 1.8M BAGEL-World dataset, comprising 24.35% reasoning, 30.37% design knowledge, and 43.69% world knowledge. The details are as follows:

**Preprocessing.** To handle web-scale data, our first goal is to efficiently filter candidate interleaved documents. The core principle of filtering is to identify documents that contain many images with strong knowledge connections. We observe that correlations among images within a webpage are often aligned with the document's topic. Therefore, we employ the document topic as an initial criterion for filtering. Since the topic is usually not provided in web data, we design a loop to label

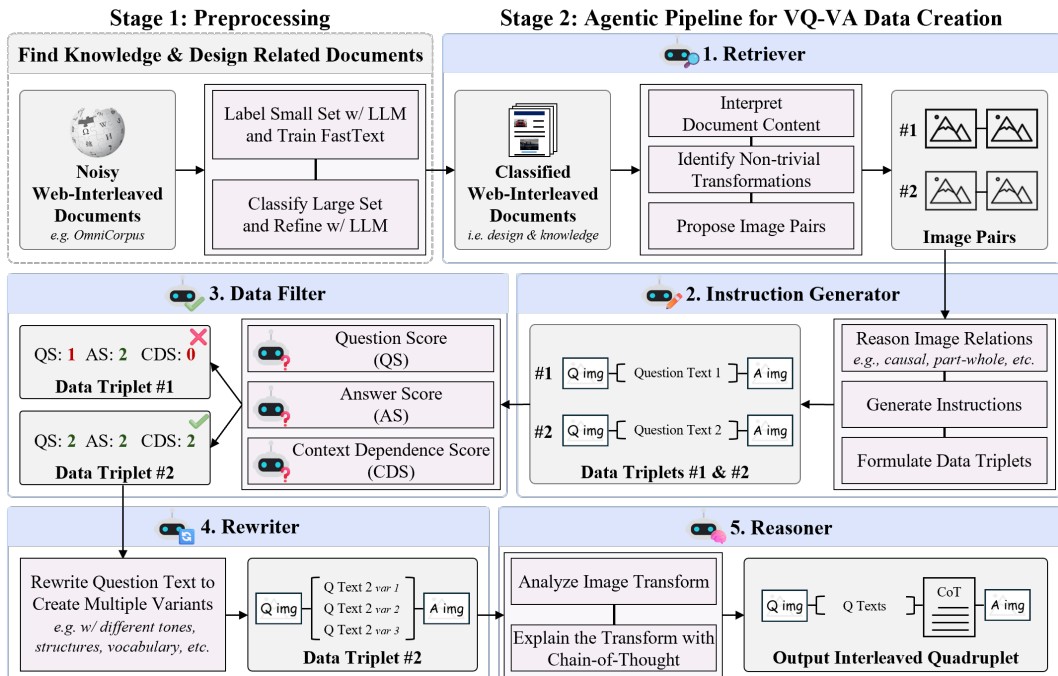

Figure 2: Illustration of the BAGEL-World framework for creating VQVA data. The framework consists of two stages: (1) preprocessing, which classifies and filters web-interleaved documents, and (2) an agentic pipeline that generates VQ-VA samples from the filtered documents. The agentic pipeline contains five sub-modules: retriever, filter, instruction generator, rewriter, and reasoner.

documents efficiently, inspired by the data pipeline proposed in DeepSeek-Math (Shao et al., 2024). Specifically, we first use an LLM (Qwen2.5-14B (Yang et al., 2025)) to label a subset of the data and identify samples of the required types. The labeled data are then used to train a lightweight FastText (Joulin et al., 2016) classifier, which enables large-scale labeling with high efficiency. Finally, we apply an LLM again to refine the coarse labels produced by FastText. The final outputs of preprocessing are web-interleaved documents containing knowledge- and design-related content.

**Agent Pipeline for VQ-VA data Creation.** We aim to build an automatic data engine that ingests web-interleaved documents and generates high-quality visual question–visual answering data requiring world knowledge, design knowledge, and reasoning. To achieve this, we design an agentic pipeline that decomposes the process into subtasks, with each agent worker handling a specific component. Each worker is powered by state-of-the-art VLMs (GPT-4o (OpenAI, 2025) and Seed1.5VL-Thinking (Seed, 2025)), and is guided by carefully designed system prompts and chain-of-thought reasoning, without memory sharing across workers. We define the agent workers below:

(1) *Retriever*: selects image pairs from interleaved documents that can serve as the basis for free-form questions. It focuses on pairs with meaningful transformations, especially those involving non-trivial relations grounded in knowledge and reasoning. We also find it beneficial for the retriever to capture the document's topic; hence, its input is the full document rather than merely the image list. The detail prompt is provided in Appendix Table 7.

(2) *Instruction Generator*: produces a natural language question about one image in the pair, with the other image serving as the answer. For example, given a pair consisting of a car wheel and a racing car equipped with that wheel, if the question image is the car wheel, the generated question might be: "What is it used for?" The instructions are deliberately designed to probe diverse forms of knowledge and reasoning, including but not limited to: temporal or causal relations (e.g., the same subject over time or ordered steps with clear causal dependencies); compositional or spatial structures (e.g., part–whole relations, inside–outside contrasts, exploded or sectional views); and scientific or analytical phenomena (e.g., visual explanations of scientific or mathematical concepts). The detail prompt is provided in Appendix Table 8.

(3) *Filter*: removes low-quality triplets ⟨Question Image, Question Text, Answer Image⟩. Through careful multi-round human-in-the-loop analysis, we identify several common issues leading to low-quality data, such as poorly formulated questions, ambiguous or irrelevant answer images, and context shortcuts (i.e., cases where the answer image can be inferred without considering the question image, relying only on textual cues). To effectively address these issues, we design a multi-score VLM-based filtering strategy with three sub-scorers: Question Score (QS), Answer Score (AS), and Context Dependence Score (CDS). The detailed prompts are provided in Appendix Table 9, 10 and 11, respectively. Each score is assigned on a three-level scale 0, 1, 2, and only cases with a summed score of 6 are retained. In addition, we manually design and iteratively refine the scoring template, and adopt a chain-of-thought approach during scoring, where the model generates an analysis before assigning scores, thereby further improving filtering effectiveness.

*(4) Rewriter*: increases instruction diversity by producing multiple variants of the original questions. The variants differ in tone, sentence structure, vocabulary, expression, and overall linguistic naturalness. This rewriting process is essential for improving instruction-following ability. The detail prompt is provided in Appendix Table 12.

*(5) Reasoner*: generates a language-based chain-of-thought reasoning that explains how to transform the input image into the output image. The process involves analyzing the question, observing the question image, identifying necessary changes, determining which elements remain consistent, and highlighting key modifications. This reasoning trace is then incorporated with the triplet to construct a new data format quadruplet ⟨Question Image, Question Text, Editing reasoning trace, Answer Image⟩. This interleaved quadruplet is later used to fine-tune a unified model, i.e., LightBAGEL, to improve both reasoning-trace generation and instruction-following ability. The detail prompt is provided in Appendix Table 13.

**High-quality subset curation.** Following prior works such as (Deng et al., 2025; Wu et al., 2025a), which typically adopt multi-stage training, we employ a two-stage strategy: continued pretraining and supervised fine-tuning (SFT). In the first stage, we train on the full large-scale dataset for additional steps to further strengthen knowledge and instruction-following ability. In the second stage, we focus on a smaller high-quality subset for fewer steps to improve overall quality. Specifically: (1) we apply stricter filtering, retaining the best one-third of the data, which yields about 500M high-quality samples; and (2) leveraging the fact that video models naturally encode temporal knowledge, we use the Seedance video model (Gao et al., 2025) to construct a smaller set of about 50k temporally related VQ-VA samples. This second stage is conducted solely on high-quality data, mixed with LightBAGEL data.

Figure 3: Examples of IntelligentBench.

### 3.2 INTELLIGENTBENCH

**Benchmark data.** The purpose of IntelligentBench is to evaluate the visual question–visual answering abilities of existing I2I models, where the questions require knowledge and reasoning to answer. The benchmark is inspired by open-ended visual question answering in the VLM domain, where a model is asked a question about an image and provides a textual response. Following this concept, we instead pose free-form questions to I2I models, but require the answers in image format. We further adopt the domain split introduced in our dataset. In total, 360 cases were manually constructed by human experts, consisting of 171 world knowledge, 88 design knowledge, and 101 reasoning cases. The construction of IntelligentBench involves three main steps: (1) Document Review. Human experts examined about 3k classified interleaved web documents and, from each, selected the image pair that best represented the document's content and exhibited strong semantic connections. (2) Question Design. For each selected image pair, experts designed free-form questions targeting world knowledge, design knowledge, or reasoning. (3) Expert Cross-Review. All candidate cases

Table 2: Results on IntelligentBench, a benchmark designed for Visual Question–Visual Answering. Fully open-source models (both training data and model weights) are shown without shading, open-weight models are shaded in light blue, and closed-source models are shaded in light gray for clarity.

| Model | World Knowledge | Design Knowledge | Reasoning | Overall |
|---|---|---|---|---|
| GPT-Image-1 (OpenAI, 2025) | 84.5 | 80.68 | 81.19 | 82.64 |
| Nano Banana (Nano Banana AI, 2025) | 81.6 | 82.95 | 80.69 | 81.67 |
| BAGELThink (Deng et al., 2025) | 61.99 | 55.11 | 62.38 | 60.42 |
| Qwen-Image (Wu et al., 2025a) | 38.07 | 33.66 | 32.75 | 34.31 |
| FLUX.1-Kontext-Dev (Labs et al., 2025) | 20.18 | 24.43 | 19.80 | 21.11 |
| OmniGen2 (Wu et al., 2025b) | 11.11 | 13.07 | 7.92 | 10.69 |
| Step1X-Edit (Liu et al., 2025) | 11.7 | 10.23 | 15.35 | 12.36 |
| UniWorld-V1 (Lin et al., 2025) | 2.92 | 0.57 | 1.49 | 1.94 |
| LightBAGEL | 6.14 | 7.39 | 7.43 | 6.81 |
| **Ours** | **43.57** | **46.02** | **46.53** | **45.00** |

were cross-checked by multiple experts, with each independently verifying the cases proposed by others. Only unanimously agreed-upon cases were retained, resulting in 360 high-quality instances (171 world knowledge, 88 design knowledge, and 101 reasoning).

**Evaluation Metric.** We use a VLM as the evaluator, following rules: (1) the VLM is provided with the question image, question text, reference answer image, the generated image, and a carefully designed system prompt; (2) the VLM is required to output a score as an integer in 0, 1, 2. A detailed explanation of how the VLM is guided to assign each level (0, 1, 2) is provided in the appendix, and the full prompt is similar to that used in BAGEL (Deng et al., 2025).

**Reverification of Evaluation Accuracy.** In this section, we examine the reliability of the metric. We employ four human experts and two state-of-the-art VLMs to assign scores to the outputs produced by four different models. We report the agreement among human experts, as well as the agreement between VLM scorers and human experts, as shown in Acc of Figure 4. The average agreement among humans is 82.5%, while the agreement between humans and GPT-4o (OpenAI, 2025) is 80.6%, and between humans and Gemini-2.5-Flash (Comanici

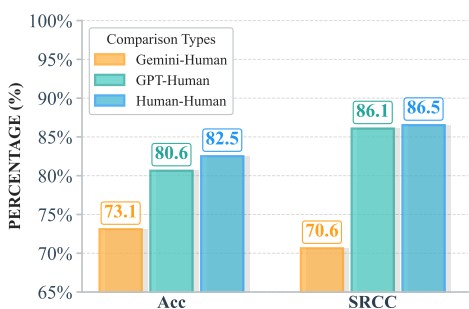

Figure 4: Alignment between VLM and human scores. We compare Gemini-2.5-Flash vs. human experts, GPT-4o vs. human experts, and agreement among human experts. We report the Accuracy and Spearman Rank Correlation Coefficient (SRCC) for comprehensive comparison.

et al., 2025) is 73.1%. Additionally, we report the Spearman Rank Correlation Coefficient (SRCC) to assess ranking alignment, and the results further show that GPT-4o is more consistent with human preferences. Therefore, we adopt GPT-4o as our automatic scorer.

## 4 Experiments

**Implementation details.** We adopt LightBAGEL Anonymous (2025) as our baseline model, since its architecture, training pipeline, and dataset are fully open source. Details of LightBAGEL are provided in the supplementary materials. Moreover, its training requires significantly less compute compared with other models. LightBAGEL is a unified model that combines Qwen2.5-VL-7B (Yang et al., 2025) as the understanding branch and Wan2.2-TI2V-5B (Wan et al., 2025) as the generation branch. We incorporate BAGELWorld Editing into the overall training data of LightBAGEL with a sampling ratio of 25%, and fine-tune the model for a total of 30k steps ($\approx$ 3 days on 32 H200 GPUs). Both branches are trained following the original LightBAGEL settings with the timestep shift set to 4. We adopt a two-stage training scheme: (1) continued train LightBAGEL on a mixed dataset for 25k steps with AdamW and a cosine learning rate schedule (peak $1 \times 10^{-5}$). The mixed BAGEL-World dataset includes the original 40M LightBAGEL, 1.8M BAGELworld-edit. (2) supervised fine-tuning on a filtered high-quality subset ($\approx$ 13 of the data) for 5k steps with a constant learning rate of $1 \times 10^{-5}$.

Table 3: Results on RISEBench. Fully open-source models are shown without shading, open-weight models are shaded in light blue, and closed-source models are shaded in light gray for clarity.

| Model | Temporal | Causal | Spatial | Logical | Overall |
|---|---|---|---|---|---|
| Nano Banana (Nano Banana AI, 2025) | 25.9 | 47.8 | 37.0 | 18.8 | 32.8 |
| GPT-4o-Image (OpenAI, 2025) | 34.1 | 32.2 | 37.0 | 10.6 | 28.9 |
| Gemini-2.0-Flash-exp (Google, 2024) | 8.2 | 15.5 | 23.0 | 4.7 | 13.3 |
| Seedream-4.0 (Bytedance Seed, 2025) | 12.9 | 12.2 | 11.0 | 7.1 | 10.8 |
| BAGELThink (Deng et al., 2025) | 5.9 | 17.7 | 21.0 | 1.1 | 11.9 |
| Qwen-Image-Edit (Wu et al., 2025a) | 4.7 | 10.0 | 17.0 | 2.4 | 8.9 |
| FLUX.1-Kontext-Dev (Labs et al., 2025) | 2.3 | 5.5 | 13.0 | 1.2 | 5.8 |
| Step1X-Edit (Liu et al., 2025) | 0.0 | 2.2 | 2.0 | 3.5 | 1.9 |
| OmniGen (Xiao et al., 2025) | 1.2 | 1.0 | 0.0 | 1.2 | 0.8 |
| EMU2 (Sun et al., 2024) | 1.2 | 1.1 | 0.0 | 0.0 | 0.5 |
| HiDream-Edit (Cai et al., 2025) | 0.0 | 0.0 | 0.0 | 0.0 | 0.0 |
| FLUX.1-Canny (Labs et al., 2025) | 0.0 | 0.0 | 0.0 | 0.0 | 0.0 |
| LightBAGEL | 1.1 | 1.1 | 3.0 | 1.1 | 1.6 |
| **Ours** | **14.1** | **21.1** | **14.0** | **1.1** | **12.7** |

Table 4: Results on KRIS-Bench. Fully open-source models are shown without shading, open-weight models are shaded in light blue, and closed-source models are shaded in light gray for clarity.

| Model | Factual | Conceptual | Procedural | Overall Average |
|---|---|---|---|---|
| GPT-4o (OpenAI, 2025) | 86.99 | 80.08 | 78.61 | 82.18 |
| Gemini-2.0 (Google, 2024) | 73.03 | 61.92 | 67.76 | 67.24 |
| Doubao (ByteDance, 2025) | 72.02 | 64.99 | 62.94 | 67.00 |
| OmniGen (Xiao et al., 2025) | 44.79 | 34.23 | 34.37 | 38.00 |
| Emu2 (Sun et al., 2024) | 57.81 | 43.75 | 43.57 | 48.69 |
| BAGEL-Think (Deng et al., 2025) | 62.75 | 62.49 | 42.76 | 57.91 |
| Step1X-Edit (Liu et al., 2025) | 53.32 | 52.51 | 37.21 | 49.17 |
| AnyEdit (Yu et al., 2025) | 52.06 | 50.96 | 37.68 | 48.21 |
| MagicBrush (Zhang et al., 2023) | 54.22 | 47.30 | 34.60 | 46.74 |
| InsPix2Pix (Brooks et al., 2023) | 33.38 | 32.47 | 25.84 | 31.22 |
| LightBAGEL | 57.62 | 50.24 | 41.06 | 50.33 |
| **Ours** | **62.10** | **60.11** | **45.02** | **57.16** |

**Evaluation setting.** For a comprehensive evaluation of BAGEL-WORLD, we consider three domains with five benchmarks: (1) Visual Question–Visual Answering, evaluated on *IntelligentBench*; (2) reasoning- and knowledge-informed image editing, evaluated on *RISEBench* (Zhao et al., 2025) and *KRIS-Bench* (Wu et al., 2025c), both of which require precise pixel alignment and strong reasoning ability; and (3) standard image editing, evaluated on *GEdit-Bench* (Liu et al., 2025), constructed from real-world user editing cases, and *ImgEdit-Bench* (Ye et al., 2025), designed to assess instruction adherence, editing quality, and detail preservation. Results on *IntelligentBench* are shown in Table 2; results on *RISEBench* and *KRIS-Bench* are shown in Tables 3 and 4; and summarized results on traditional image editing tasks (*GEdit-Bench* and *ImgEdit-Bench*) are presented in Table 5. Following (Deng et al., 2025), for all knowledge-intensive benchmarks, the model is configured to first output reasoning content before generating the image, whereas for traditional image editing benchmarks, we directly generate the image. For all benchmarks, we adopt a double-CFG strategy when evaluating both our model and the baseline LIGHTBAGEL, with the image CFG scale set to 2 and the text CFG scale set to 4. The time shift is fixed at 4 for both training and evaluation.

## 4.1 RESULTS ON VISUAL QUESTION–VISUAL ANSWERING

Based on IntelligentBench, we evaluate our BAGEL-World model along with other state-of-the-art closed-source and open-source models, as shown in Table 2. We report the normalized score (ranging from 0-100) for each subdomain as well as the average score. For each evaluated model, instances where no result image is returned are assigned 0 points. The results show that BAGEL-World achieves the best performance among fully open-source models, and the large gap between the baseline model LightBAGEL and BAGEL-World further supports the effectiveness of our dataset. (2) BAGEL-World even surpasses Qwen-Image, which was pretrained on large-scale proprietary data and adopted RL for further improvement. (3) Compared with leading proprietary models such

Table 5: Results on Standard Image Editing Benchmarks (GEdit-Bench-EN and ImgEdit-Bench). Higher scores are better. Fully open-source models are shown without shading, open-weight models are shaded in light blue, and closed-source models are shaded in light gray for clarity.

| Model | GEdit-Bench-EN | | | ImgEdit-Bench |
|---|---|---|---|---|
| | SC | PQ | Overall | Overall |
| GPT-4o (OpenAI, 2025) | 7.85 | 7.62 | 7.53 | 4.20 |
| Gemini-2.0-flash (Google, 2024) | 6.73 | 6.61 | 6.32 | - |
| Instruct-Pix2Pix (Brooks et al., 2023) | 3.58 | 5.49 | 3.68 | 1.88 |
| MagicBrush (Zhang et al., 2023) | 4.68 | 5.66 | 4.52 | 1.90 |
| AnyEdit (Yu et al., 2025) | 3.18 | 5.82 | 3.21 | 2.45 |
| ICEdit (Zhang et al., 2025) | 5.11 | 6.85 | 4.84 | 3.05 |
| Step1X-Edit (Liu et al., 2025) | 7.09 | 6.76 | 6.70 | 3.06 |
| OmniGen2 (Wu et al., 2025b) | 7.16 | 6.77 | 6.41 | 3.43 |
| BAGEL (Deng et al., 2025) | 7.36 | 6.83 | 6.52 | 3.20 |
| Ovis-U1 (Wang et al., 2025a) | – | – | 6.42 | 3.98 |
| UniPic (Wang et al., 2025b) | 6.72 | 6.18 | 5.83 | 3.49 |
| UniPic 2.0 (Wei et al., 2025) | – | – | 7.10 | 4.06 |
| UniWorld-V1 (Lin et al., 2025) | 4.93 | **7.43** | 4.85 | 3.26 |
| LightBagel | 6.56 | 7.06 | 6.06 | 3.65 |
| **Ours** | **6.58** | 7.00 | **6.13** | **3.76** |

as GPT-4o and Gemini, a performance gap remains, although it has been substantially reduced. Full qualitative results of all models are provided in Appendix Figure 5- 33.

## 4.2 RESULTS ON REASONING-IMAGE EDITING BENCHMARK

In this domain, we evaluate our model on RISEBench and KRIS-Bench, as shown in Table 3 and Table 4, respectively. On RISEBench, the results indicate that: (1) our model achieves performance comparable to BAGELThink while requiring far less training data; (2) compared with the baseline model, our data substantially improve its performance; and (3) some large data-privacy models such as Qwen-Image-Edit and FLUX.1-Kontext-Dev underperform our model, highlighting potential issues of unbalanced data distribution and the necessity of free-form data like BAGEL-World. Furthermore, our results in Table 4 show a similar trend: BAGEL-World consistently outperforms existing fully open-source models. These findings further support the effectiveness of BAGEL-World and the benefits brought by enhanced VQVA capability. Qualitative results on RISEBench are provided in Appendix 34.

## 4.3 RESULTS ON STANDARD IMAGE EDITING BENCHMARK

Here we report standard image editing performance on GEdit-Bench-EN and ImgEdit-Bench, as shown in Table 5. The complete ImgEdit-Bench results for each subdomain (e.g., add/remove) are provided in the Appendix Table 6. Our model slightly improves upon LightBAGEL on both benchmarks. It is worth noting that this minor improvement, compared with the large gains in VQVA and knowledge-based editing tasks, further highlights the significant domain difference.

## 5 CONCLUSION

In this work, we introduced Visual Question–Visual Answering, where the answer to a visual question is itself an image. To bridge the capability gap with proprietary models, we proposed BAGEL-World, a scalable data-centric framework driven by an agentic pipeline for constructing high-quality, diverse training data. Through our web-scale pipeline, we curated 1.8 million high-quality samples and introduced IntelligentBench to rigorously evaluate this new capability. Experiments show that training LightBAGEL with our data delivers large gains, which markedly surpasses prior baselines and closes much of the gap to proprietary systems. We hope this work sheds light on future research.

## 6 ETHICS STATEMENT

We have read and will adhere to the ICLR 2026 Code of Ethics, including its principles on responsible stewardship, fairness, avoidance of harm, transparency, and respect for others' work. We explicitly acknowledge this Code and reflect on the wider impacts of our work in line with ICLR 2026 guidance.

**Data sources and licensing**. This paper constructs a large-scale training dataset and benchmark, BAGEL-World 1.8M, from publicly available, web-interleaved documents; IntelligentBench items are drawn from real-world web content with real reference answer images. We will release models, data, and pipelines following the original licenses of the sources, and respect takedown/opt-out requests where applicable.

**Human subjects and privacy**. This work does not involve interaction with human participants, clinical/behavioral intervention, or the collection of non-public personal data. Human involvement was limited to curation/quality review of publicly available content. We did not annotate or process biometric identifiers for the purpose of identification, and we took care to avoid re-identification risks in line with the Code's privacy guidance.

**Safety and misuse mitigation**. To reduce risks of harmful or misleading outputs, our data engine emphasizes knowledge/design cases rather than sensitive personal content and applies multi-stage, human-in-the-loop, and VLM-assisted filtering to remove low-quality or sensitive items before training and evaluation. We will accompany any released artifacts with usage guidelines that discourage malicious use (e.g., harassment, impersonation, or deceptive media).

**Evaluation transparency and bias**. We evaluate the dataset with both human raters and VLM-based judges (e.g., Gemini/GPT-4o). While these scorers improve scalability, they may encode societal biases; accordingly, we report agreement with human experts and will release evaluation prompts/protocols to support reproducibility and scrutiny.

**Legal compliance and research integrity**. We respect confidentiality and intellectual property, and we commit to accurate reporting of methods and results. If concerns are raised, we will follow ICLR's remediation processes.

**Conflicts of interest**. The authors declare no conflicts of interest and no sponsorship that would unduly influence the research or its presentation.

## 7 REPRODUCIBILITY STATEMENT

We support full replication by (1) detailing the data construction pipeline (Section 3.1; Figure 2)—including preprocessing of web-interleaved documents, the multi-score VLM filtering with QS/AS/CDS, and instruction rewriting and reasoning traces—and reporting dataset composition for BAGEL-World (1.8M items; proportions across reasoning, design, and world knowledge). (2) specifying modeling and training in Section 4—baseline sampling ratios, steps, optimizers/schedules, and compute —with hyperparameters used in the main experiments.

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

## A  APPENDIX

### A.1  LLM USAGE

During the preparation of this manuscript, we used OpenAI's GPT-5 model for minor language refinement and smoothing of the writing. The AI tool was not used for generating original content, conducting data analysis, or formulating core scientific ideas. All conceptual development, experimentation, and interpretation were conducted independently without reliance on AI tools.

## A.2 COMPLETE RESULTS ON INTELLIGENTBENCH OF DIFFERENT MODELS.

## A.3 COMPLETE RESULTS ON IMGEDIT

Table 6: Evaluation of image editing ability on ImgEdit-Bench. Higher scores are better for all metrics.

| Model | Add | Adjust | Extract | Replace | Remove | Background | Style | Hybrid | Action | Overall |
|---|---|---|---|---|---|---|---|---|---|---|
| GPT-4o | 4.61 | 4.33 | 2.90 | 4.35 | 3.66 | 4.57 | 4.93 | 3.96 | 4.89 | 4.20 |
| MagicBrush (Zhang et al., 2023) | 2.84 | 1.58 | 1.51 | 1.97 | 1.58 | 1.75 | 2.38 | 1.62 | 1.22 | 1.90 |
| Instruct-Pix2Pix (Brooks et al., 2023) | 2.45 | 1.83 | 1.41 | 2.01 | 1.44 | 1.44 | 3.55 | 1.20 | 1.46 | 1.88 |
| AnyEdit (Yu et al., 2025) | 3.18 | 2.95 | 1.14 | 2.49 | 2.21 | 2.88 | 3.82 | 1.56 | 2.65 | 2.45 |
| UltraEdit (Zhao et al., 2024) | 3.44 | 2.81 | 2.00 | 2.96 | 2.45 | 2.83 | 3.76 | 1.91 | 2.98 | 2.70 |
| Step1X-Edit (Liu et al., 2025) | 3.88 | 3.41 | 1.76 | 3.40 | 2.83 | 3.16 | 6.63 | 2.52 | 2.52 | 3.06 |
| ICEdit (Zhang et al., 2025) | 3.58 | 3.39 | 1.73 | 3.15 | 2.93 | 3.08 | 3.84 | 2.04 | 3.68 | 3.05 |
| OmniGen2 (Wu et al., 2025b) | 3.74 | 3.54 | 1.77 | 3.21 | 2.77 | 3.57 | 4.81 | 2.30 | 4.14 | 3.43 |
| BAGEL (Deng et al., 2025) | 3.56 | 3.31 | 1.88 | 2.62 | 2.88 | 3.44 | 4.49 | 2.38 | 4.17 | 3.20 |
| Ovis-U1 (Wang et al., 2025a) | 4.12 | 3.92 | 2.36 | 4.09 | 3.57 | 4.22 | 4.69 | 3.23 | 3.61 | 3.98 |
| UniPic (Wang et al., 2025b) | 3.66 | 3.51 | 2.06 | 4.31 | 2.77 | 3.77 | 4.76 | 2.56 | 4.04 | 3.49 |
| UniPic 2.0 (Wei et al., 2025) | - | - | - | - | - | - | - | - | - | 4.06 |
| UniWorld-V1 (Lin et al., 2025) | 3.82 | 3.66 | 2.31 | 3.45 | 3.02 | 2.99 | 4.71 | 2.96 | 2.74 | 3.26 |
| LightBagel | 4.21 | 3.39 | 1.58 | 4.09 | 3.39 | 4.37 | 4.38 | 3.47 | 3.99 | 3.65 |
| Ours | 4.24 | 3.12 | 1.39 | 4.23 | 3.68 | 4.21 | 4.47 | 3.90 | 4.59 | 3.76 |

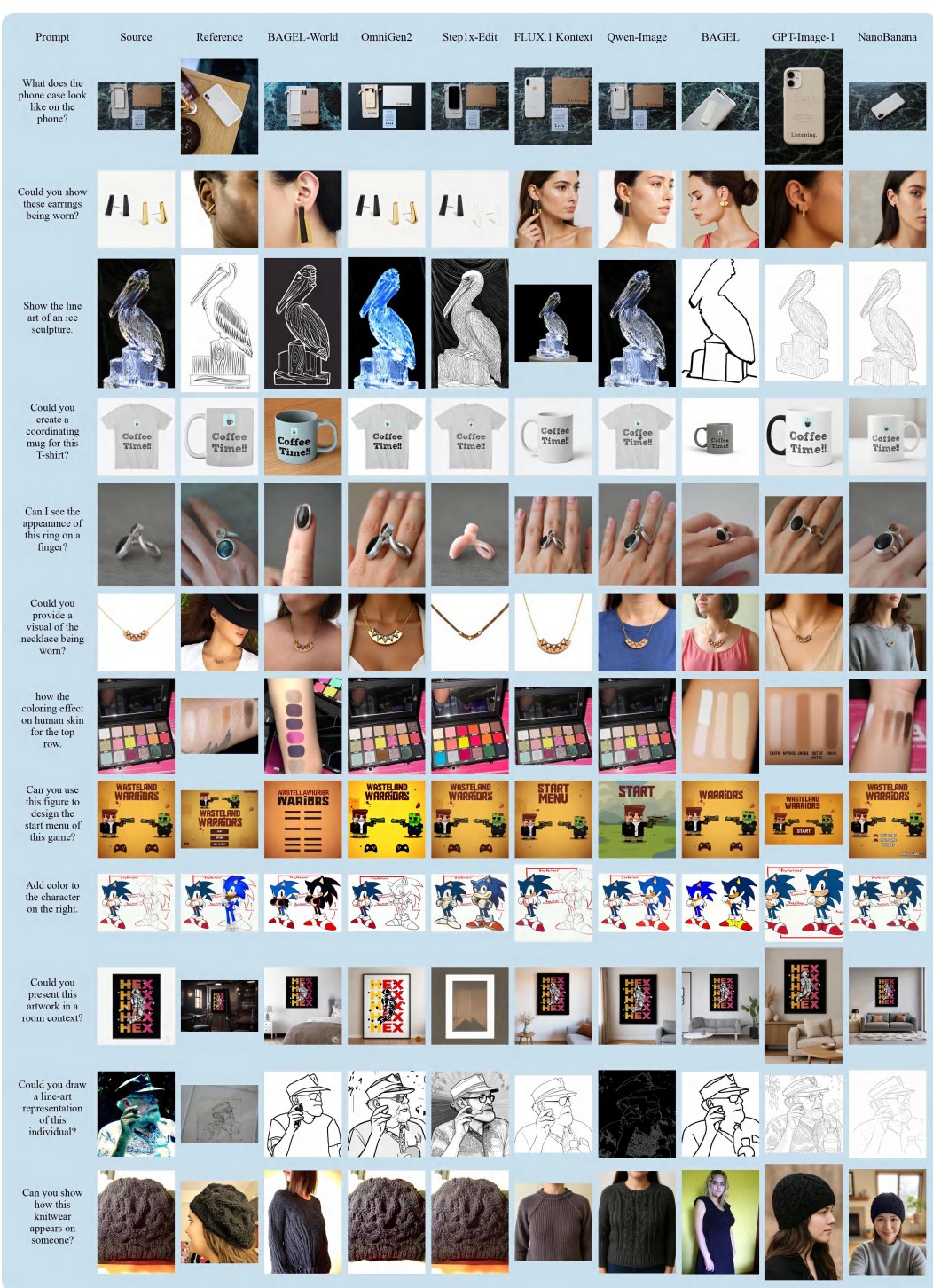

Figure 5: Comprehensive visualization of model performance on IntelligentBench (Subset Design, part 1/8).

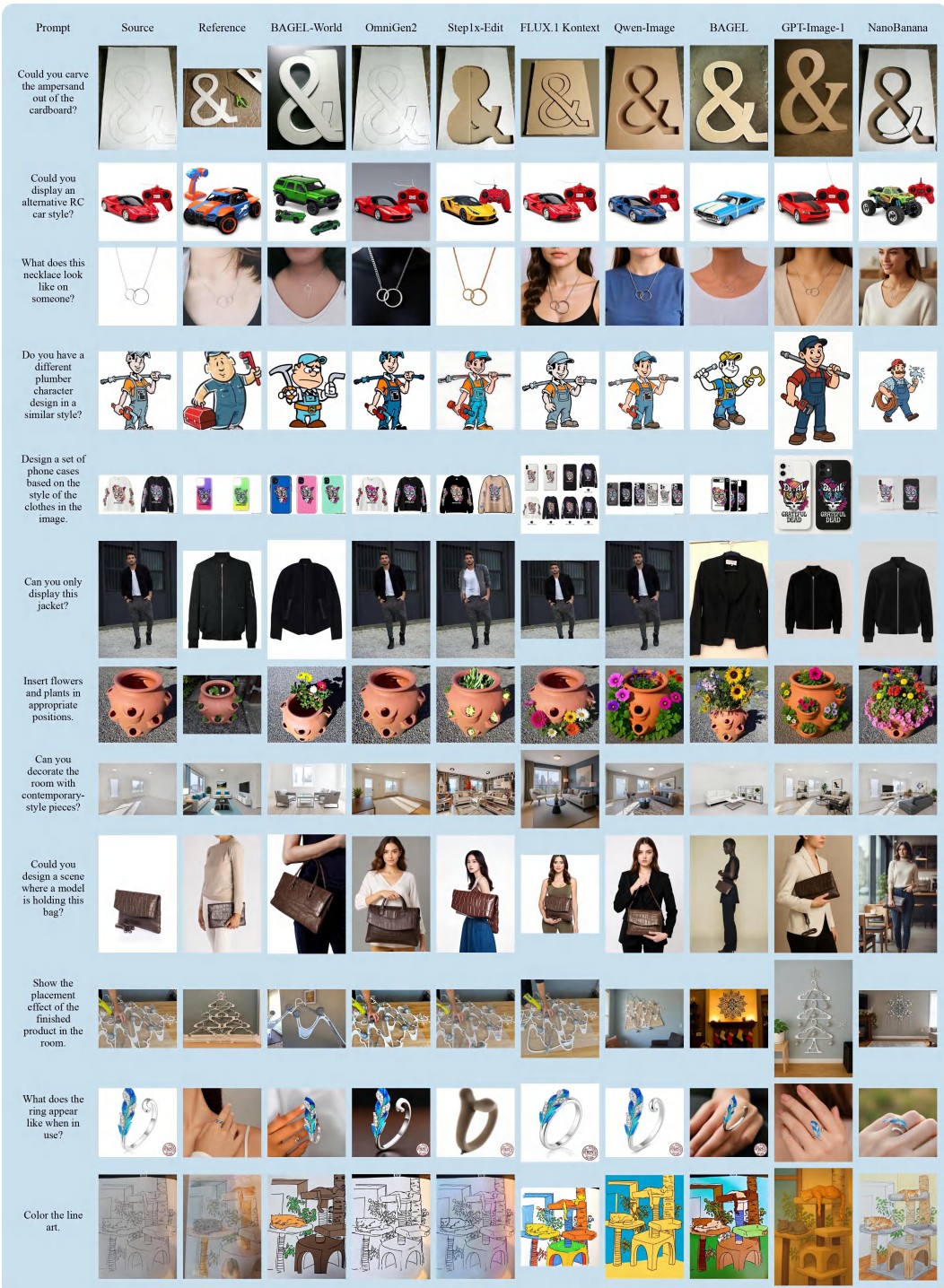

Figure 6: Comprehensive visualization of model performance on IntelligentBench (Subset Design, part 2/8).

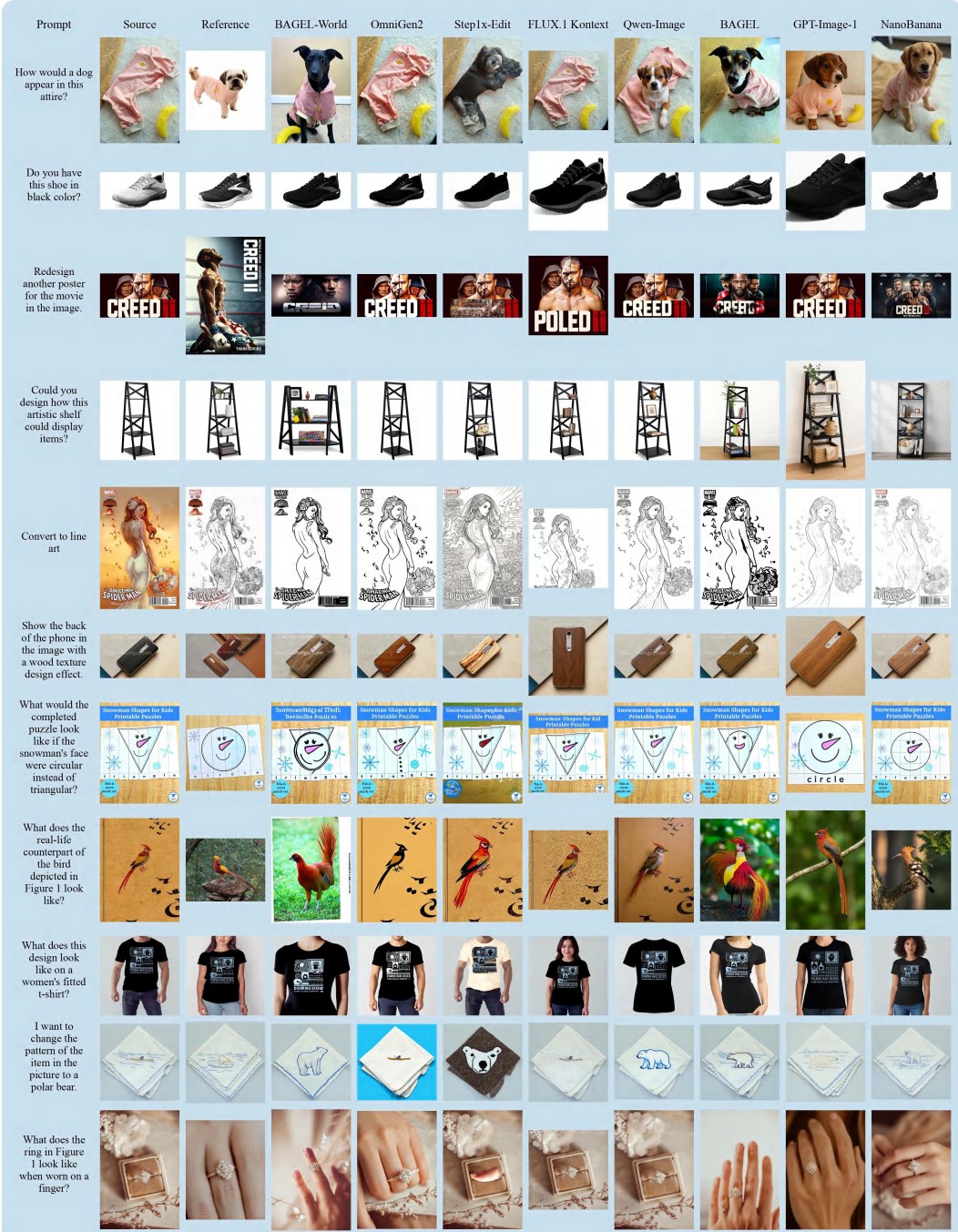

Figure 7: Comprehensive visualization of model performance on IntelligentBench (Subset Design, part 3/8).

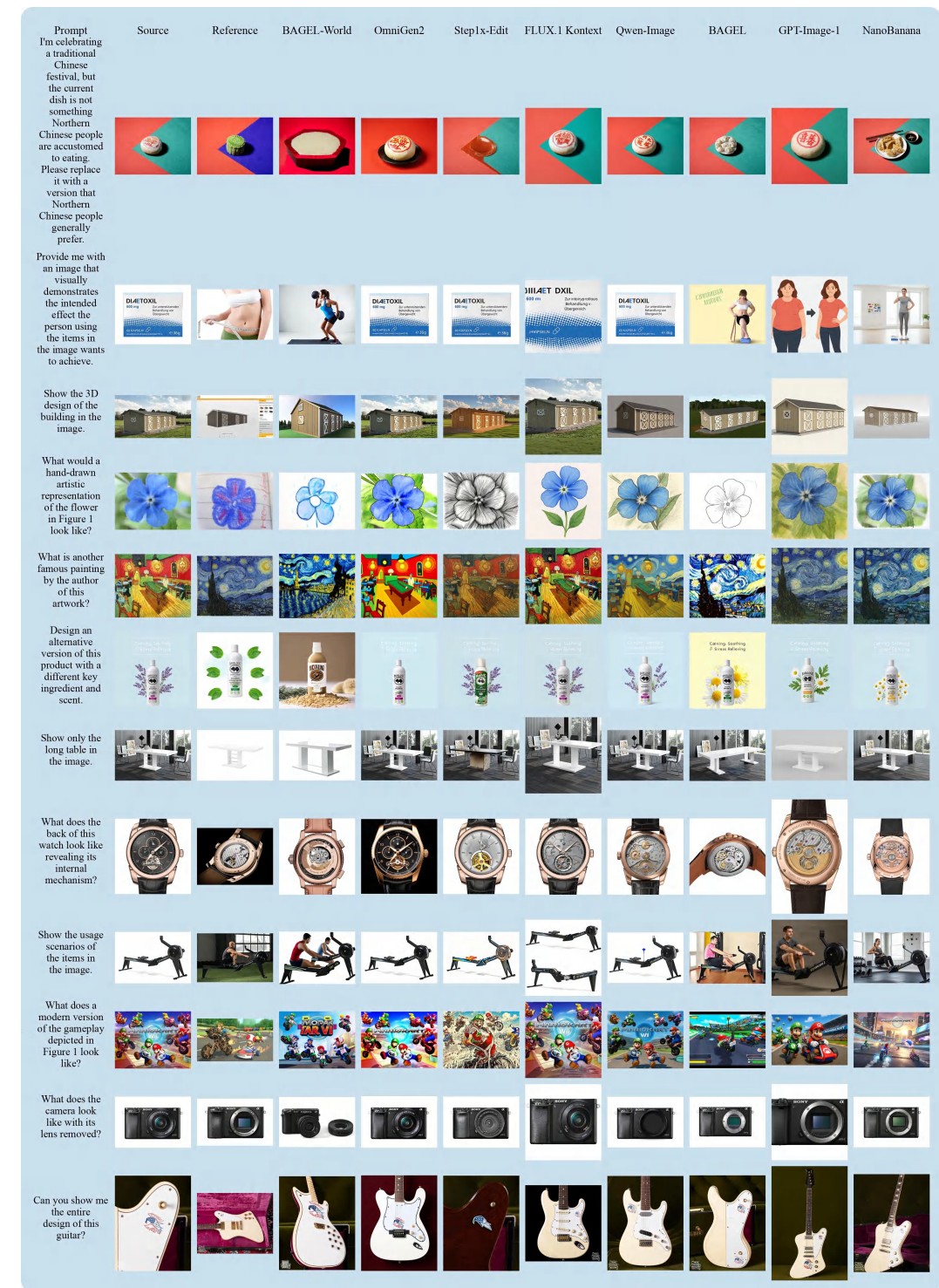

Figure 8: Comprehensive visualization of model performance on IntelligentBench (Subset Design, part 4/8).

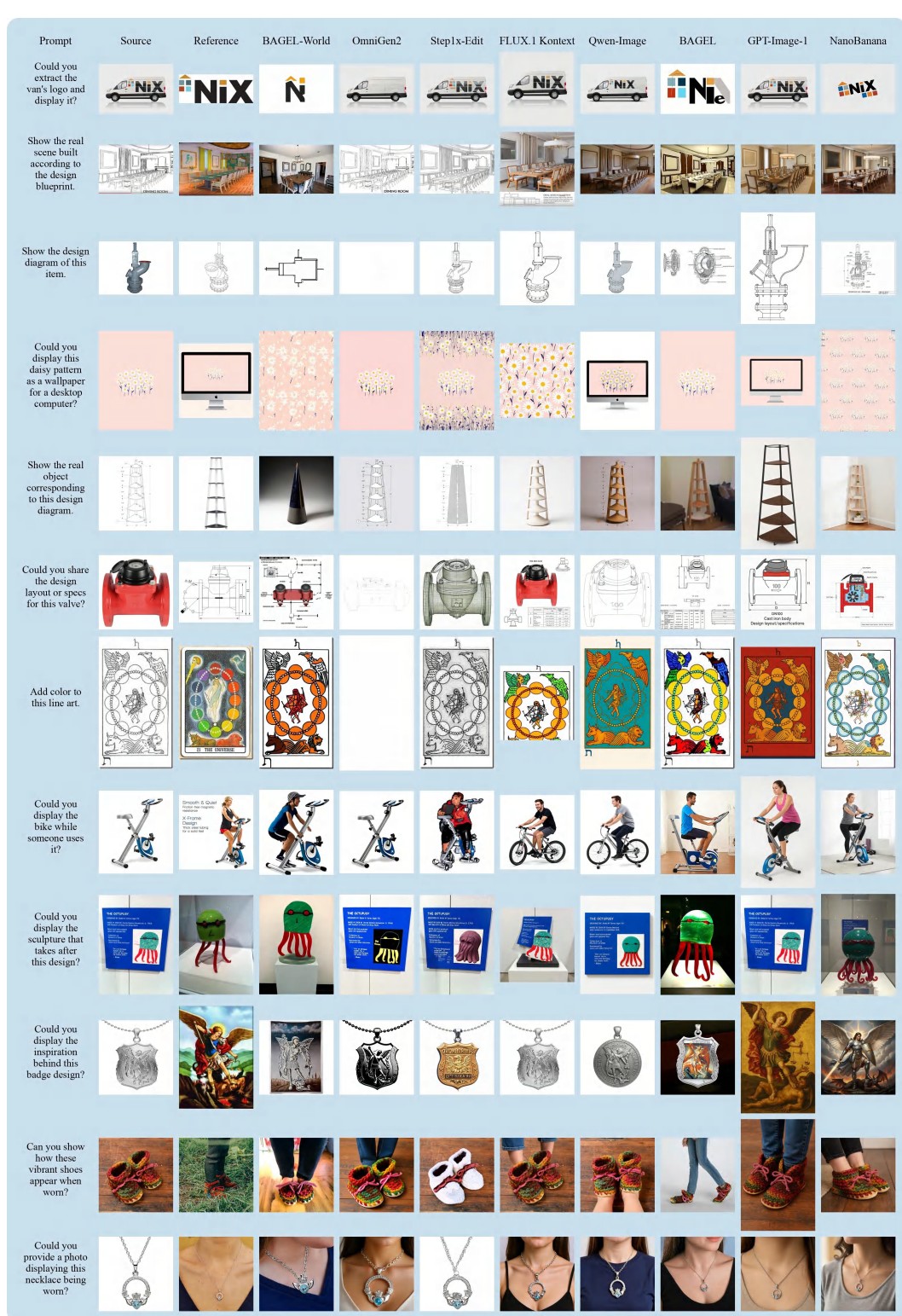

Figure 9: Comprehensive visualization of model performance on IntelligentBench (Subset Design, part 5/8).

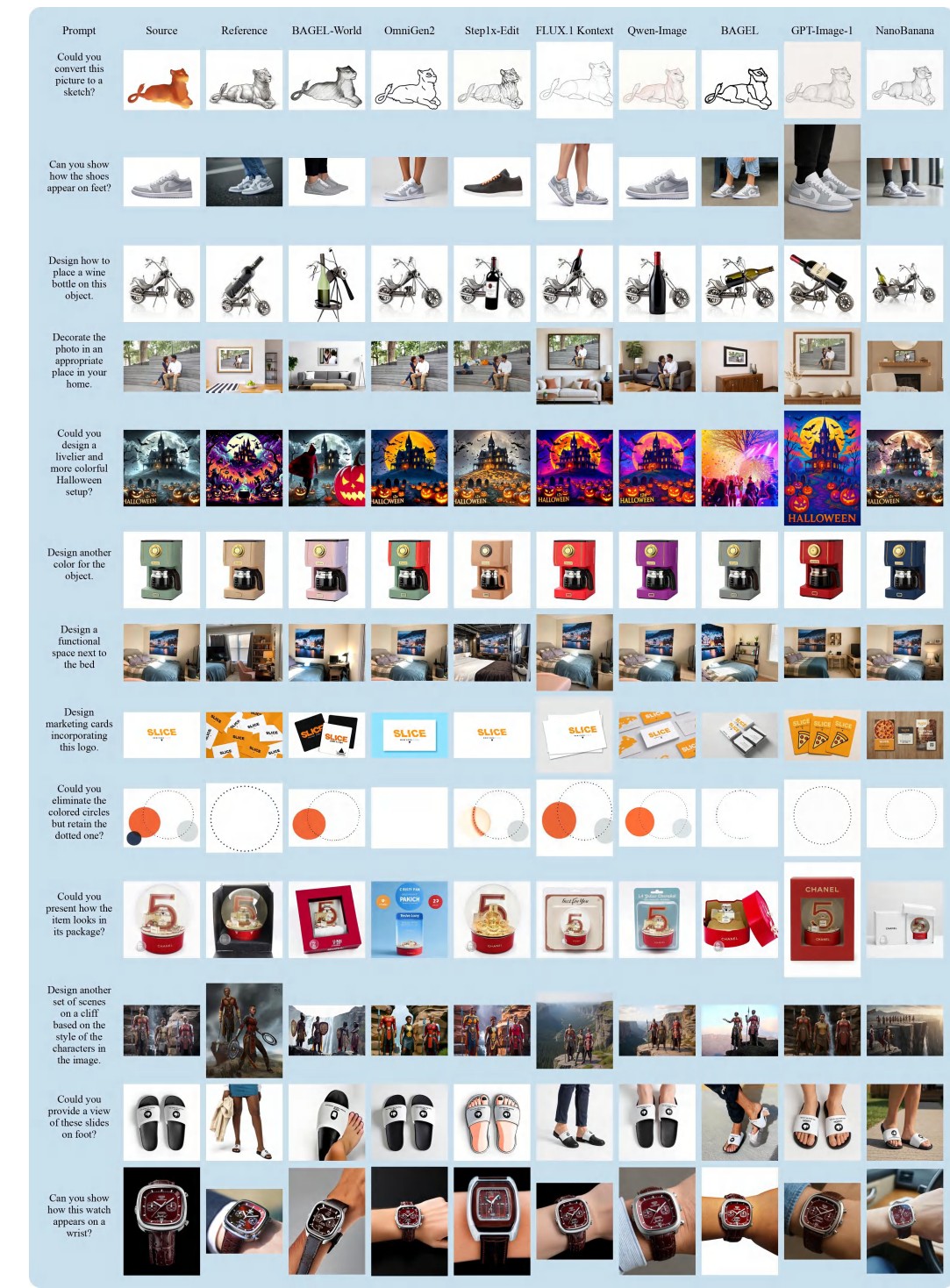

Figure 10: Comprehensive visualization of model performance on IntelligentBench (Subset Design, part 6/8).

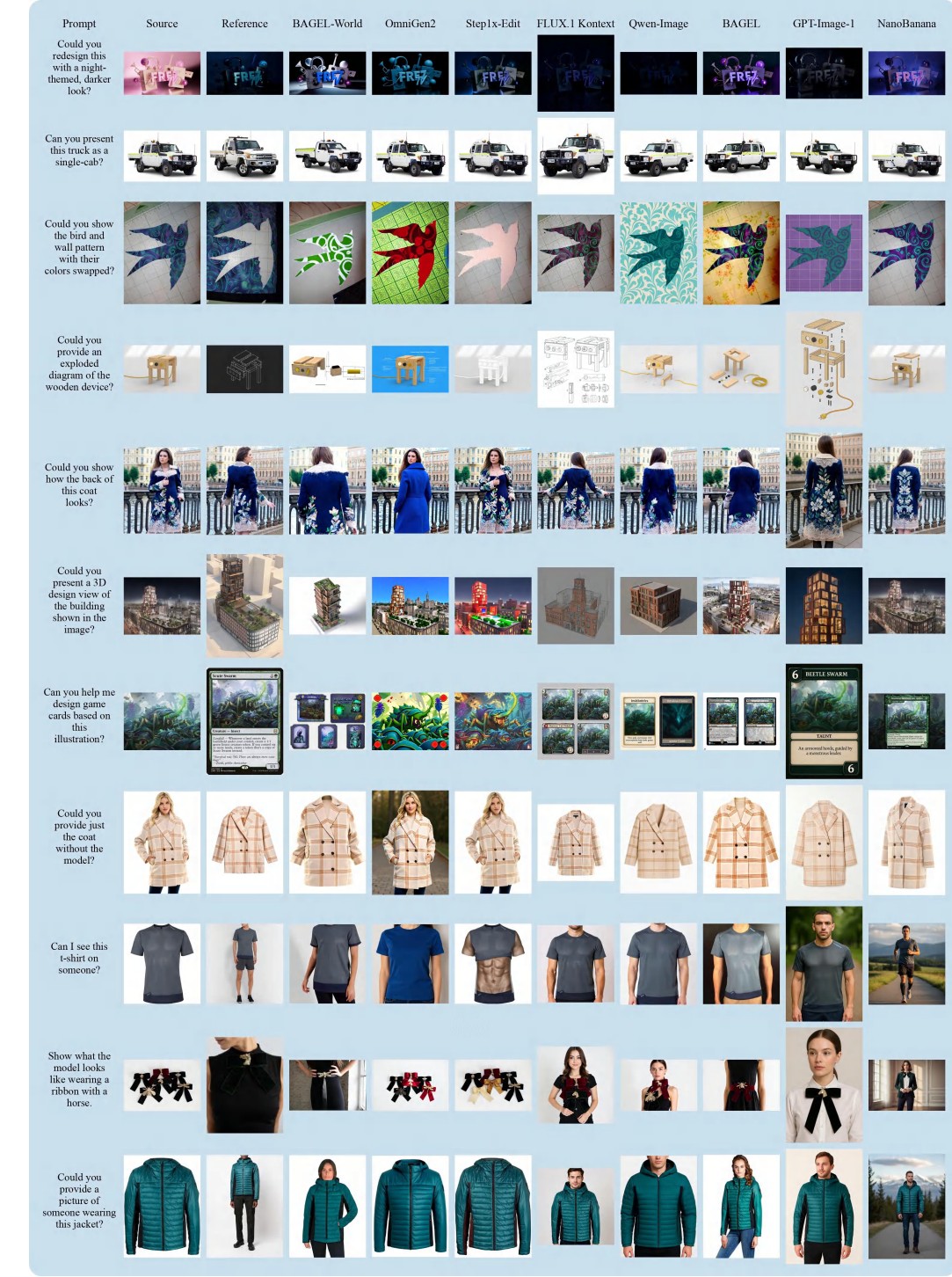

Figure 11: Comprehensive visualization of model performance on IntelligentBench (Subset Design, part 7/8).

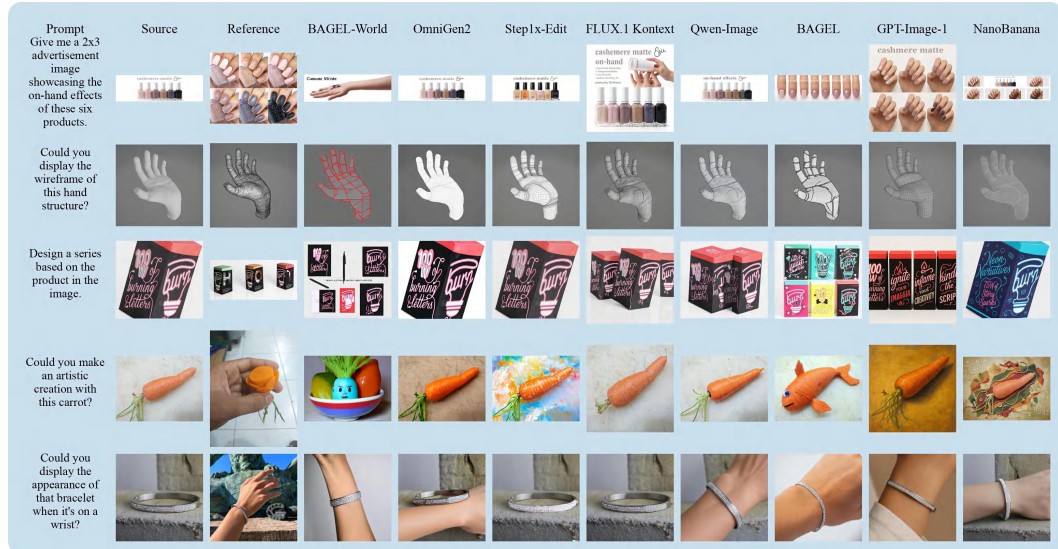

Figure 12: Comprehensive visualization of model performance on IntelligentBench (Subset Design, part 8/8).

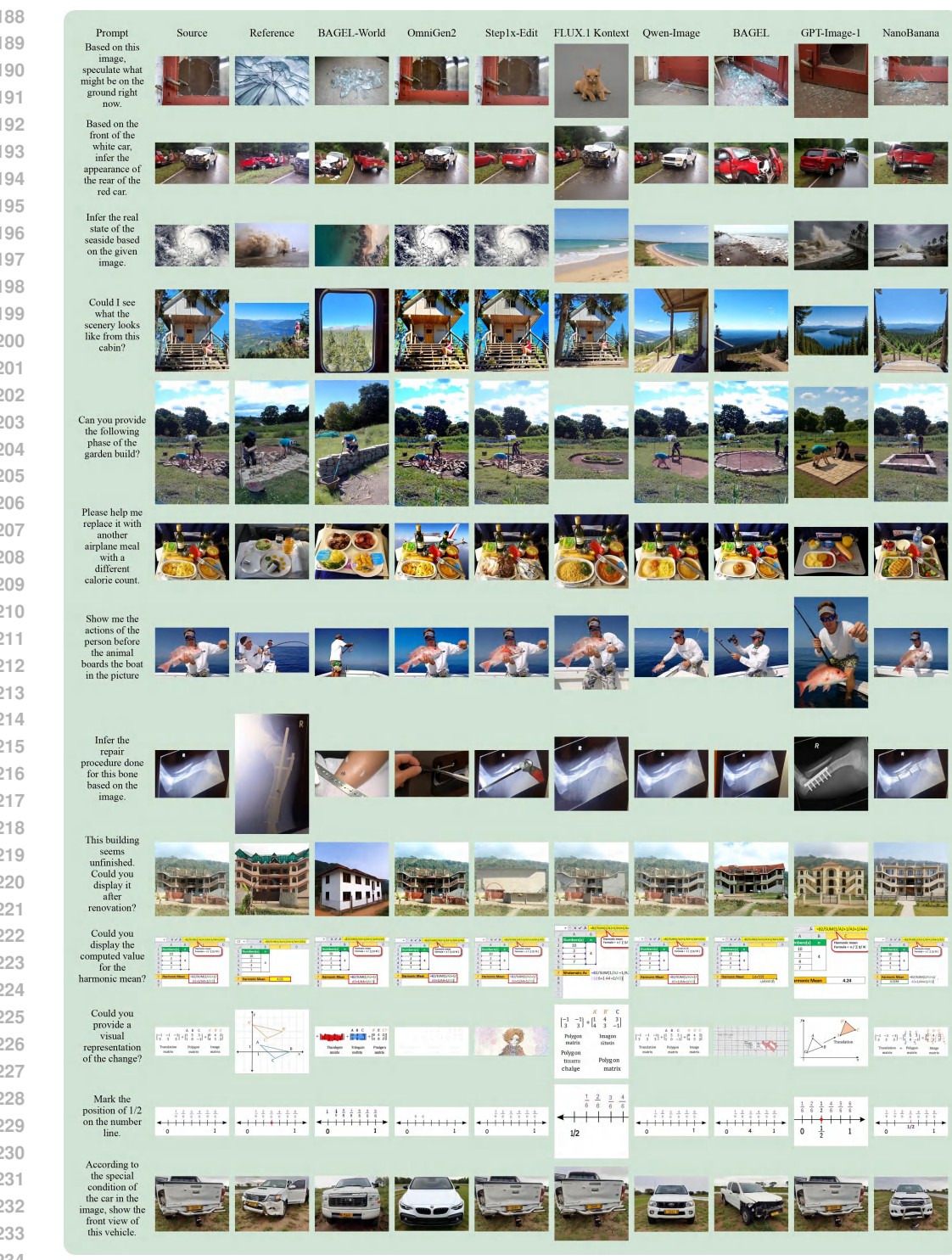

Figure 13: Comprehensive visualization of model performance on IntelligentBench (Subset Reasoning, part 1/8).

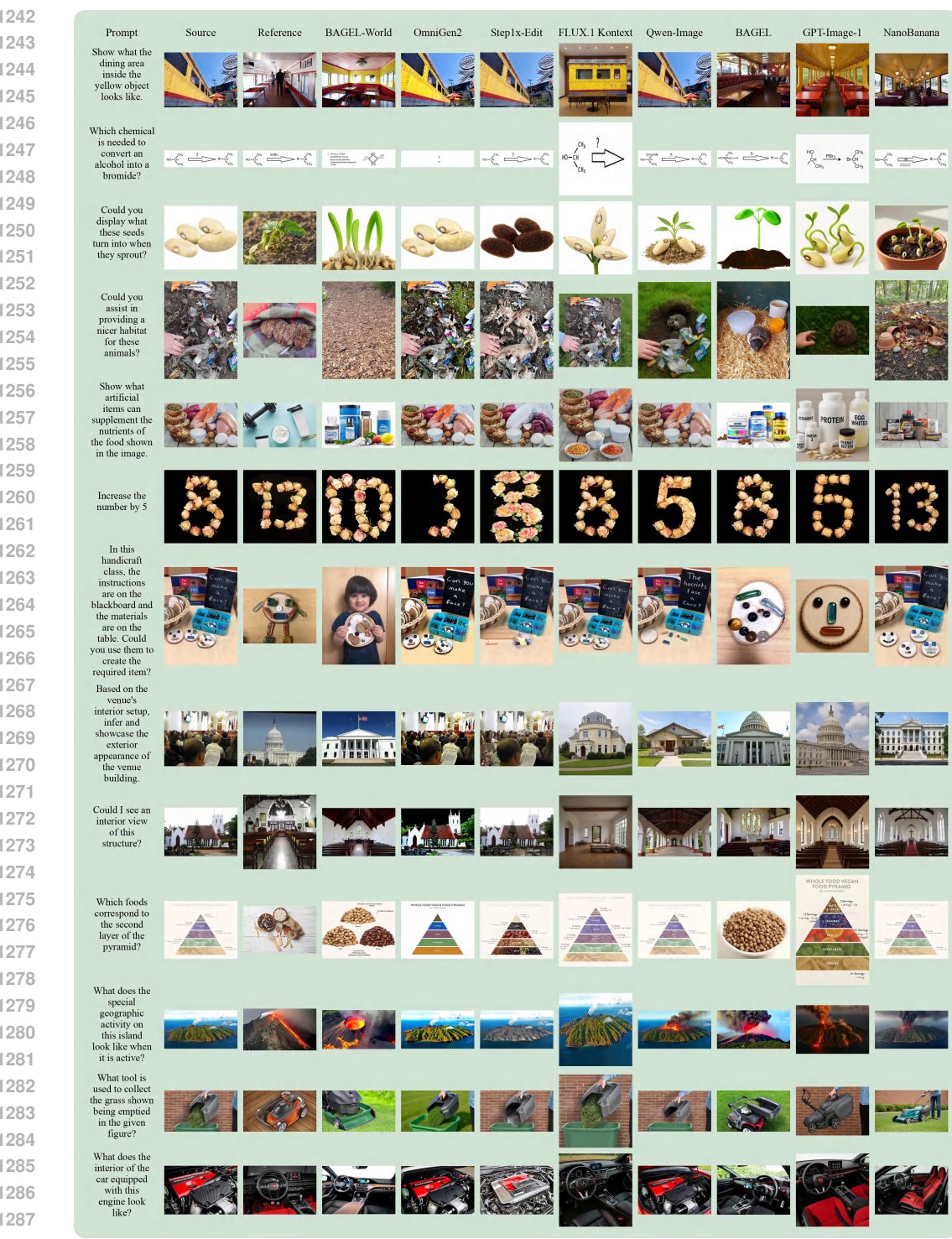

Figure 14: Comprehensive visualization of model performance on IntelligentBench (Subset Reasoning, part 2/8).

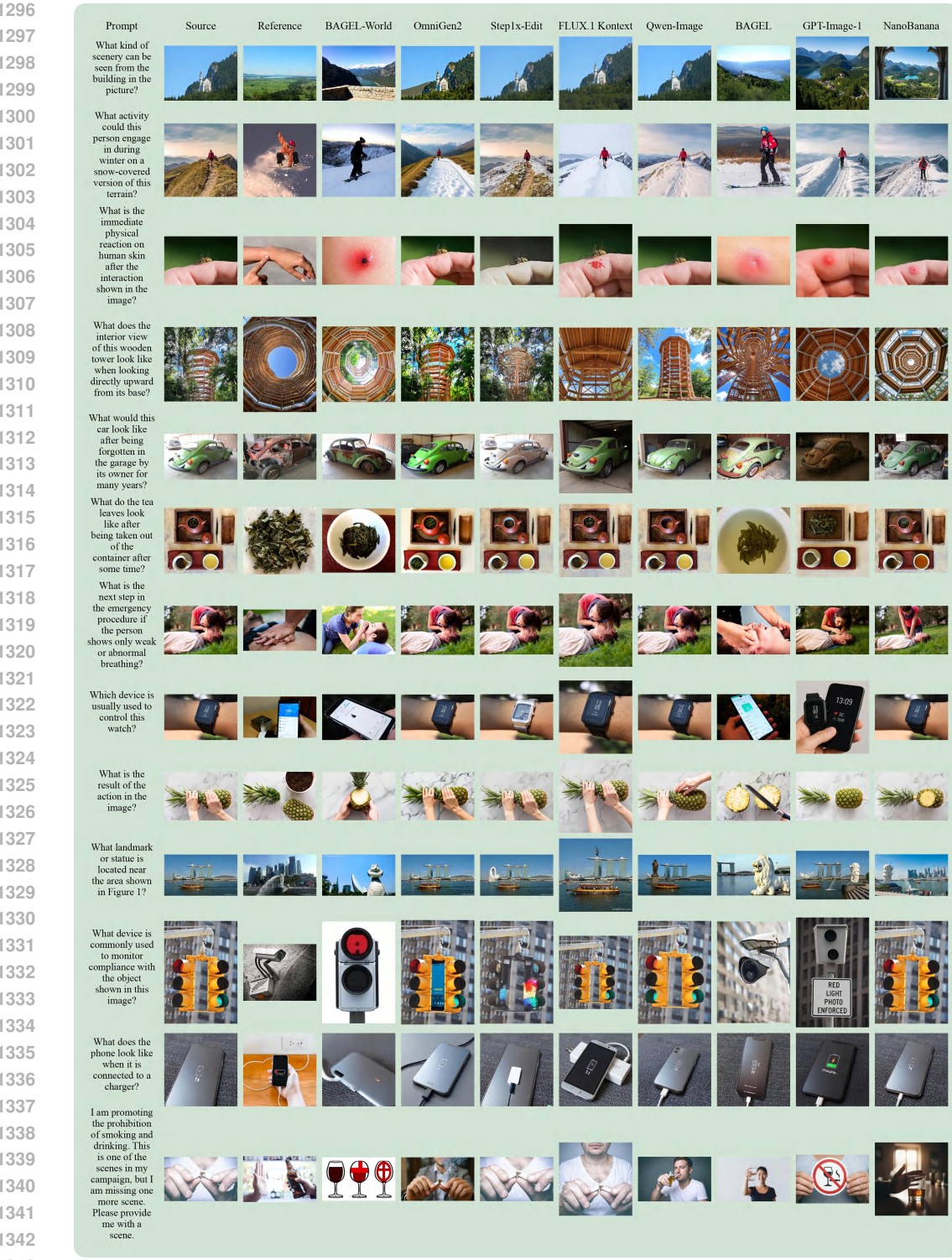

Figure 15: Comprehensive visualization of model performance on IntelligentBench (Subset Reasoning, part 3/8).

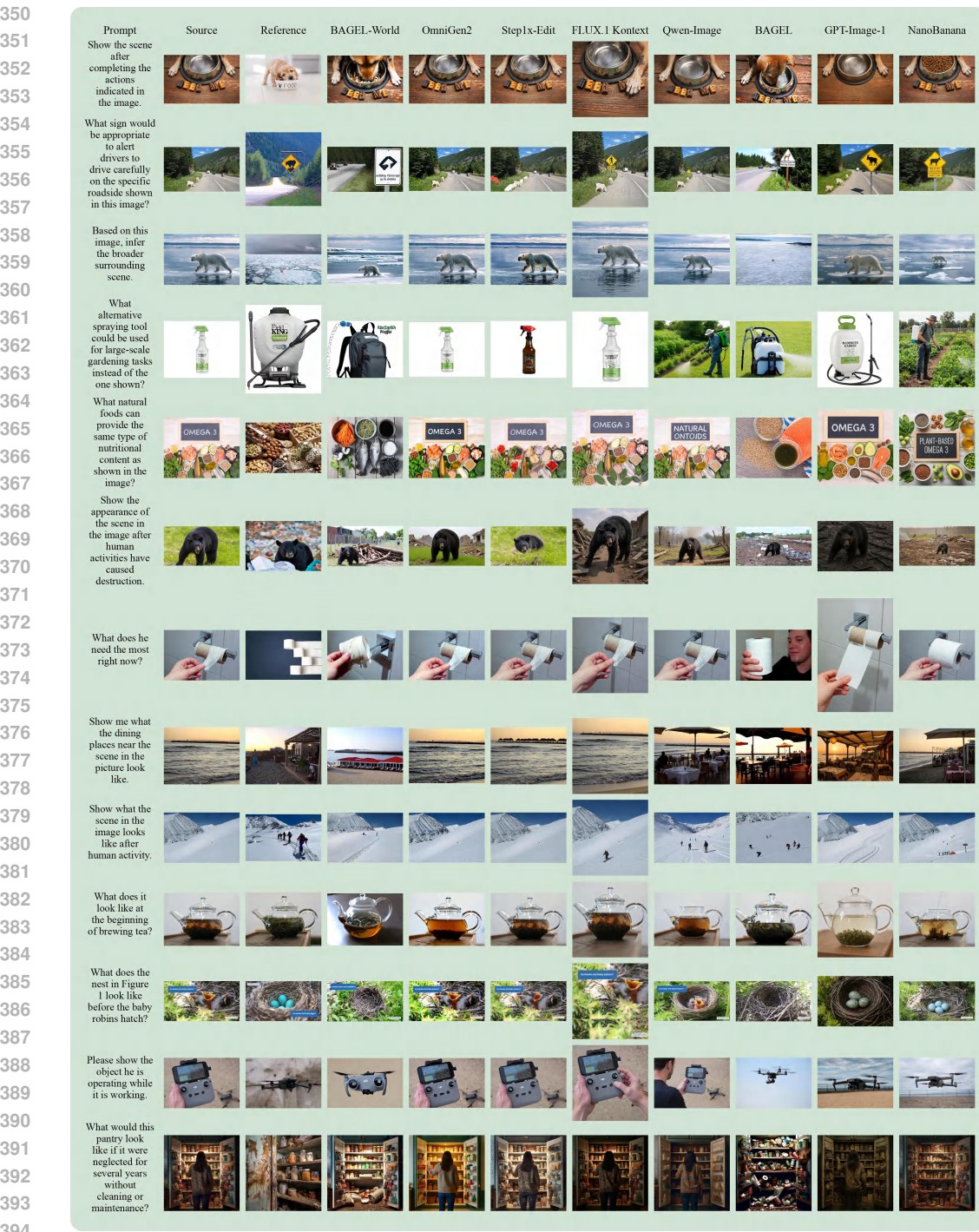

Figure 16: Comprehensive visualization of model performance on IntelligentBench (Subset Reasoning, part 4/8).

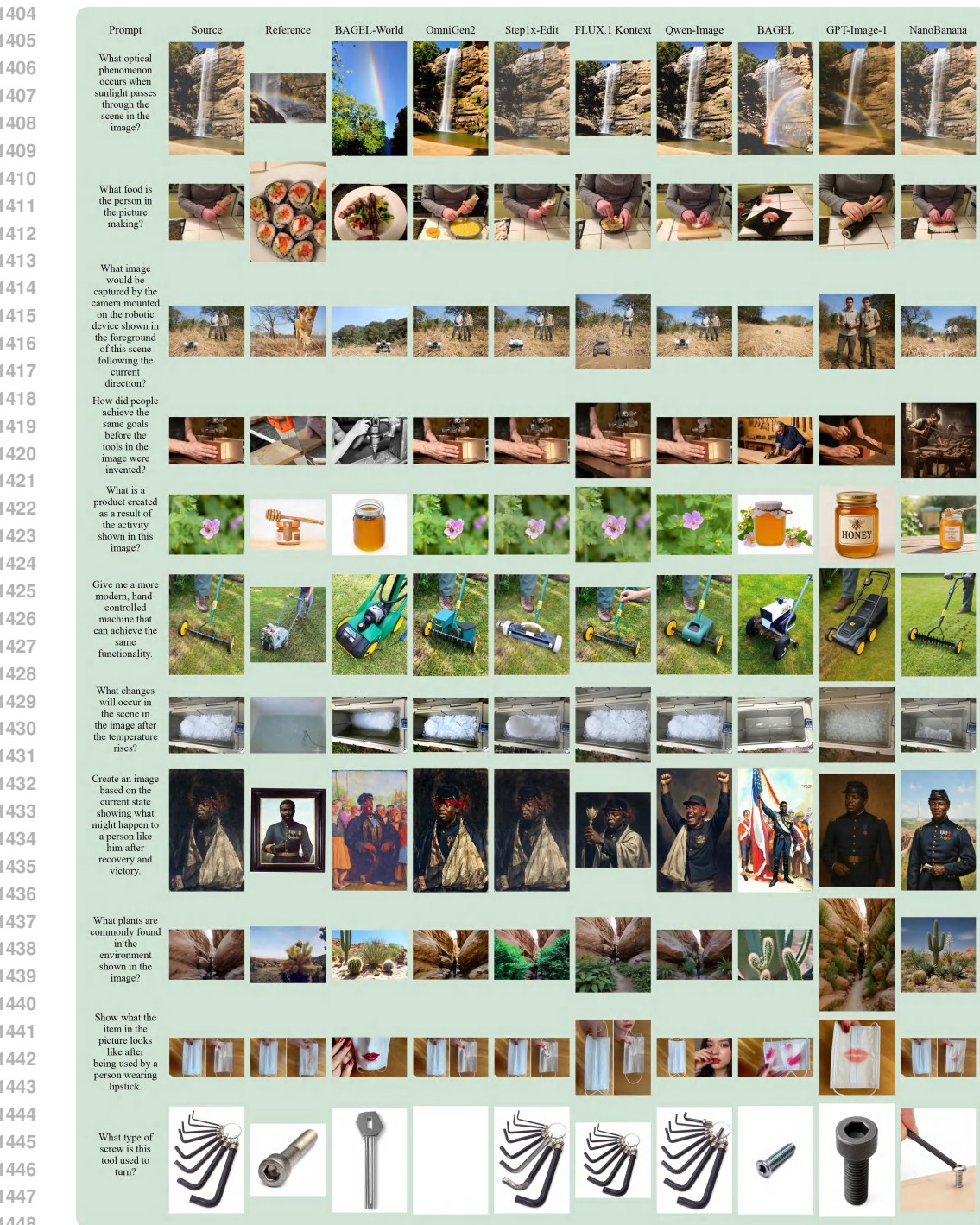

Figure 17: Comprehensive visualization of model performance on IntelligentBench (Subset Reasoning, part 5/8).

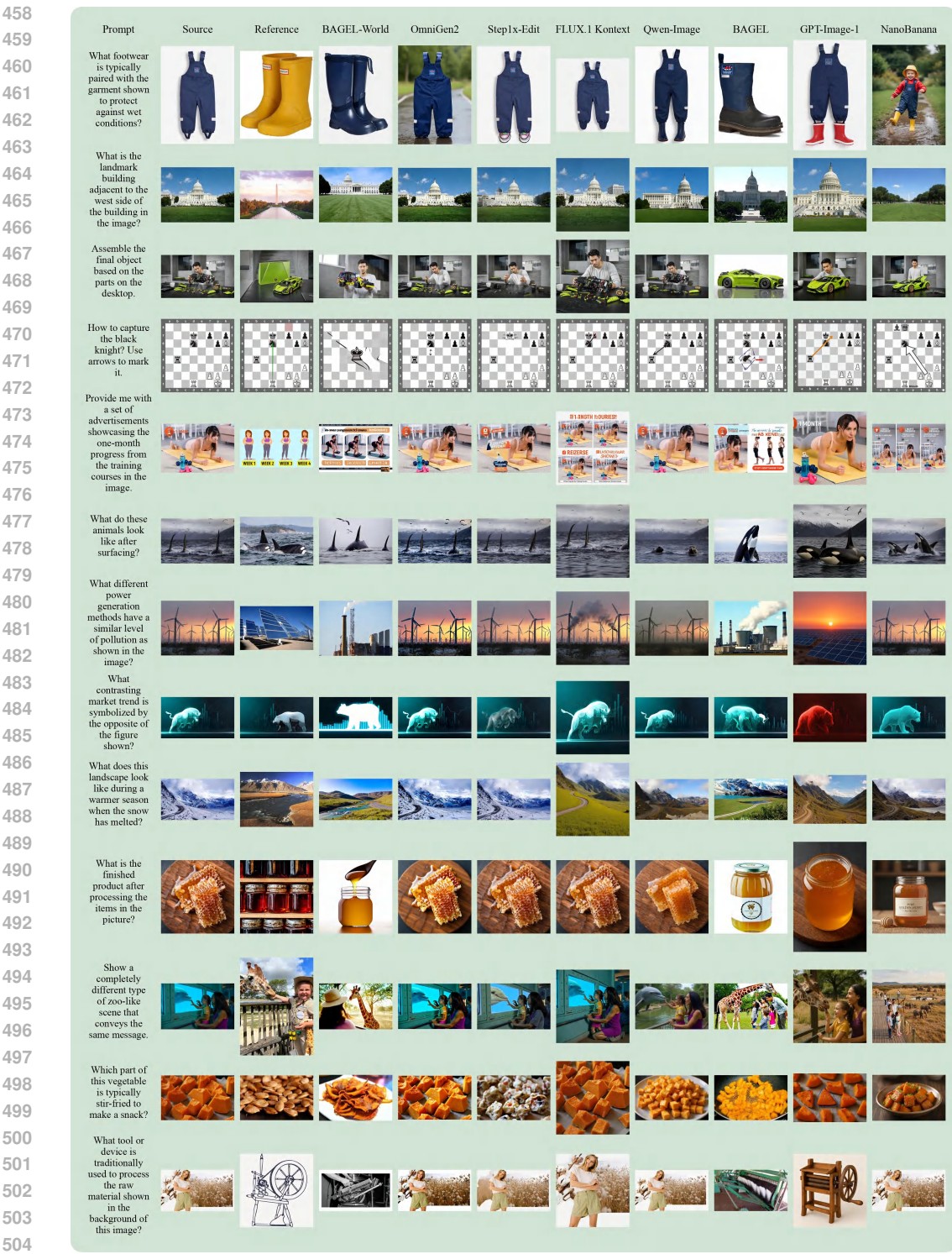

Figure 18: Comprehensive visualization of model performance on IntelligentBench (Subset Reasoning, part 6/8).

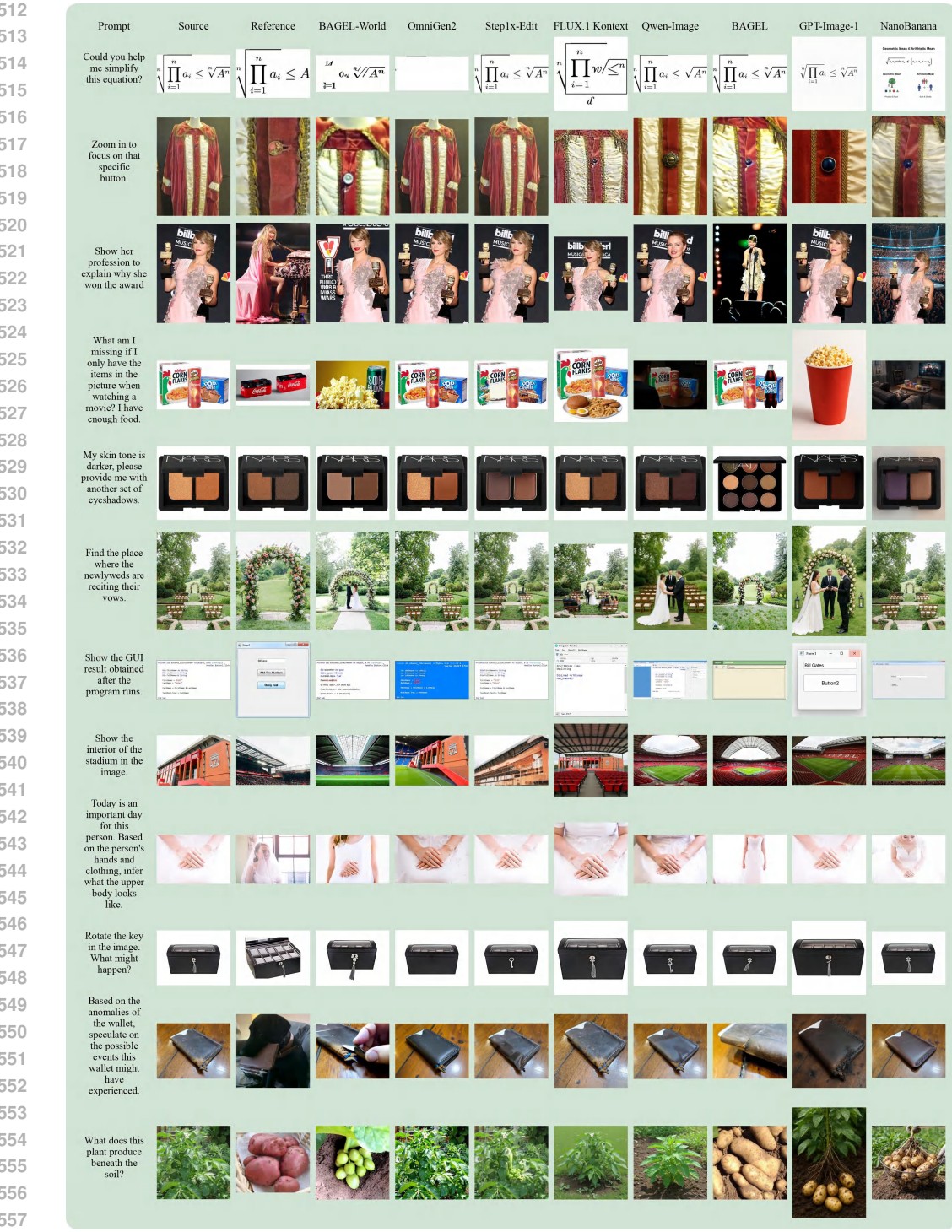

Figure 19: Comprehensive visualization of model performance on IntelligentBench (Subset Reasoning, part 7/8).

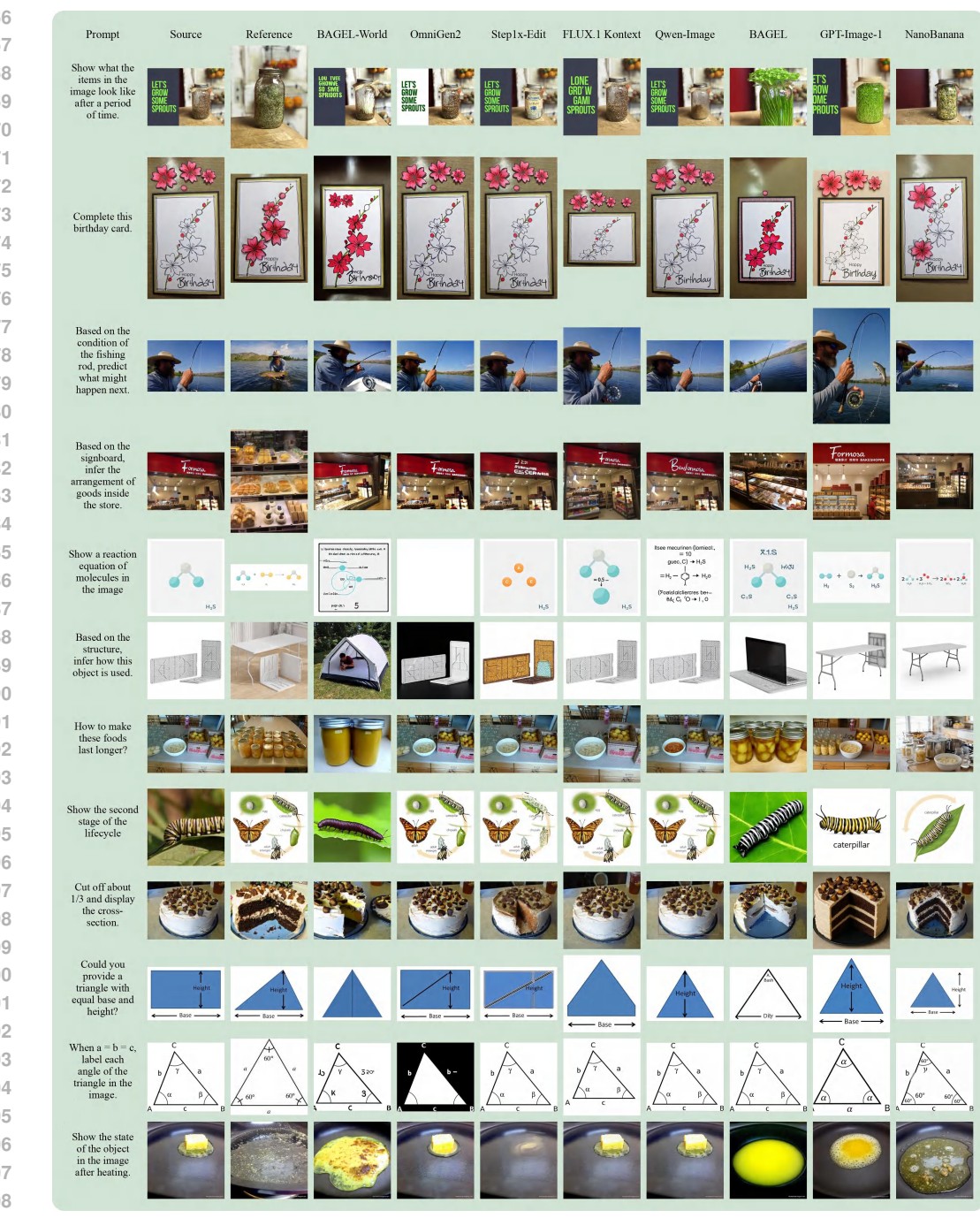

Figure 20: Comprehensive visualization of model performance on IntelligentBench (Subset Reasoning, part 8/8).

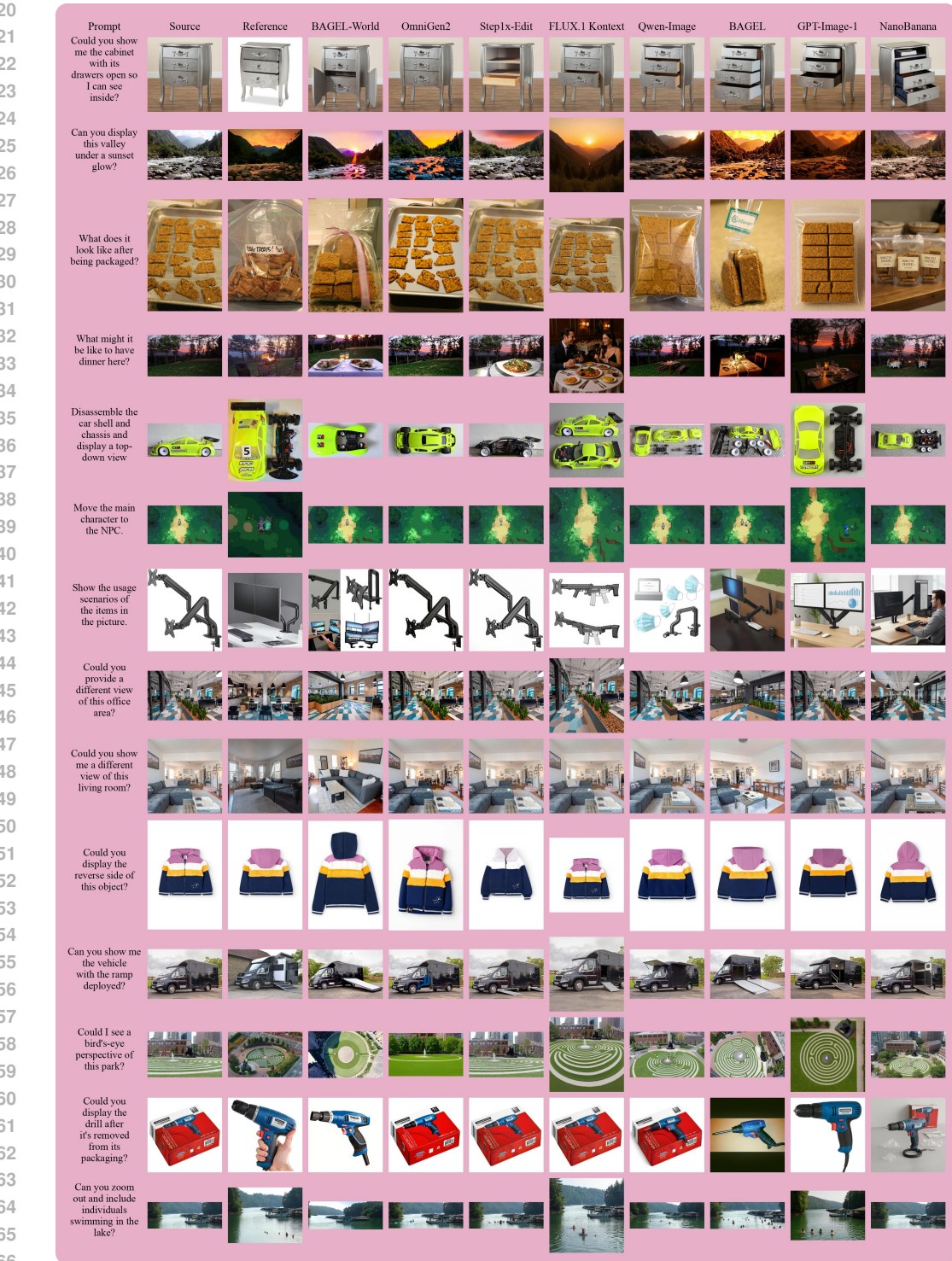

Figure 21: Comprehensive visualization of model performance on IntelligentBench (Subset World knowledge, part 1/13).

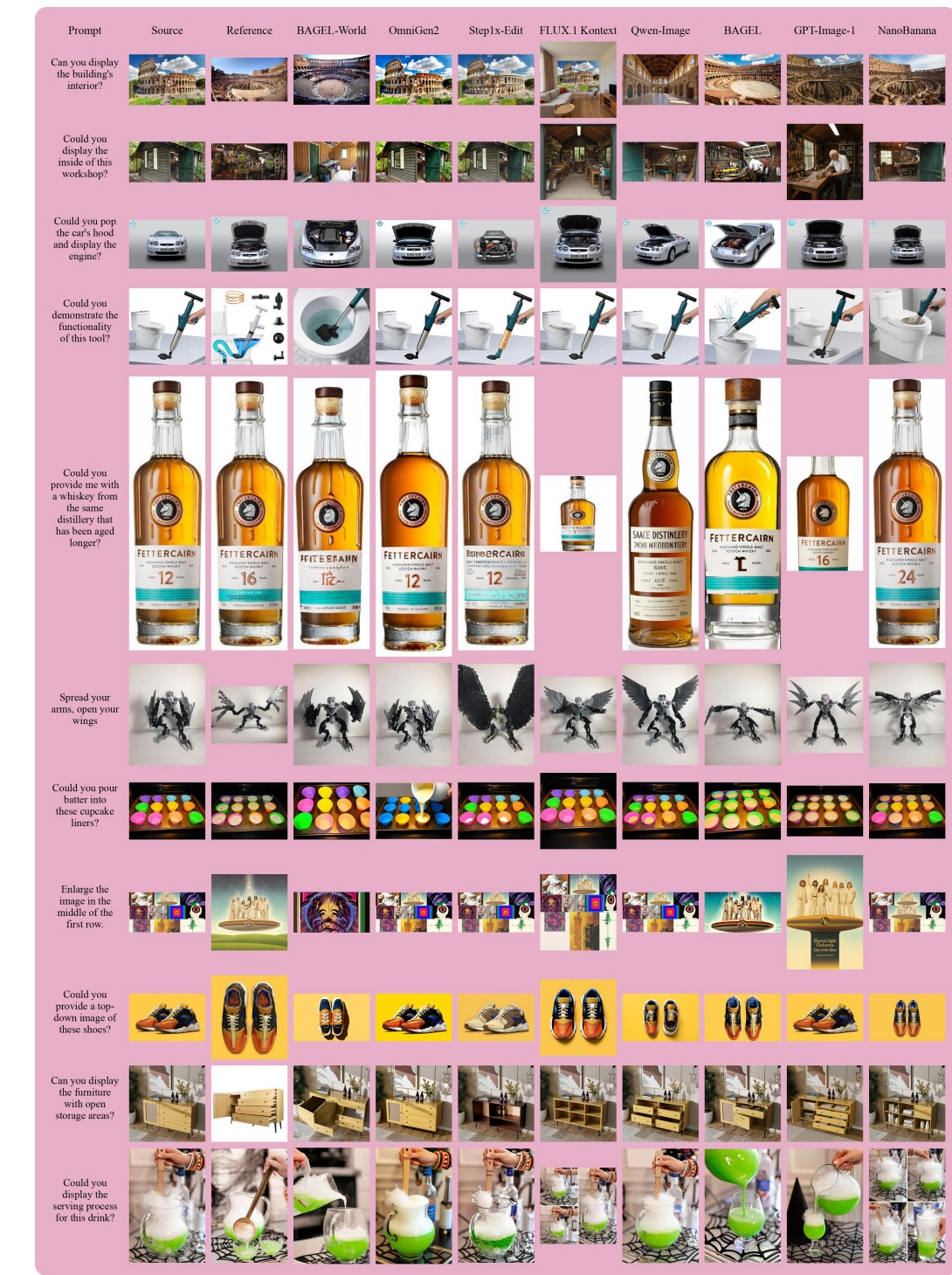

Figure 22: Comprehensive visualization of model performance on IntelligentBench (Subset World knowledge, part 2/13).

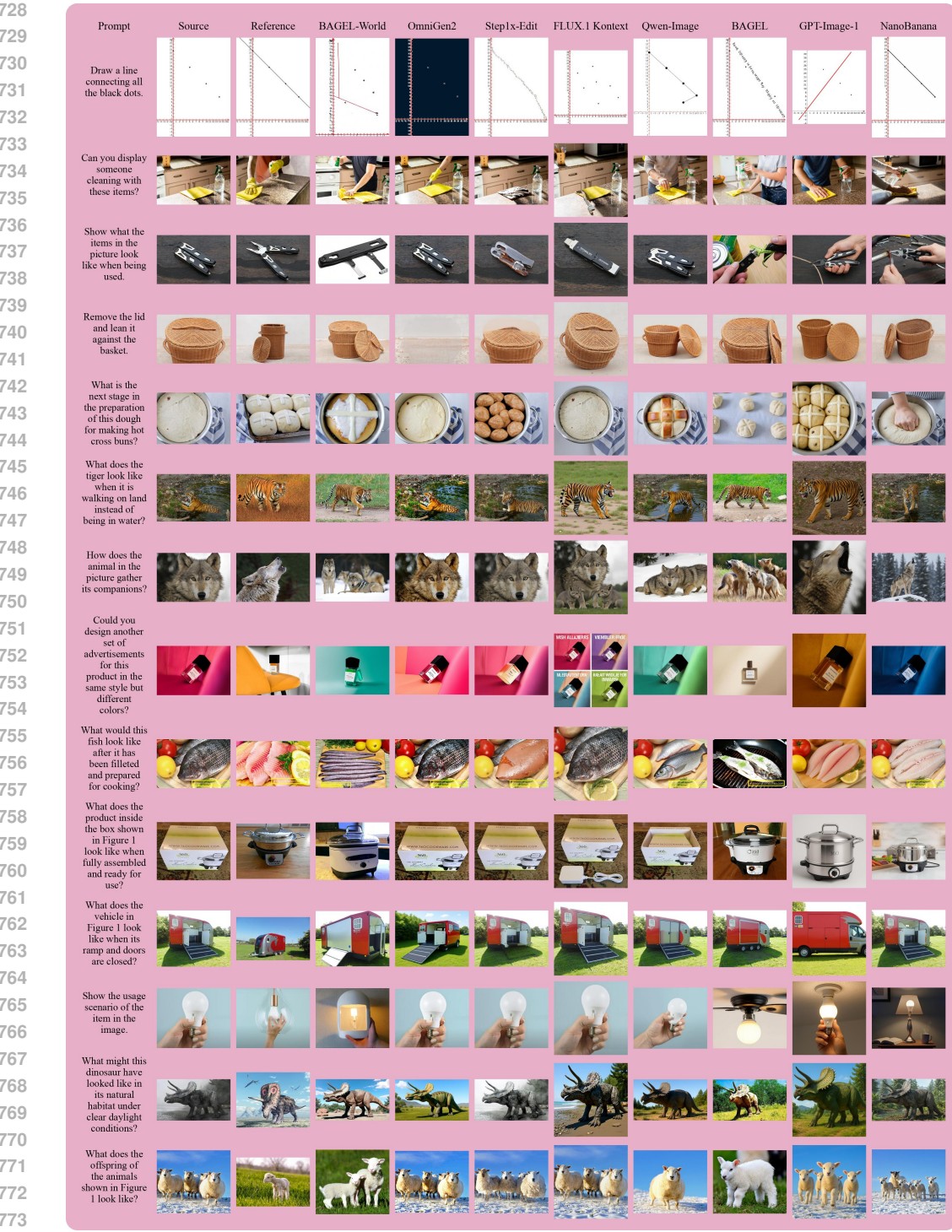

Figure 23: Comprehensive visualization of model performance on IntelligentBench (Subset World knowledge, part 3/13).

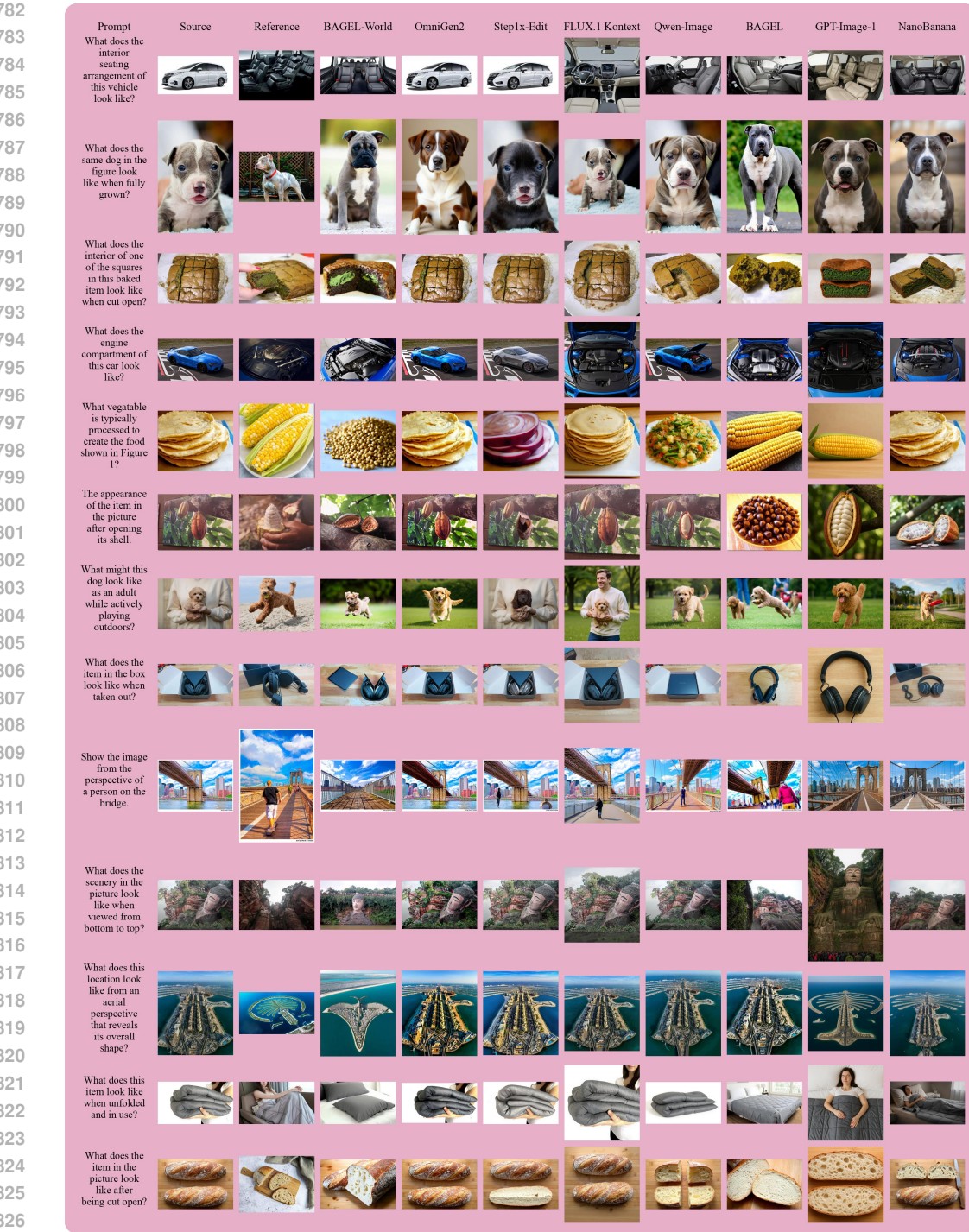

Figure 24: Comprehensive visualization of model performance on IntelligentBench (Subset World knowledge, part 4/13).

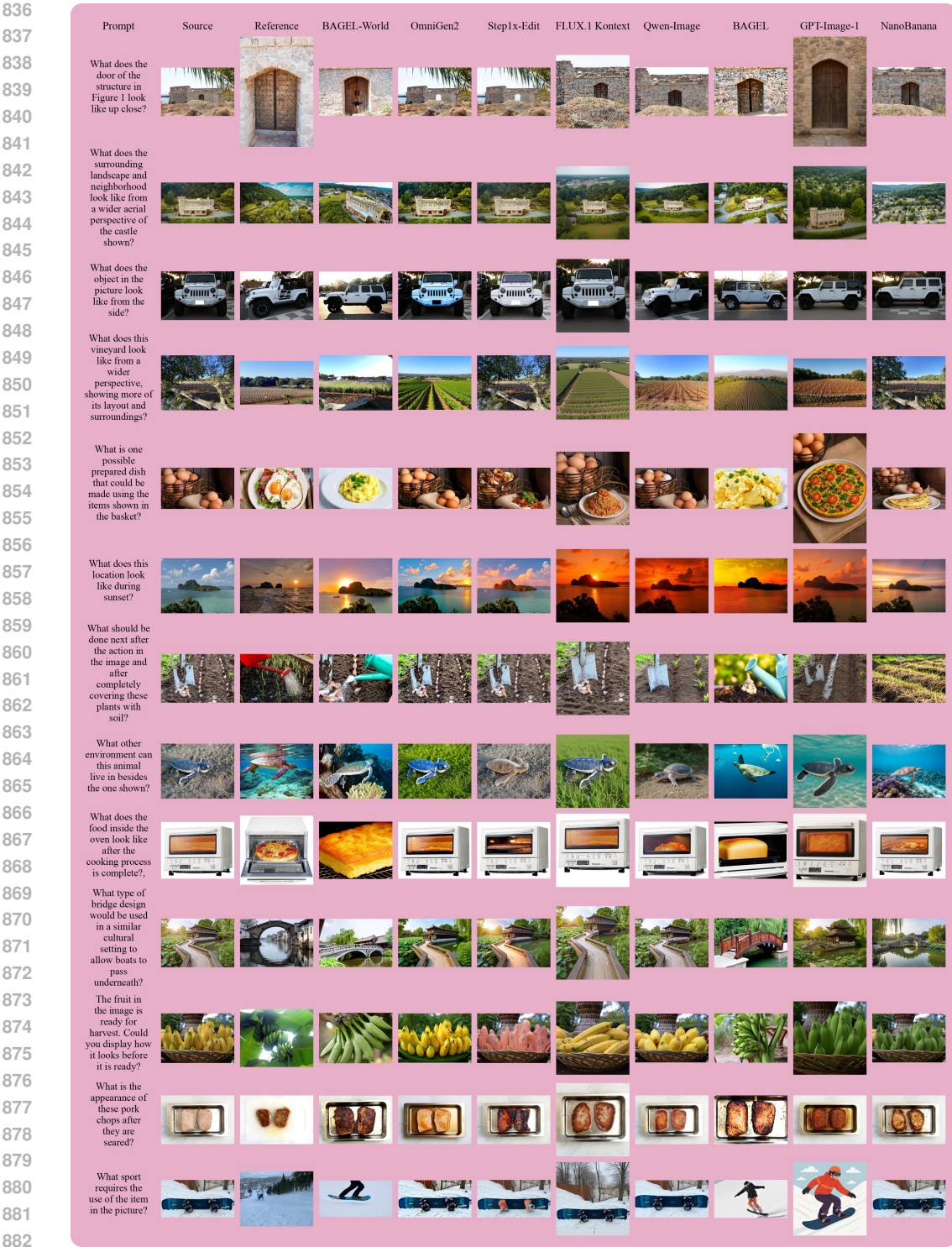

Figure 25: Comprehensive visualization of model performance on IntelligentBench (Subset World knowledge, part 5/13).

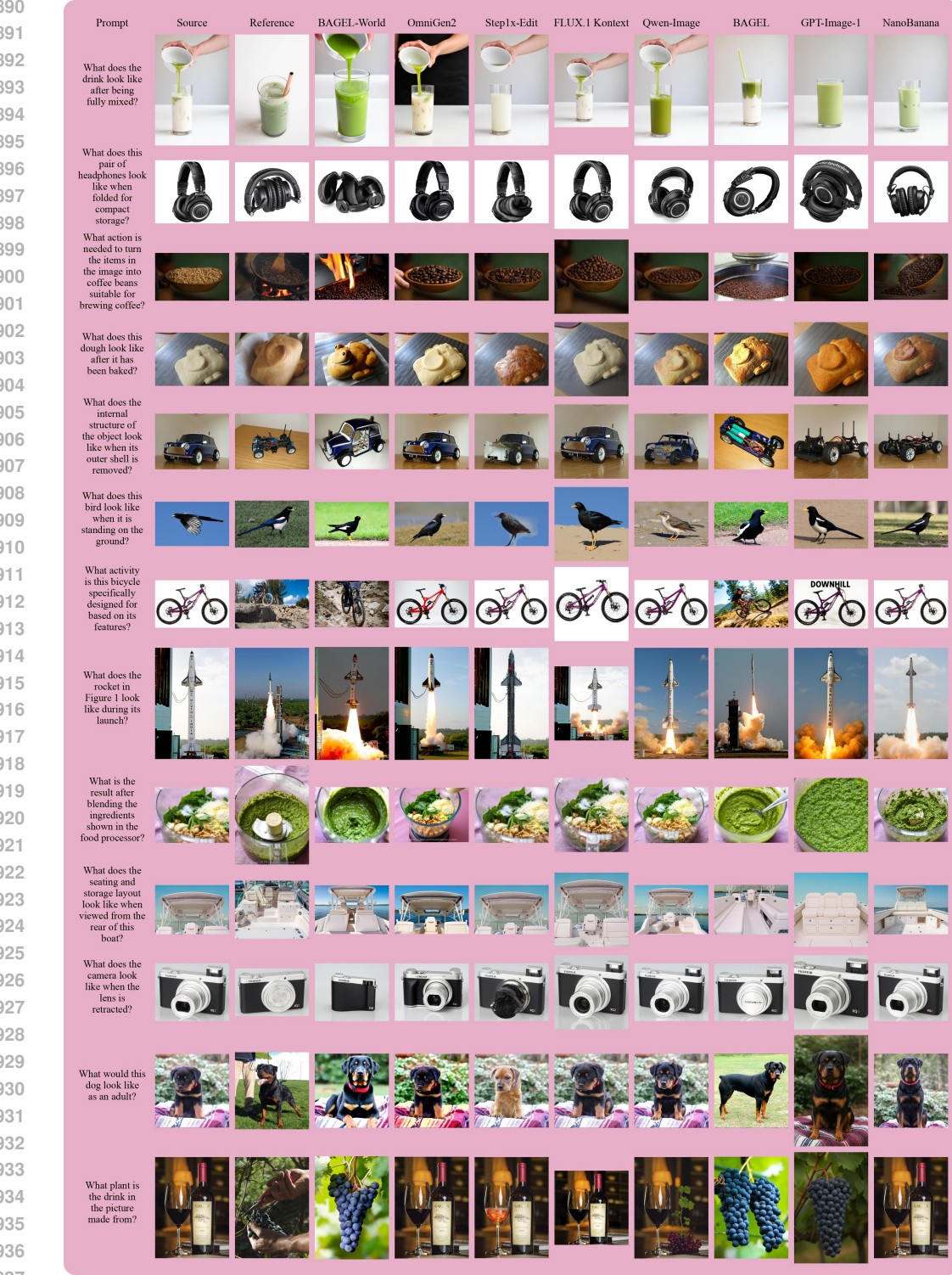

Figure 26: Comprehensive visualization of model performance on IntelligentBench (Subset World knowledge, part 6/13).

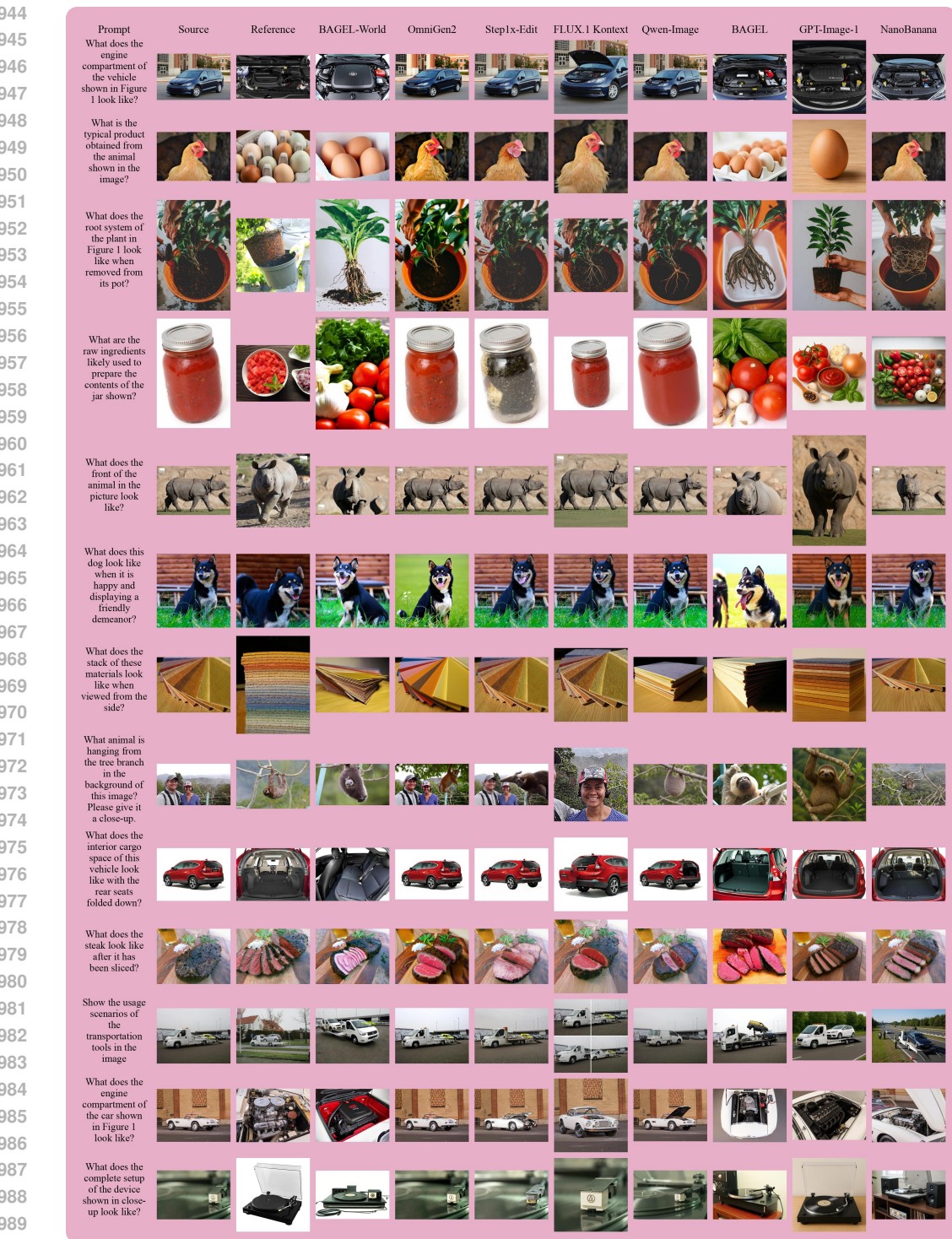

Figure 27: Comprehensive visualization of model performance on IntelligentBench (Subset World knowledge, part 7/13).

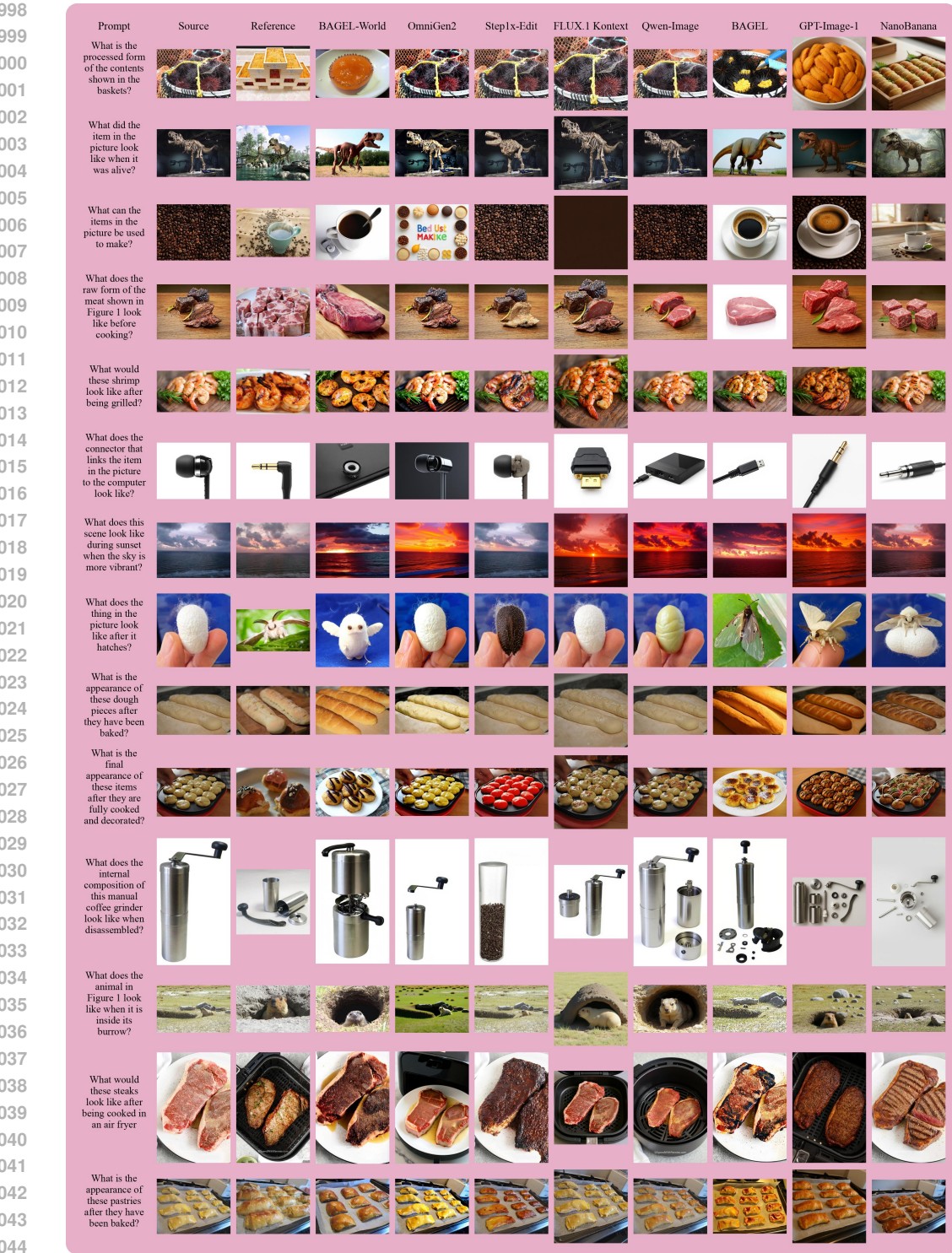

Figure 28: Comprehensive visualization of model performance on IntelligentBench (Subset World knowledge, part 8/13).

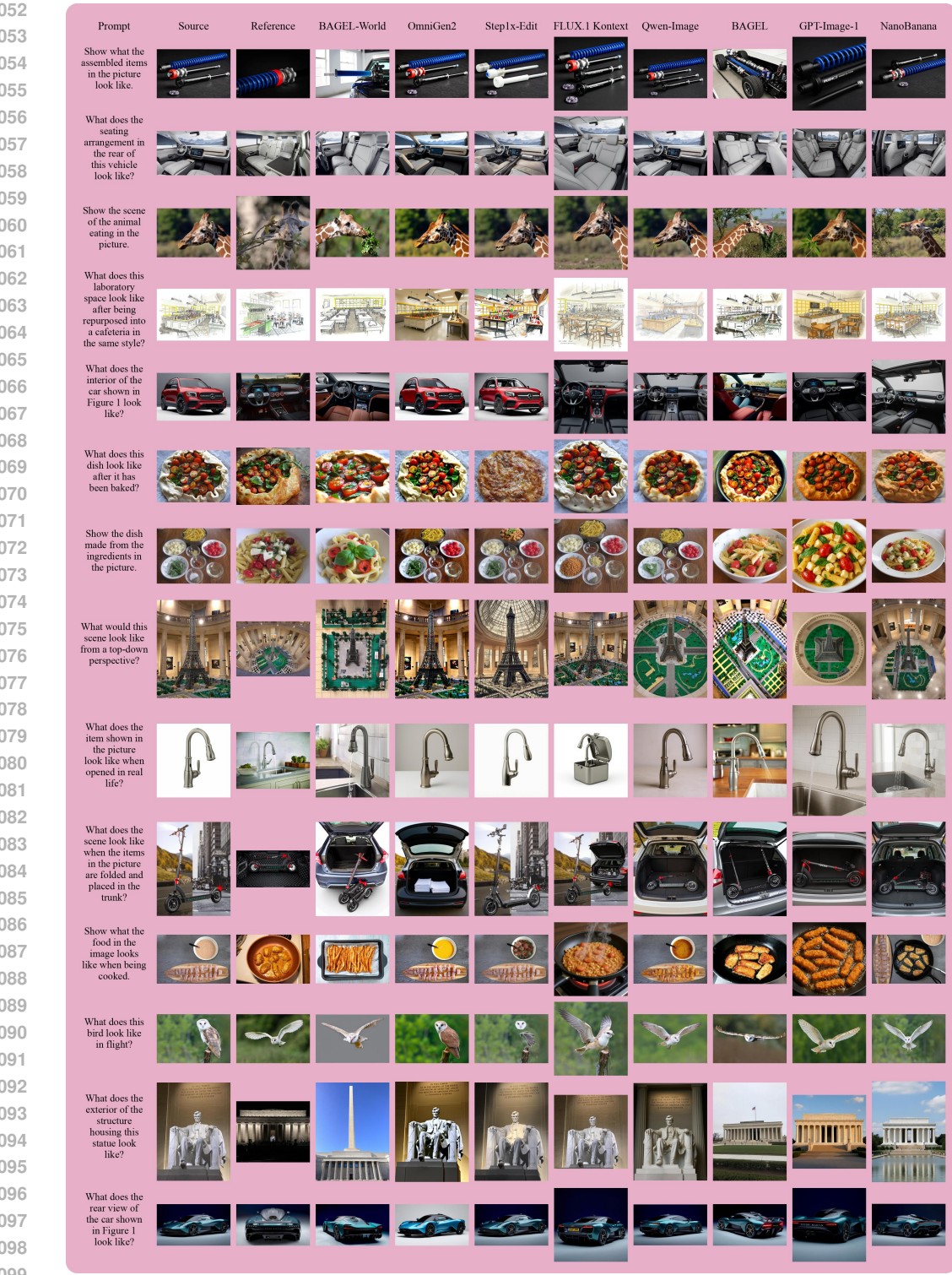

Figure 29: Comprehensive visualization of model performance on IntelligentBench (Subset World knowledge, part 9/13).

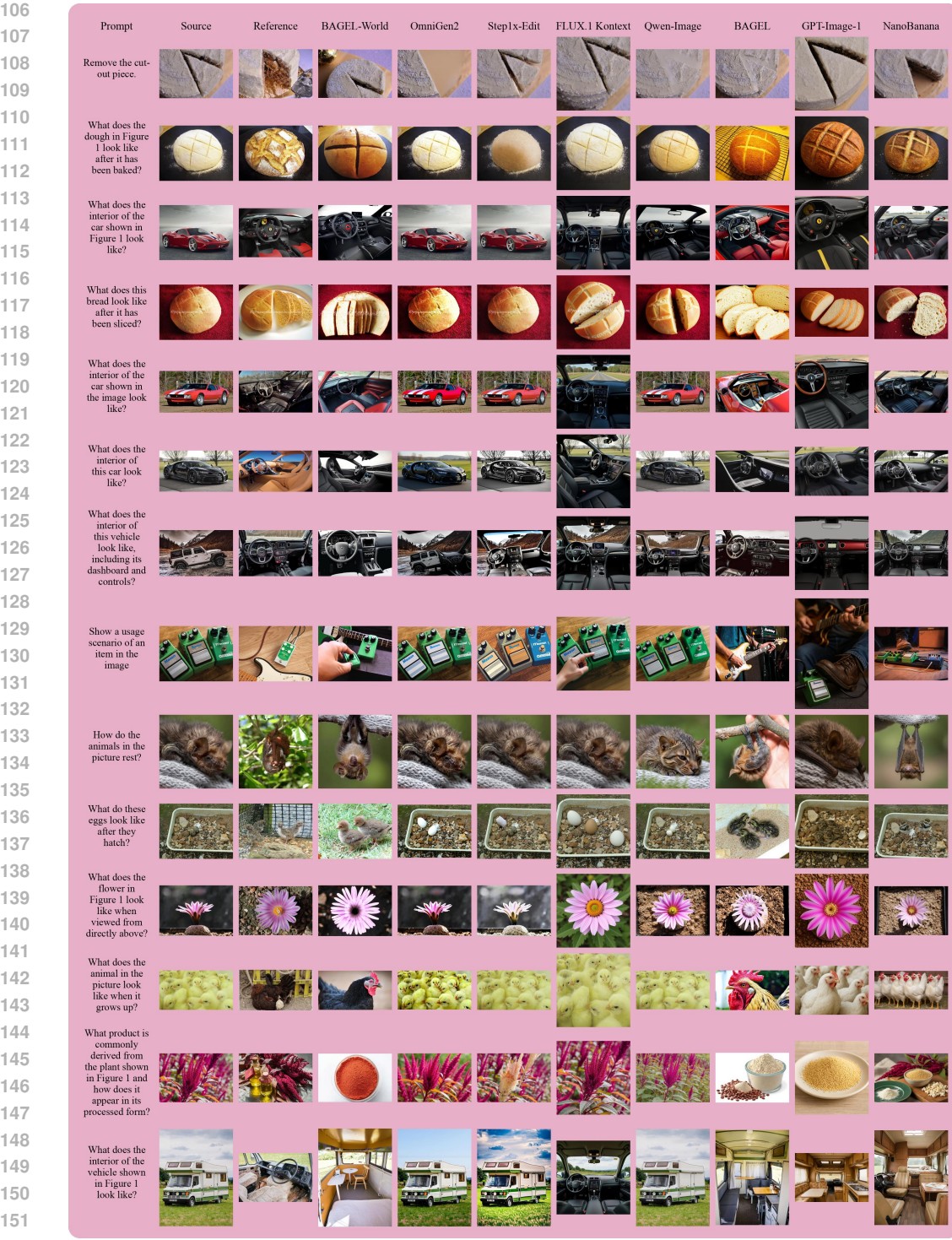

Figure 30: Comprehensive visualization of model performance on IntelligentBench (Subset World knowledge, part 10/13).

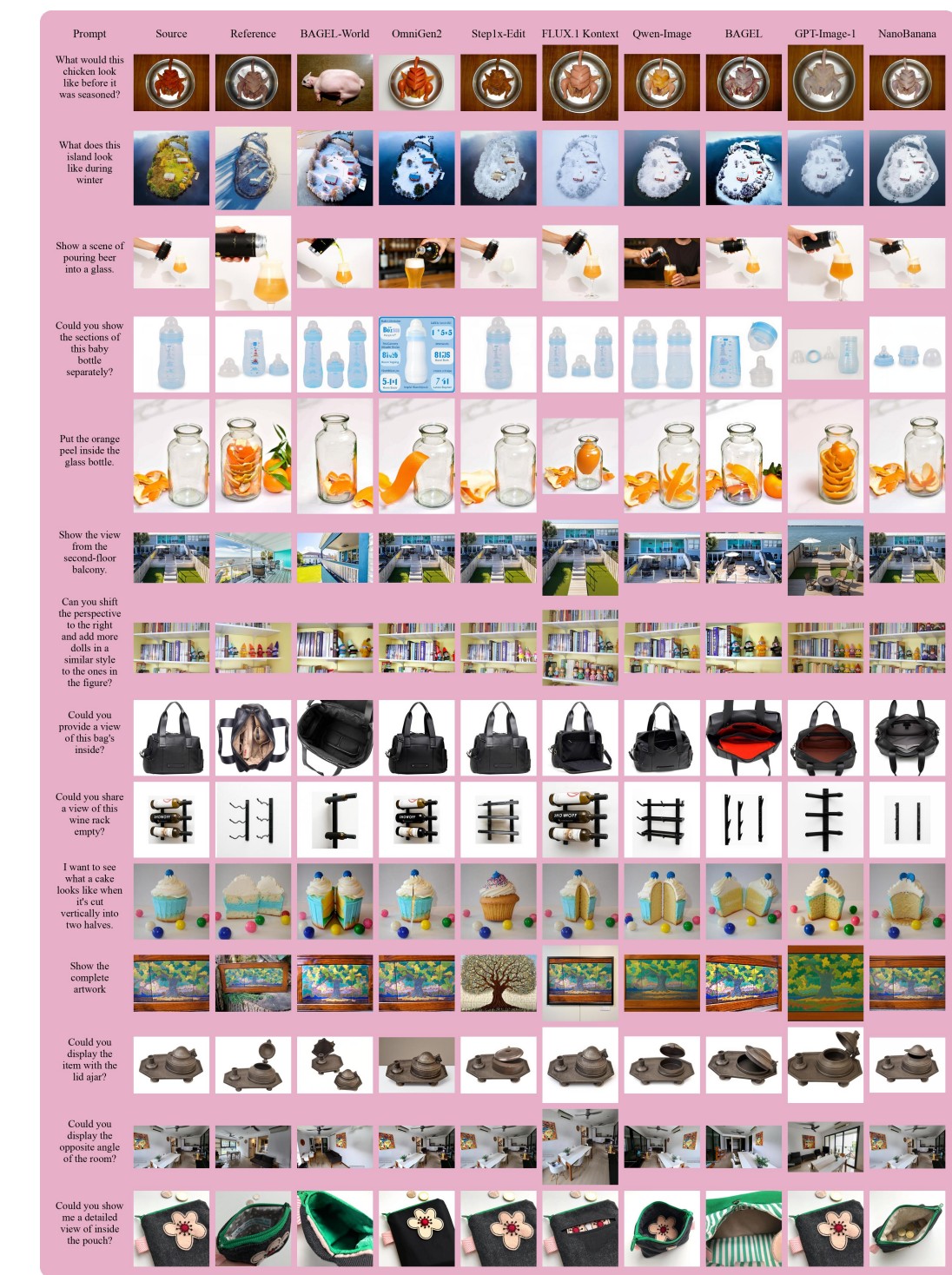

Figure 31: Comprehensive visualization of model performance on IntelligentBench (Subset World knowledge, part 11/13).

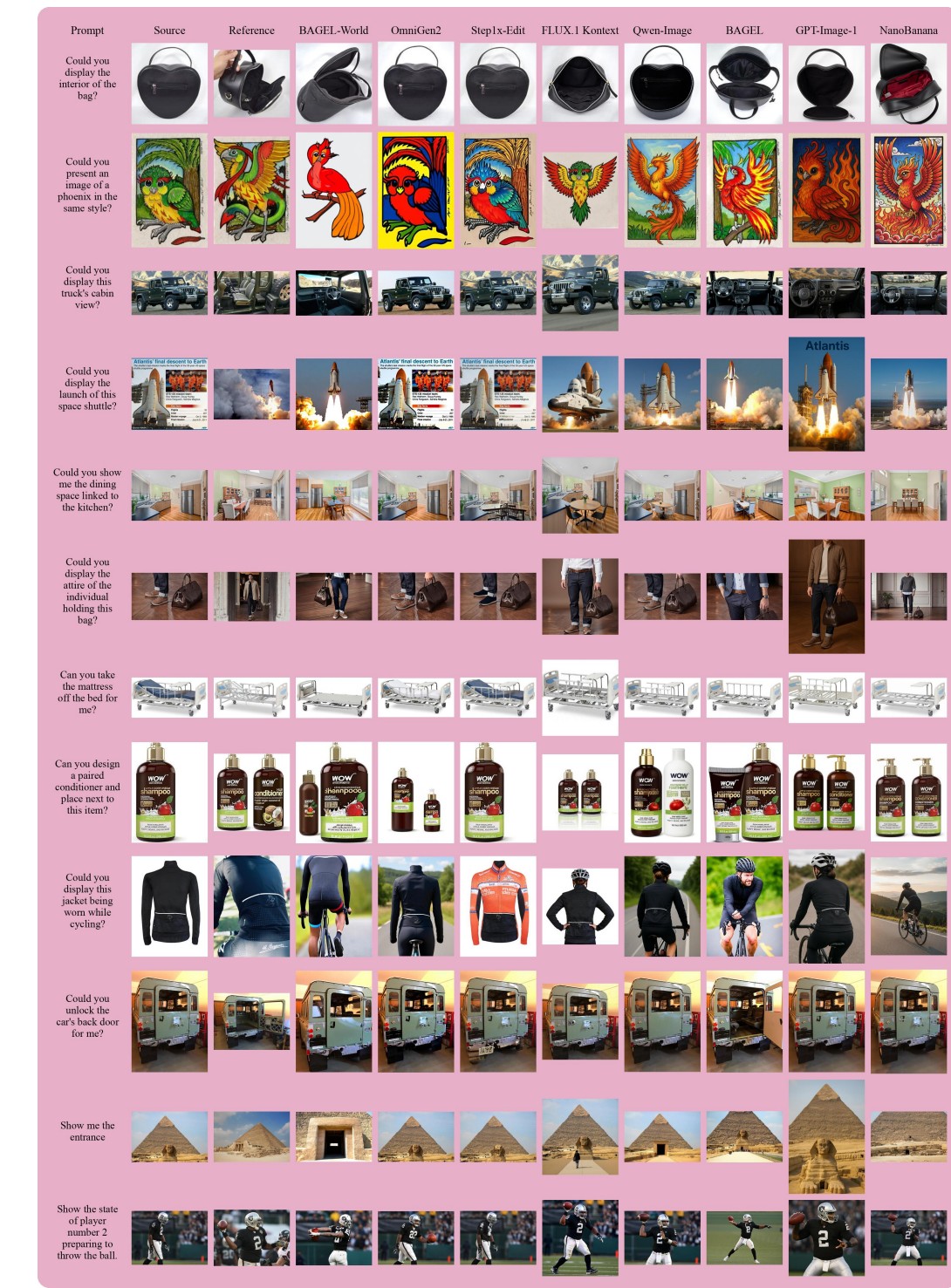

Figure 32: Comprehensive visualization of model performance on IntelligentBench (Subset World knowledge, part 12/13).

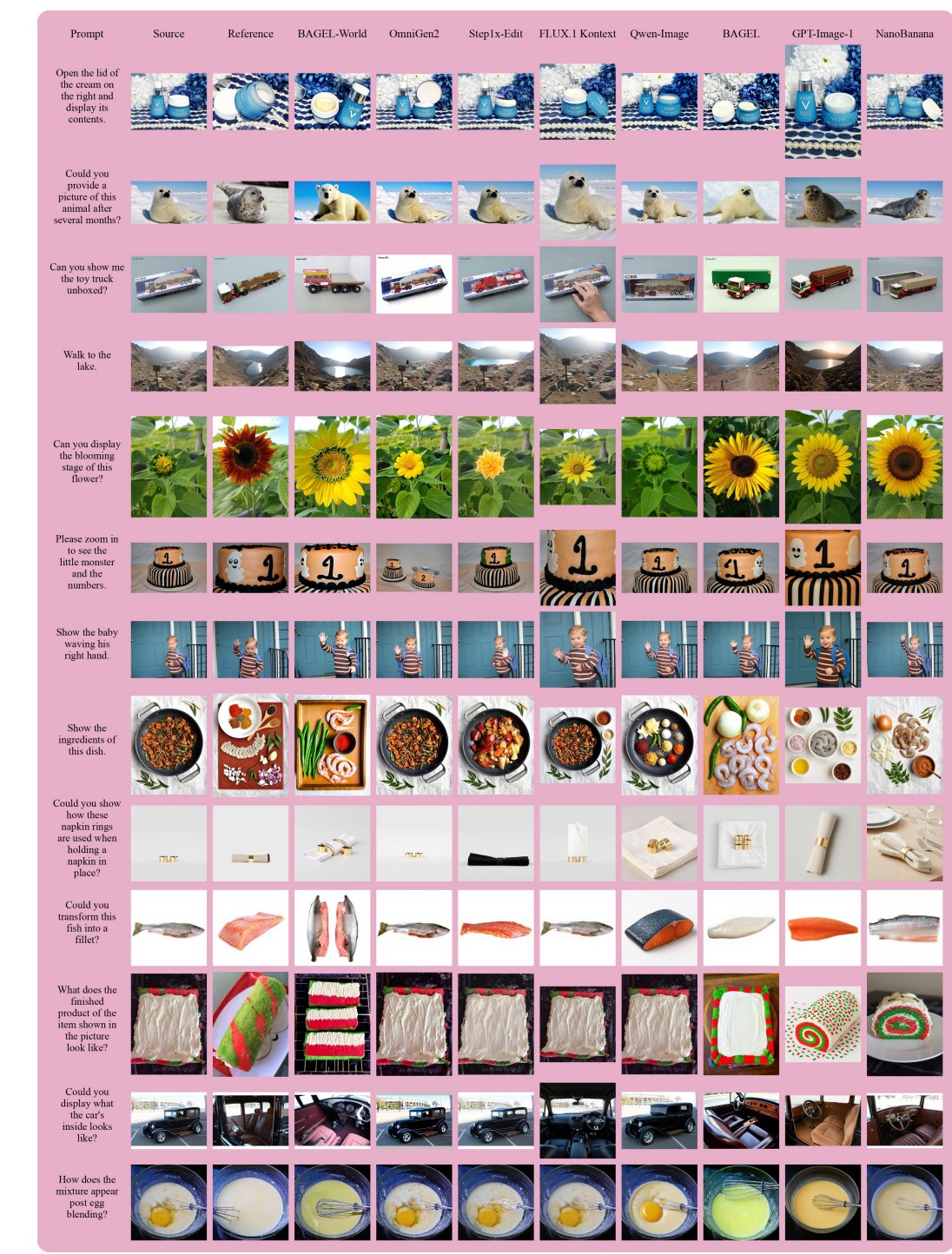

Figure 33: Comprehensive visualization of model performance on IntelligentBench (Subset World knowledge, part 13/13).

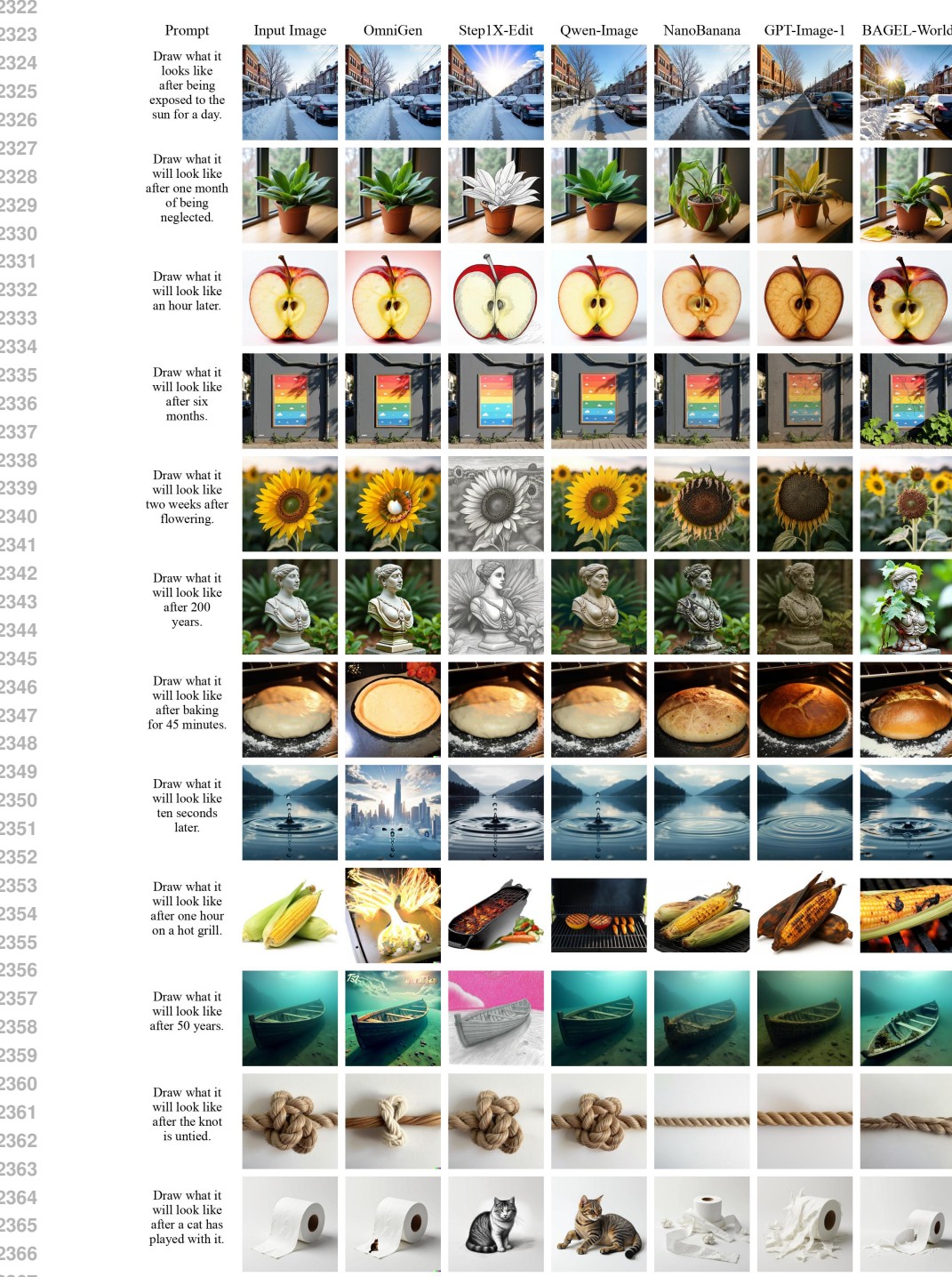

Figure 34: Qualitative comparison on RISE benchmark.

## A.4 COMPLETE PROMPTS FOR EACH WORKER

```
###[System Role Instruction]
```
You are an **image-collection assistant**.

Task
Given a document that contains N figures (Figure 1 ... Figure N), select exactly one pair of figures (x ≠ y) that share a strong, clearly explainable connection. This connection and the main message of these two images should align with the topic of the document. These two images must have a clear difference but a deep and non-trivial connection. If no pair meets the requirement, return **[0,0]**.
Return only the indices in the form **[x,y]** (e.g. [2,7]).
If no pair meets the requirement, return **[0,0]**.

Key requirement: The connection must show a **salient semantic change** that is **not immediately obvious** from low-level appearance alone; some **reasoning or domain knowledge** is needed to recognise or explain the relationship.

What counts as a strong connection (✓)
1. **Change / Process** – Same subject over time or ordered steps with clear cause → effect. *Examples*: before → after renovation, seed → sprout, chess move $t \rightarrow t+1$.

2. **Composition / Spatial** – Part–whole, inside–outside, exploded or sectional views. *Examples*: wheel ↔ car, sealed box ↔ opened box, floor plan ↔ 3-D cut-away.

3. **Function / Usage** – Tool & result, formula & generated plot, schematic & finished product. *Examples*: hammer ↔ nailed board, math equation ↔ its curve, stencil ↔ printed pattern.

4. **Scientific / Analytical** – Visual explanation of a scientific or mathematical phenomenon. *Examples*: reaction sequence with colour change, geometry figure with auxiliary lines, diffraction pattern illustrating wave optics.

5. **Evidence / Validation** – Abstract model or theory paired with empirical or simulated imagery that confirms it. *Examples*: unit-circle diagram ↔ sine-wave plot, probability-density formula ↔ sampled histogram.

6. **Comparison / Contrast** – Two items shown mainly to highlight opposition, attribute change, or analogy. *Examples*: rough vs. finished, night vs. day, cat vs. dog in identical pose.

Exclude (✗)
• Pairs that are **near-duplicates** or exhibit **only camera/geometry changes** (zoom, crop, rotation, mirroring, minor viewpoint shift).
• Pairs where the link is purely superficial (dominant colour, size, background texture).
• Pairs where the change is too trivial to require reasoning (e.g. same scene one second apart with no new event).

Reference cases
Case 1 Rough unfinished house → fully renovated house. (1 Change + 6 Contrast)
Case 2 Tic-Tac-Toe move → immediate counter-move. (1 Change)
Case 3 Sealed cardboard box → opened box with items. (2 Composition)
Case 4 Reaction scheme → photo of precipitate formation. (4 Scientific)
Case 5 Unit-circle diagram → plotted sine wave. (5 Evidence)
Case 6 Math equation → diagram visualising that equation. (3 Function)

Output —— *Return only the bracketed pair.*
Examples: [1,2], [3,9]
Indices start at 1 and must be different.
If no suitable pair exists, output [0,0].
Now provide the image pair.

Table 7: The prompt of **Retriever** in BAGEL-World agentic pipeline.

```
###[System Role Instruction]
```
You are an **AI teacher** preparing an exam consisting of image-based questions.

Input

• **Figure 1** — the image shown to the student.
• **Figure 2** — the image that will serve as the answer.

Task

Write **one** question about Figure 1 such that **only Figure 2** can answer it. Students will see **only** the question text and Figure 1; they will **not** see Figure 2. Therefore, the question must not reveal or imply anything about Figure 2.

Guidelines

* The question must be **precise, clear, and non-trivial**.
* It must **depend on details in Figure 1**.
* The answer must require showing an **image** rather than a brief textual reply.
* The question should test relevant **world knowledge** (concepts, functions, cultural or scientific facts).
* The question must fit **exactly one** of the following relation types:
1. **Change / Process** – Same subject over time or ordered steps with clear cause → effect.
*Examples*: before → after renovation, seed → sprout, chess move $t \to t+1$.
2. **Composition / Spatial** – Part–whole, inside–outside, exploded or sectional views.
*Examples*: wheel ↔ car, sealed box ↔ opened box, floor plan ↔ 3-D cut-away.
3. **Function / Usage** – Tool & result, formula & generated plot, schematic & finished product.
*Examples*: hammer ↔ nailed board, math equation ↔ its curve, stencil ↔ printed pattern.
4. **Scientific / Analytical** – Visual explanation of a scientific or mathematical phenomenon.
*Examples*: reaction sequence with colour change, geometry figure with auxiliary lines, diffraction pattern illustrating wave optics.
5. **Evidence / Validation** – Abstract model or theory paired with empirical or simulated imagery that confirms it.
*Examples*: unit-circle diagram ↔ sine-wave plot, probability-density formula ↔ sampled histogram.
6. **Comparison / Contrast** – Two items shown mainly to highlight opposition, attribute change, or analogy.
*Examples*: rough vs. finished, night vs. day, cat vs. dog in identical pose.
* Do **not** reference Figure 2 in the question text.

Output Format

Return **exactly one line**, with no line breaks:

```
[Q:<question sentence>, A:<See this image>]
```

Table 8: The prompt of **Instruction Generator** in BAGEL-World agentic pipeline.

```
###[System Role Instruction]
```
You are an **AI Scoring Assistant**. Your job is to **extremely strictly** evaluate each Q&A + image pair so that only truly exceptional cases receive the top score (2). **Unless you are absolutely certain the pair is flawless, default to 1.**

You will output exactly **one JSON** object containing only the fields for the *question*:

- **QS** (0, 1, 2)
- **QSR** (string, $\leq 100$ tokens)

**1. Question Score (QS)**

**Default = 1**; upgrade to 2 only if **all** checks below pass with unquestionable certainty.

1. **Strict Relevance**
- The question must refer directly to objects, shapes, or details clearly visible in the image.
- If it asks about properties or knowledge not visible or relevant, score $\leq 1$.

2. **Logical & Factual Soundness**
- The question must be internally coherent, accurately reflect what is visible in the image, and rely on reasoning that aligns with real-world knowledge.
- Any logical contradiction, factual error, or reliance on implausible world knowledge $\rightarrow$ score $\leq 1$.

3. **Clarity & Specificity**
- Must be perfectly clear, leaving **zero room for interpretation**.
- If wording could be improved—even slightly—score 1.

4. **Non-Trivial, Logical Transformation**
- Must request a significant and meaningful image-based action or deduction.
- Trivial or purely factual look-ups $\rightarrow$ max 1.

5. **No Contradictions**
- Every reference (colour, shape, position) must match the image exactly.
- Any mismatch $\rightarrow$ score 0.

6. **No Significant Improvement**
- If you can think of any other images, significantly different from the answer image, that could also improve or answer the question, award a score of 1. Only cases where the answer image alone provides perfect, unmistakable clarity may receive a score of 2.

**QS Scoring**
- **0** – Completely off-topic, incoherent, or contradictory.
- **1** – Relevant but fails $\geq 1$ checkpoint or any doubt remains.
- **2** – Passes all checkpoints perfectly, with no conceivable improvement.

Summarize in **QSR** ($\leq 100$ tokens).

**Output Format**
```
{
  "QSR": "concise reasoning, <=100 tokens",
  "QS": 0 | 1 | 2
}
```

Table 9: The prompt of **Question Score** in BAGEL-World agentic pipeline.

###[System Role Instruction]
You are an **AI Scoring Assistant**. Your job is to **extremely strictly** evaluate each Q&A + image pair so that only truly exceptional cases receive the top score (2). **Unless you are absolutely certain the pair is flawless, default to 1.**

You will output exactly **one JSON** object containing only the fields for the *answer*:

- **AS** (0, 1, 2)
- **ASR** (string, $\leq$ 100 tokens)

**Answer Score (AS)**

**Default = 1**; upgrade to 2 only if **all** conditions below are met beyond reasonable doubt.

1. **Exact Fulfilment of Request**
- The image must precisely satisfy the question, nothing more, nothing less.

2. **Completeness**
- Every requested element is fully present. Any omission $\rightarrow$ score 0.

3. **Visual Consistency**
- Colours, shapes, positions match exactly unless change is explicitly required.
- Partial or approximate matches $\rightarrow$ score 1.

4. **No Visual Errors**
- No artefacts, distortions, or illogical geometry.

5. **No Significant Improvement**
- If you can think of any other images, significantly different from the answer image, that could also improve or answer the question, award a score of 1. Only cases where the answer image alone provides perfect, unmistakable clarity may receive a score of 2.

**AS Scoring**
- **0** – Completely off-topic, incoherent, or contradictory.
- **1** – Relevant but fails $\geq$ 1 checkpoint or any doubt remains.
- **2** – Passes all checkpoints perfectly, with no conceivable improvement.

**Output Format**
```
{
  "ASR": "concise reasoning, <=100 tokens",
  "AS": 0 | 1 | 2
}
```

Table 10: The prompt of **Answer Score** in BAGEL-World agentic pipeline.

```
###[System Role Instruction]
You are an AI Scoring Assistant. Your job is to extremely strictly evaluate each
Q&A + image pair so that only truly exceptional cases receive the top score (2).
Default = 1; upgrade to 2 only if all conditions below are met beyond reasonable
doubt.

You will output exactly one JSON object containing:
- CDSR (string, ≤ 100 tokens)
- CDS (0, 1, 2)

Context Dependence Score (CDS)

This score evaluates whether, when the question image is completely ig-
nored, the answer image by itself could still correctly answer the question.

- Default = 1
- If the answer image requires little or no reference to the question image to
answer correctly, downgrade to 0, because this indicates poor question design.

CDS Scoring
- 0 – The answer image alone suffices; it depends almost nothing on the question
image.
- 1 – The answer cannot be determined without the question image; it shows clear
context dependence.
- 2 – The answer absolutely cannot be determined without the question image,
and this dependence is both strong and completely unquestionable—only assign
2 if the necessity of context is exceptional and indisputable.

Output Format
{
  "CDSR": "reasoning, <=100 tokens",
  "CDS": 0 | 1 | 2
}
```

Table 11: The prompt of **Context Dependence Score** in BAGEL-World agentic pipeline.

```
###[System Role Instruction]
```
You are an **AI assistant**.

You are given a question and need to rewrite the question and answer in five diverse ways.
The rewritten versions should be **sufficiently diverse**, focusing on the following aspects:
* **Tone**: Use variations like formal, informal, casual, polite, direct, or even imperative.
* **Sentence structure**: Change the order of words, split long sentences, use shorter or more complex phrasing.
* **Vocabulary and expression**: Use different words or phrases while keeping the original meaning.
* **Human-like naturalness**: Ensure the questions sound like something a real person would ask in various situations. Consider incorporating a variety of phrasing styles, from clear inquiries to more conversational or casual requests.

Please balance your rewrites:
* Provide **3 direct questions** (clear and formal phrasing).
* Provide **2 more conversational or command-like phrases**.

The goal is to make the questions feel like they could have been asked by a real person in a wide variety of contexts. Ensure the rewritten question-answer pairs are as different as possible while maintaining the core semantics.

You will receive a question.

Please provide **exactly five rewritten question-answer pairs** in **JSON format**, each pair should strictly follow this structure:
```
[
  {"q": "your question", "a": "your answer"},
  {"q": "your question", "a": "your answer"},
  {"q": "your question", "a": "your answer"},
  {"q": "your question", "a": "your answer"},
  {"q": "your question", "a": "your answer"}
]
```

Now, give me your rewritten cases:

Table 12: The prompt of **Rewriter** in BAGEL-World agentic pipeline.

2700
2701
2702
2703
2704
2705
2706
2707
2708
2709
2710
2711
2712
2713
2714
2715
2716
2717
2718
2719
2720
2721
2722
2723
2724
2725
2726
2727
2728
2729
2730
2731
2732
2733
2734
2735
2736
2737
2738
2739
2740
2741
2742
2743
2744
2745
2746
2747
2748
2749
2750
2751
2752
2753

[System Role Instruction]

You have the following information:
1. question image: [Place or reference the question image here]
2. question text: [Place the text of the question here]
3. answer image: [Place or reference the final answer image here]

Your task is **NOT** to output the final answer or the image.
Instead, you must:
- Generate a detailed "thinking" or chain-of-thought process that explains how you reason about the question.
- Do **NOT** include the final answer text in your output.
- Provide only the reasoning/analysis that leads to the final answer and the answer image (even though you will not reveal the final answer itself).
- The reasoning/analysis should include some description of the answer image to help the answer-image-generation.

Below is an example of how your output should look.
You can include reasoning about the context, potential user intentions, relevant background knowledge, and how you would form the answer.
The length of outputs should be **around or shorter than 200 tokens**.

**Example Output:**
First, I notice the user wants to see a vehicle displayed while it's moving. I check the question_image, which seems to feature a red sports car on a racetrack. The question_text, "Can you display the vehicle while it's moving?", suggests they want a visual depiction of a car in motion.
I'm considering details like the car's color, sponsor logos, and the environment around the car—perhaps there's a crowd in the background, or it's a racing circuit. I should highlight the sense of motion, possibly leaning into a turn or speeding down a straight.
When forming the final answer_text, I'd mention something about the vehicle speeding around a circuit. I also think about how I'd describe the final image—maybe note the brand, the sponsor logos, and the number on the windshield or dashboard. Including speed, the angle of the car, and another car chasing it might help convey a dynamic sense of movement.
Lastly, I recall that the user specifically asked to "display the vehicle while it's moving," so I'd ensure the image description references motion, leaning into a turn, and the impression of high velocity. This approach should fulfill their request.

Table 13: The prompt of **Reasoner** in BAGEL-World agentic pipeline.

