# LightBagel 🥯: A Light-weighted, Double Fusion Framework for Unified Multimodal Understanding and Generation

## Abstract

Unified multimodal models have recently shown remarkable gains in both capability and versatility, yet most leading systems are still trained from scratch and require substantial computational resources. In this paper, we show that competitive performance can be obtained far more efficiently by *strategically* fusing publicly available models. Specifically, our key design is to retain the original VLM and DiT blocks while additionally interleaving multimodal self-attention blocks throughout the network. This *double fusion mechanism* (1) effectively enables rich multi-modal fusion while largely preserving the original strengths of the base models, and (2) catalyzes synergistic fusion of high-level semantic representations from a ViT encoder with low-level spatial signals from a VAE encoder. By training with only ~35B tokens, this approach achieves strong results across multiple benchmarks: 0.89 on GenEval for compositional text-to-image generation, 82.28 on DPG-Bench for complex text-to-image generation, 6.06 on GEditBench, and 3.65 on ImgEdit-Bench for image editing. We will fully release the entire suite of code, model weights, and datasets to support future research on unified multimodal modeling.

## 1 Introduction

In recent years, artificial intelligence has witnessed a shift from specialized models to a new generation of unified multimodal systems. The central innovation of these models lies in their ability to natively understand both language and vision while generating linguistic and visual outputs within a single modeling stack. Production-scale systems such as GPT-4o (Hurst et al., 2024) and Gemini 2.0 Flash (Google, 2025) have further demonstrated the promise of this paradigm, with remarkable prompt adherence and dialogue-based image generation and manipulation capabilities, highlighting the potential of "any-to-any" paradigm.

Within the research community, there has also been growing efforts trying to push the boundaries of Unified Multimodal Models (UMMs). The early works (Team, 2024; Wang et al., 2024; Zhou et al., 2024; Wu et al., 2025a; Chen et al., 2025c) adopt a single transformer over mixed-modality sequences. However, such approaches face fundamental optimization challenges due to competing autoregressive and diffusion learning objectives. Later works (Deng et al., 2025; Wu et al., 2025b; Wang et al., 2025b; Wei et al., 2025) introduce decoupled understanding and generation pathways to process tokens from different modalities, easing optimization and improving performance. However, these methods require large-scale training schedules and substantial computational resources, limiting accessibility to the wider community. Other attempts to build UMMs efficiently—such as employing a lightweight connector to map the last layer output of a pre-trained Vision-Language Model (VLM) to a Diffusion Transformer (DiT) for conditional generation (Chen et al., 2025a; Lin et al., 2025)—achieve only partial success, often constrained by limited performances or task coverage, leaving room for further exploration.

In this work, we propose a lightweight yet powerful UMM training framework by strategically fusing VLM and DiT models. Specifically, we keep the original VLM and DiT blocks as they are, and insert zero-initialized multimodal self-attention block after each understanding and generation block. This design keeps the strong autoregressive and diffusion capabilities of base VLM and DiT models,

Figure 1: Token efficiency comparison on T2I and image editing benchmarks. Our LightBagel outperforms many leading unified models that uses significantly more tokens for training, showing great token efficiency. Note that we use our best estimate for the number of seen tokens of OmniGen2 and UniPic since their original training recipe is unclear to the public.

while enabling early and continuous cross-modal interaction throughout the network. Controlled experiments demonstrate clear advantages over the widely adopted "Shallow Fusion" approach (Tang et al., 2025), which relies only on the final-layer output of the understanding branch as a conditioning signal (Chen et al., 2025a; Lin et al., 2025). Furthermore, the dual-pathway design allows a clean separation of ViT tokens (processed in the understanding pathway) and VAE tokens (processed in the generation pathway), effectively integrating high-level semantic representations with low-level signals. We refer to this approach as *Double Fusion*, as it simultaneously fuses understanding and generation branches, as well as ViT- and VAE-derived features.

To support this architecture, we curate a UMM tuning dataset emphasizing data quality, task balance, and diversity, by combining post-processed public text-to-image and image-editing data with carefully generated synthetic data. Together, these components enables efficient training, achieving strong performance with substantially fewer tokens and compute compared to prior models. We refer to the resulting model as *LightBagel*.

LightBagel delivers state-of-the-art results across multiple benchmarks, including a GenEval score of 0.89 for compositional text-to-image generation, an 82.28 DPG-Bench score for complex text-to-image generation, and scores of 6.06 on GEditBench and 3.65 on ImgEdit-Bench for image editing. Remarkably, trained on only ~35B seen tokens, LightBagel achieves performance on par with or surpassing leading models like UniPiC and OmniGen2, which were trained with orders of magnitudes of more tokens. These results highlight LightBagel's efficiency and suggest new directions for the design of future UMM architectures. We will release all code, models, and datasets to promote reproducibility and further progress in the community.

Our key contributions are summarized as follows:

- We introduce a novel dual-branch architecture that fuses pretrained understanding and generation models in a layer-by-layer fashion, while natively supporting the integration of ViT-based high-level semantics and VAE-based low-level representations.

- We present LightBagel, a unified multimodal model that achieves state-of-the-art performance across text-to-image and image-editing benchmarks, with substantially greater token and compute efficiency compared to existing leading UMMs.

- We will release our code, model weights, and training dataset, with hope to facilitate reproducibility and inspire future work in UMM research.

## 2 RELATED WORKS

**Text-to-Image Generation.** Diffusion models (Ho et al., 2020; Song et al., 2020) have emerged as the dominant paradigm for open-domain image synthesis, surpassing GANs (Goodfellow et al., 2020) with improved stability and semantic fidelity. Seminal works, such as Stable Diffusion and its successors (Rombach et al., 2022; Podell et al., 2023; Esser et al., 2024), DALL·E (Ramesh et al., 2022), and Imagen (Imagen 3 Team, 2024), demonstrated the power of large-scale pretraining and latent denoising for high-resolution, text-aligned generation. Follow-up studies advanced diffusion-based pipelines by improving model architecture (Peebles & Xie, 2023; Yu et al., 2025), emphasizing data quality (Chen et al., 2024b;a), and enhancing controllability through spatial or semantic constraints (Zhang et al., 2023b; Li et al., 2023; Mou et al., 2024). More recently, flow-matching models (Lipman et al., 2022) have been proposed as a powerful alternative to diffusion, modeling vector fields between noise and data distributions, and have achieved strong results in large-scale systems (Labs, 2024; Wan et al., 2025).

**Image Editing.** Image editing has been widely studied as an extension of text-to-image generation. InstructPix2Pix (Brooks et al., 2023) pioneered the supervised paradigm based on *{instruction, source image, target image}* triplets. Subsequent works improved this approach by diversifying training data and improving quality (Zhang et al., 2023a; Hui et al., 2025; Wei et al., 2024; Ye et al., 2025), or by incorporating stronger control signals (Tan et al., 2023; Liu et al., 2025). Parallel directions include mask-based editing (Ju et al., 2024; Zhuang et al., 2024) and subject-driven generation through lightweight fine-tuning methods (Ruiz et al., 2023; Raj et al., 2023; Ruiz et al., 2024). More recent frameworks such as OmniGen (Xiao et al., 2025) and UniReal (Chen et al., 2025b) unify common image generation tasks (*e.g.* T2I generation, image editing, personalization) by processing various conditional inputs with modified attention mechanisms, demonstrating the potential of a single model for multiple generation tasks.

**Unified Multimodal Models.** Unified Multimodal Models (UMMs) have rapidly gained attention for their ability to support both understanding and generation, enabling smooth cross-modality knowledge transfer. Early works such as Chameleon (Team, 2024), EMU3 (Wang et al., 2024), Transfusion (Zhou et al., 2024), Janus (Wu et al., 2025a; Chen et al., 2025c), and Show-o (Xie et al., 2024) employed a single Transformer backbone to process interleaved image and text tokens. Among the, hybrid approaches, including Transfusion (Zhou et al., 2024) and Show-o (Xie et al., 2024), combine autoregressive text decoding with diffusion-based image generation, while others rely on discrete image tokenizers to autoregressively predict both image and text tokens.

Recent work has explored how to adapt powerful pretrained models into UMMs. LMFusion (Shi et al., 2024) extends text-only Large Language Models (LLMs) with a parallel image-generation branch, a design that inspires our own approach of fusing well-trained understanding and generation models as parallel pathways via additional attention layers. Other approaches such as MetaQueries (Pan et al., 2025) and UniWorld (Lin et al., 2025) propose efficient schemes to combine frozen multimodal LLMs with trainable diffusion models, enabling knowledge-augmented image generation while maintaining strong understanding ability. Finally, large-scale pretraining efforts (Deng et al., 2025; Liao et al., 2025; Wu et al., 2025b; Wang et al., 2025b; Wei et al., 2025) focus on massive interleaved image-text corpora, demonstrating broad generalization across understanding, generation, editing, and other downstream tasks.

## 3 METHODS

### 3.1 MODEL ARCHITECTURE

As shown in Fig. 2, LightBagel adopts a mixture-of-experts (MoE) style architecture inspired by LMFusion (Shi et al., 2024) and BAGEL (Deng et al., 2025). The design interleaves multimodal self-attention blocks across Vision-Language Model (VLM) and Diffusion Transformer (DiT) blocks. To fully leverage the strength of open-source models with extensive pre-training and post-training, we employ QWen2.5-VL-7B (Bai et al., 2025) for the understanding pathway and Wan2.2-TI2V-5B (Wan et al., 2025) for the generation pathway. As QWen2.5-VL-7B has slightly fewer layers than Wan2.2-TI2V-5B (two layers fewer), its final-layer output is reused as input to the last two multimodal self-attention blocks.

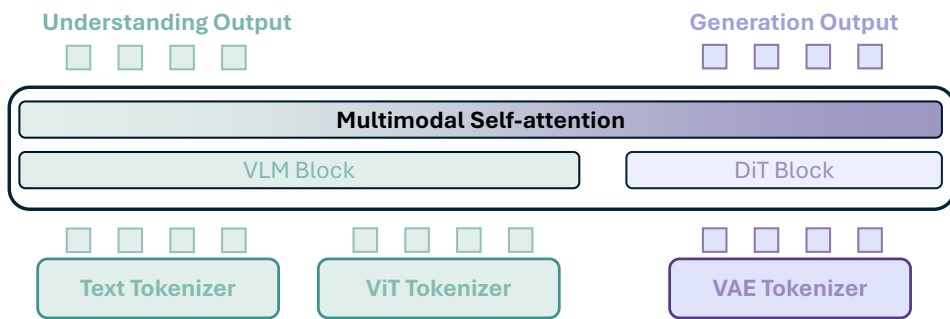

Figure 2: Overview of the LightBagel architecture. Text and ViT tokens (understanding pathway) and VAE tokens (generation pathway) are processed by pre-trained VLM and DiT blocks, respectively. At each layer, a zero-initialized multimodal self-attention module enables cross-modal interactions without altering the original model architectures.

In this architecture, the VLM blocks process understanding tokens (i.e., text and ViT tokens), while the DiT blocks operate on generation tokens (i.e., VAE tokens). The multimodal self-attention blocks span all token types, enabling rich cross-modal interactions. To facilitate this, we adopt the generalized causal attention mechanism from BAGEL (Deng et al., 2025), allowing tokens from different modalities and tokenizers to attend to one another. Importantly, each multimodal self-attention block is zero-initialized, ensuring that the feature distributions of the VLM and DiT remain intact at the start of training, thereby preserving their strong autoregressive and denoising capabilities. For image editing tasks, both ViT and VAE tokens are extracted from the source image and provided as conditioning signals. The ViT tokens are extracted using the QWen2.5-VL vision encoders. The VAE tokens are extraced by the 3D causal VAE from Wan2.2-TI2V-5B (Wan et al., 2025), which provides 16× spatial compression and 4× temporal compression.

This design offers several key advantages:

- **Seamless model integration**. The framework incorporates powerful pre-trained VLM and DiT models without altering their architectures, offering a straightforward and generalizable method for fusing publicly available models into a unified multimodal system. In line with prior findings (Tang et al., 2025), we observe that this "deep fusion" strategy consistently outperforms "shallow fusion" approaches while delivering superior token efficiency.

- **Dual-pathway visual representation**. By naturally integrating ViT tokens (high-level semantics) and VAE tokens (low-level signals), the architecture achieves precise and consistent image editing, effectively balancing global understanding with fine-grained detail.

- **Information-preserving multimodal interaction**. Leveraging hidden states from all understanding and generation layers within the multimodal self-attention avoids compressing conditioning inputs into a fixed-length representation (Pan et al., 2025; Chen et al., 2025a; Wei et al., 2025), ensuring rich, loss-free cross-modal interactions.

## 3.2 DATASET

Our training corpus comprises approximately 45 million samples, encompassing both text-to-image generation and image-editing tasks. Data are sourced from a variety of publicly available datasets, including BLIP3o (Chen et al., 2025a), Civitai (civ, 2024), OmniGen (Xiao et al., 2025), OmniEdit (Wei et al., 2024), GPT-IMAGE-EDIT-1.5M (Wang et al., 2025c), and UniWorld-V1 (Lin et al., 2025), supplemented by a synthetic self-curated dataset of ~4.5 million samples. Notably, we apply a vision–language model to refine the editing instructions of public image editing data conditioned on source–target pairs, thereby enhancing instruction precision.

## 3.3 TRAINING

We adopt NaViT-style image processing to preserve native aspect ratios (Dehghani et al., 2023), constraining inputs to a minimum short side of 512 pixels and a maximum long side of 1024 pixels, thereby improving generation quality. ViT tokens are extracted with input sizes ranging from a

Table 1: Comparison of different models across understanding, generation, editing, and in-context Generation tasks. † refers to the methods using LLM rewriter.

| Model | # Params | # Unified Model | # Open-source | Understanding | | | Image Generation | | Image Editing | |
|---|---|---|---|---|---|---|---|---|---|---|
| | | | | MMB | MMMU | MM-Vet | GenEval | DPG-Bench | ImgEdit-Bench | GEdit-Bench-EN |
| LLaVA-1.5 | – | – | – | 36.4 | 67.8 | 36.3 | – | – | – | – |
| LLaVA-NeXT | – | – | – | 79.3 | 51.1 | 57.4 | – | – | – | – |
| SDXL | – | – | – | – | – | – | 0.55 | 74.7 | – | – |
| SD3-medium | – | – | – | – | – | – | 0.62 | 84.1 | – | – |
| FLUX.1-dev (Labs, 2024) | – | – | – | – | – | – | 0.66 | 84.0 | – | – |
| Instruct-P2P | – | – | – | – | – | – | – | – | 1.88 | 3.68 |
| MagicBrush | – | – | – | – | – | – | – | – | 1.90 | 1.86 |
| AnyEdit | – | – | – | – | – | – | – | – | 2.45 | 3.21 |
| Step1X-Edit | – | – | – | – | – | – | – | – | 3.06 | 6.70 |
| IC-Edit | – | – | – | – | – | – | – | – | 3.05 | 4.84 |
| Janus-Pro | – | Yes | Model | 75.5 | 36.3 | 39.8 | 0.80 | 84.19 | – | – |
| Emu3 | – | Yes | Model | 58.5 | 31.6 | 37.2 | 0.54 / 0.66† | 80.60 | – | – |
| Show-o2 | 7B | Yes | Model | 79.3 | 48.9 | – | 0.76 | 86.14 | – | – |
| MetaQuery-XL | 7B + 1.6B | Yes | Model | 83.5 | 58.6 | 66.6 | 0.80† | 82.05 | – | – |
| UniPic | 1.5B | Yes | Model | – | – | – | 0.86 | 85.50 | 3.49 | 5.83 |
| UniPic 2.0 | 7B + 2B | Yes | Model | 83.5 | 58.6 | 67.1 | 0.90† | 83.79 | 4.06 | 7.10 |
| Ovis-U1 | 2.4B + 1.2B | Yes | Model | 77.8 | 51.1 | 66.7 | 0.89 | 83.72 | 4.00 | 6.42 |
| OmniGen | 3.8B | Yes | Model | – | – | – | 0.68 | 81.16 | 2.96 | 5.06 |
| OmniGen2 | 3B + 4B | Yes | Model | 79.1 | 53.1 | 61.8 | 0.80 / 0.86† | 83.57 | 3.44 | 6.42 |
| BAGEL | 7B + 7B | Yes | Model | 85.0 | 55.3 | 67.2 | 0.82 / 0.88† | 85.07 | 3.20 | 6.52 |
| BLIP3-o 4B | 3B + 1.4B | Yes | Full | 78.6 | 46.6 | 60.1 | 0.81† | 79.36 | – | – |
| BLIP3-o 8B | 7B + 1.4B | Yes | Full | 83.5 | 58.6 | 66.6 | 0.84† | 81.60 | – | – |
| UniWorld-V1 | 7B + 12B | Yes | Full | 83.5 | 58.6 | 67.1 | 0.84† | 81.38 | 3.26 | 4.85 |
| LightBagel | 7B + 5B + 3B | Yes | Full | 83.5 | 58.6 | 67.1 | 0.89† | 82.28 | 3.65 | 6.06 |

minimum short side of 224 pixels to a maximum long side of 532 pixels. The LightBagel model is trained for 70K steps using the AdamW optimizer, with 2K warmup steps and a fixed learning rate of 0.00003. The sequence length is configured between 16,384 (minimum) and 20,480 (maximum). To enable classifier-free guidance, text tokens, VAE tokens, and ViT tokens are randomly dropped with probabilities of 0.1, 0.1, and 0.5, respectively. The understanding branch is frozen during the entire training time to keep the strong understanding performance of QWen2.5-VL-7B.

## 4 EXPERIMENTS

In this section, we evaluate the performance of our LightBagel model across a diverse set of image understanding, generation, and editing tasks. Sec. 4.1 presents the visual understanding results. Text-to-image generation results are reported in Sec. 4.2, while Sec. 4.3 focuses on image editing. Sec. 4.4 presents our ablation study, analyzing key design choices of the model. Our experiment results show that LightBagel achieves strong performances over a wide spectrum of tasks and benchmarks, demonstrating the effectiveness of our double fusion approach.

### 4.1 VISUAL UNDERSTANDING

By freezing the understanding branch, our model preserves the strong multimodal reasoning capabilities of the pre-trained QWen2.5-VL-7B. As reported in Table 1, LightBagel attains competitive performances of 83.5 on MMBench (Liu et al., 2024), 58.6 on MMMU (Yue et al., 2024), and 67.1 on MM-Vet (Yu et al., 2023). This architectural choice aligns with recent state-of-the-art designs, such as UniWorld-V1(Lin et al., 2025), OmniGen2(Wu et al., 2025b), and UniPiC 2.0(Wang et al., 2025b), which similarly prioritize maintaining robust understanding performance. As a result, LightBagel effectively mitigates potential degradation in understanding capabilities and surpasses several strong competitors, including Ovis-U1(Wang et al., 2025a), Show-o2(Xie et al., 2025), and Janus-Pro(Chen et al., 2025c).

### 4.2 TEXT-TO-IMAGE GENERATION

We employ two widely recognized benchmarks, GenEval and DPG-Bench, to evaluate LightBagel's text-to-image generation performance. These benchmarks primarily assess the model's capabilities in compositional image generation and dense prompt following, respectively. In addition, we provide qualitative visualization examples in Fig. 3. Both the quantitative results and qualitative illustrations demonstrate that LightBagel is capable of producing high-fidelity, aesthetically compelling images.

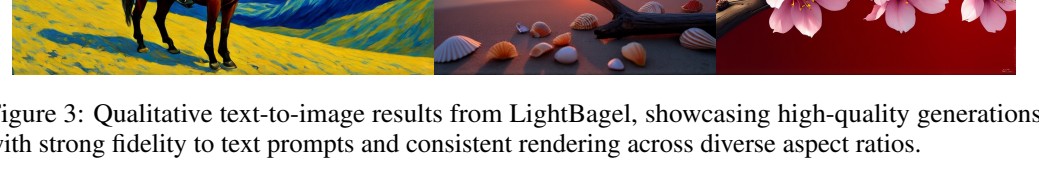

Figure 3: Qualitative text-to-image results from LightBagel, showcasing high-quality generations with strong fidelity to text prompts and consistent rendering across diverse aspect ratios.

**GenEval.** As reported in Table 2, LightBagel attains an overall score of 0.89 on GenEval when evaluated with LLM-rewritten prompts, highlighting its strong compositional understanding across diverse generative tasks. This performance surpasses several competitive baselines, including UniPiC (0.86), OmniGen2 (0.86), and BAGEL (0.88), while approaching the state-of-the-art UniPiC2.0 model (0.90). Notably, LightBagel is trained with over a order of magnitude of less seen tokens, underscoring its remarkable efficiency in terms of both data and compute resources.

**DPG-Bench.** As shown in Table 3, On DPG-Bench, LightBagel achieves an overall score of 82.28, demonstrating competitive performance in long-prompt adherence and complex scene generation. This result surpasses, other strong unified models such as BLIP3-o 8B (81.60) and UniWorld-V1 (81.38). Detailed breakdowns in Table 3 indicate that LightBagel maintains consistently strong performance across multiple dimensions of evaluation, including global coherence, entity recognition, attribute understanding, and relational reasoning.

## 4.3 IMAGE EDITING

We evaluate LightBagel's image editing capabilities using two widely adopted benchmarks: GEdit-Bench-EN(Liu et al., 2025) and ImgEdit-Bench(Ye et al., 2025). GEdit-Bench-EN consists of real-world user editing instances, while ImgEdit-Bench encompasses nine distinct editing tasks (e.g., add, remove, alter). We further provide qualitative examples of LightBagel's editing results in Fig. 4. Both quantitative and qualitative results demonstrate that LightBagel delivers strong performance in instruction-based image editing, excelling in both editing accuracy and content preservation.

**GEdit-Bench-EN.** As shown in Table 4, LightBagel achieves an overall score of 6.06, positioning it among the top-tier unified models. The model demonstrates particular strength in semantic consistency (SC), attaining a score of 6.56, which reflects robust instruction-following capabilities. This result notably surpasses that of UniWorld-V1 (4.85) by a large margin, underscoring the effectiveness of our hybrid ViT+VAE feature fusion strategy compared to only using ViT tokens as condition in image editing task.

Table 2: Evaluation of text-to-image generation ability on GenEval benchmark. † refers to the methods using LLM rewriter.

| Method | Single object↑ | Two object↑ | Counting↑ | Colors↑ | Position↑ | Color attribution↑ | Overall↑ |
|--------|------|------|------|------|------|------|------|
| SDv2.1 | 0.98 | 0.51 | 0.44 | 0.85 | 0.07 | 0.17 | 0.50 |
| SDXL | 0.96 | 0.74 | 0.39 | 0.85 | 0.15 | 0.23 | 0.55 |
| IF-XL | 0.97 | 0.74 | 0.66 | 0.81 | 0.13 | 0.35 | 0.61 |
| LUMINA-Next | 0.92 | 0.46 | 0.48 | 0.70 | 0.09 | 0.13 | 0.46 |
| SD3-medium | 0.99 | 0.94 | 0.72 | 0.89 | 0.33 | 0.60 | 0.74 |
| FLUX.1-dev | 0.99 | 0.81 | 0.79 | 0.74 | 0.20 | 0.47 | 0.67 |
| NOVA | 0.99 | 0.91 | 0.62 | 0.85 | 0.33 | 0.56 | 0.71 |
| OmniGen | 0.98 | 0.84 | 0.66 | 0.74 | 0.40 | 0.43 | 0.68 |
| TokenFlow-XL | 0.95 | 0.60 | 0.41 | 0.81 | 0.16 | 0.24 | 0.55 |
| Janus | 0.97 | 0.68 | 0.30 | 0.84 | 0.46 | 0.42 | 0.61 |
| Janus Pro† | 0.99 | 0.89 | 0.59 | 0.90 | 0.79 | 0.66 | 0.80 |
| Emu3-Gen† | 0.99 | 0.81 | 0.42 | 0.80 | 0.49 | 0.45 | 0.66 |
| Show-o2† | 1.00 | 0.87 | 0.58 | 0.92 | 0.52 | 0.62 | 0.76 |
| MetaQuery-XL† | – | – | – | – | – | – | 0.80 |
| UniPic | 0.98 | 0.92 | 0.74 | 0.91 | 0.89 | 0.72 | 0.86 |
| UniPic 2.0† | – | – | – | – | – | – | 0.90 |
| Ovis-U1† | 0.98 | 0.98 | 0.90 | 0.92 | 0.79 | 0.75 | 0.89 |
| BAGEL† | 0.98 | 0.95 | 0.84 | 0.95 | 0.78 | 0.77 | 0.88 |
| OmniGen2† | 0.99 | 0.96 | 0.74 | 0.98 | 0.71 | 0.75 | 0.86 |
| BLIP3-o† 8B | – | – | – | – | – | – | 0.84 |
| UniWorld-V1† | 0.98 | 0.93 | 0.81 | 0.89 | 0.74 | 0.71 | 0.84 |
| LightBagel† | 1.00 | 0.98 | 0.91 | 0.94 | 0.82 | 0.76 | 0.89 |

Table 3: Evaluation of text-to-image generation ability on DPG-Bench benchmark.

| Method | Global↑ | Entity↑ | Attribute↑ | Relation↑ | Other↑ | Overall↑ |
|--------|------|------|------|------|------|------|
| LUMINA-Next | 82.82 | 88.65 | 86.44 | 80.53 | 81.82 | 74.63 |
| SDXL | 83.27 | 82.43 | 80.91 | 86.76 | 80.41 | 74.65 |
| PlayGroundv2.5 | 83.06 | 82.59 | 81.20 | 84.08 | 83.50 | 75.47 |
| Hunyuan-DiT | 84.59 | 80.59 | 88.01 | 74.36 | 86.41 | 78.87 |
| PixArt-Σ | 86.89 | 82.89 | 88.94 | 86.59 | 87.68 | 80.54 |
| DALLE3 | 90.97 | 89.61 | 88.39 | 90.58 | 89.83 | 83.50 |
| SD3-medium | 87.90 | 91.01 | 88.83 | 80.70 | 88.68 | 84.08 |
| FLUX.1-dev | 82.10 | 89.50 | 88.70 | 91.10 | 89.40 | 84.00 |
| OmniGen | 87.90 | 88.97 | 88.47 | 87.95 | 83.56 | 81.16 |
| TokenFlow-XL | 78.72 | 79.22 | 81.29 | 85.22 | 71.20 | 73.38 |
| Janus | 82.33 | 87.38 | 87.70 | 85.46 | 86.41 | 79.68 |
| Janus Pro | 86.90 | 88.90 | 89.40 | 89.32 | 89.02 | 84.19 |
| Show-o2 | 89.00 | 91.78 | 89.96 | 91.81 | 91.64 | 86.14 |
| EMU3 | 85.21 | 86.68 | 86.84 | 90.22 | 83.15 | 80.60 |
| UniPic | 89.65 | 87.78 | 90.84 | 91.89 | 91.95 | 85.50 |
| UniPic 2.0 | - | - | - | - | - | 83.79 |
| Ovis-U1 | 82.37 | 90.08 | 88.68 | 93.35 | 85.20 | 83.72 |
| BAGEL | 88.94 | 90.37 | 91.29 | 90.82 | 88.67 | 85.07 |
| OmniGen2 | 88.81 | 88.83 | 90.18 | 89.37 | 90.27 | 83.57 |
| BLIP3-o 8B | – | – | – | – | – | 81.60 |
| UniWorld-V1 | 83.64 | 88.39 | 88.44 | 89.27 | 87.22 | 81.38 |
| LightBagel | 89.58 | 88.10 | 89.14 | 88.83 | 90.46 | 82.28 |

**ImgEdit-Bench.** As can be seen in Table 5), LightBagel achieves an overall score of 3.65, outperforming other strong open-source competitors such as UniWorld-V1 (3.26) and OmniGen2 (3.44). Importantly, LightBagel ranks first among open-source models in several key categories, including Add (4.21), Background (4.32), and Hybrid (3.47), which highlights the model's robust and consistent editing ability across a broad spectrum of tasks.

## 4.4 ABLATION STUDY

For all ablation study experiments, we keep the general training setup while oping for 40K trainig steps for faster experiment cycles.

**Deep Fusion vs Shallow Fusion.** Previous efficient UMM approaches typically employ a lightweight connector that maps the final output of the understanding branch as a conditional input to the

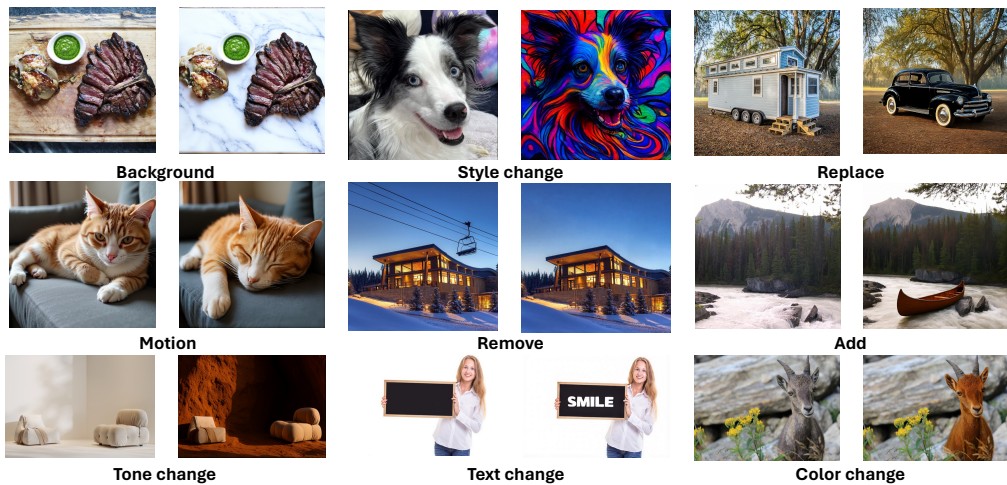

Figure 4: Qualitative image editing results generated by LightBagel. The model produces high-quality outputs across a diverse range of editing tasks.

Table 4: Evaluation of image editing ability on GEdit-Bench-EN

| Model | SC ↑ | PQ ↑ | Overall ↑ |
|---|---|---|---|
| Gemini-2.0-flash | 6.73 | 6.61 | 6.32 |
| GPT-4o | 7.85 | 7.62 | 7.53 |
| Instruct-Pix2Pix | 3.58 | 5.49 | 3.68 |
| MagicBrush | 4.68 | 5.66 | 4.52 |
| AnyEdit | 3.18 | 5.82 | 3.21 |
| ICEdit | 5.11 | 6.85 | 4.84 |
| Step1X-Edit | 7.09 | 6.76 | 6.70 |
| OmniGen2 | 7.16 | 6.77 | 6.41 |
| BAGEL | 7.36 | 6.83 | 6.52 |
| Ovis-U1 | – | – | 6.42 |
| UniPic | 6.72 | 6.18 | 5.83 |
| UniPic 2.0 | – | – | 7.10 |
| UniWorld-V1 | 4.93 | 7.43 | 4.85 |
| LightBagel | 6.56 | 7.06 | 6.06 |

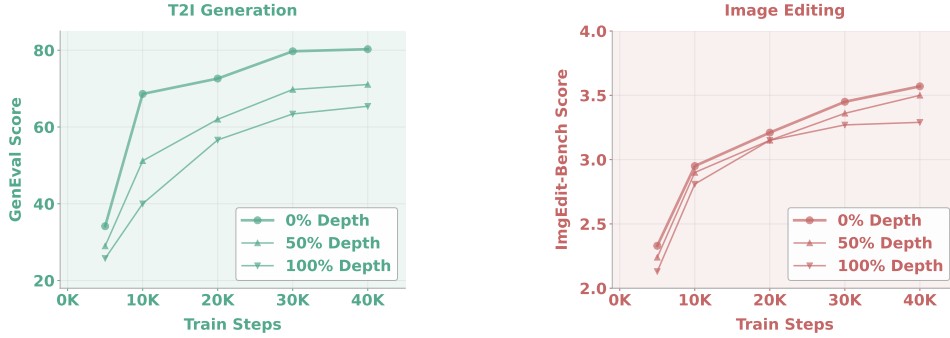

Figure 5: Deep fusion *vs*. shallow fusion design choices.

generation branch. In contrast, our Double Fusion design allows language and visual tokens to interact from the earliest layers, enabling deeper and more continuous cross-modal integration.

To systematically compare the two strategies, we vary the depth at which features from the VLM are injected into the generation pathway. Specifically, "0% Depth" denotes the case where the $i$-th DiT block is conditioned on the $i$-th VLM block's output. When the VLM blocks are exhausted,

Table 5: Evaluation of image editing ability on ImgEdit-Bench.

| Model | Add | Adjust | Extract | Replace | Remove | Background | Style | Hybrid | Action | Overall |
|---|---|---|---|---|---|---|---|---|---|---|
| GPT-4o | 4.61 | 4.33 | 2.90 | 4.35 | 3.66 | 4.57 | 4.93 | 3.96 | 4.89 | 4.20 |
| MagicBrush | 2.84 | 1.58 | 1.51 | 1.97 | 1.58 | 1.75 | 2.38 | 1.62 | 1.22 | 1.90 |
| Instruct-Pix2Pix | 2.45 | 1.83 | 1.41 | 2.01 | 1.44 | 1.44 | 3.55 | 1.20 | 1.46 | 1.88 |
| AnyEdit | 3.18 | 2.95 | 1.14 | 2.49 | 2.21 | 2.88 | 3.82 | 1.56 | 2.65 | 2.45 |
| UltraEdit | 3.44 | 2.81 | 2.00 | 2.96 | 2.45 | 2.83 | 3.76 | 1.91 | 2.98 | 2.70 |
| Step1X-Edit | 3.88 | 3.41 | 1.76 | 3.40 | 2.83 | 3.16 | 6.63 | 2.52 | 2.52 | 3.06 |
| ICEdit | 3.58 | 3.39 | 1.73 | 3.15 | 2.93 | 3.08 | 3.84 | 2.04 | 3.68 | 3.05 |
| OmniGen2 | 3.74 | 3.54 | 1.77 | 3.21 | 2.77 | 3.57 | 4.81 | 2.30 | 4.14 | 3.43 |
| BAGEL | 3.56 | 3.31 | 1.88 | 2.62 | 2.88 | 3.44 | 4.49 | 2.38 | 4.17 | 3.20 |
| Ovis-U1 | 4.12 | 3.92 | 2.36 | 4.09 | 3.57 | 4.22 | 4.69 | 3.23 | 3.61 | 3.98 |
| UniPic | 3.66 | 3.51 | 2.06 | 4.31 | 2.77 | 3.77 | 4.76 | 2.56 | 4.04 | 3.49 |
| UniPic 2.0 | - | - | - | - | - | - | - | - | - | 4.06 |
| UniWorld-V1 | 3.82 | 3.66 | 2.31 | 3.45 | 3.02 | 2.99 | 4.71 | 2.96 | 2.74 | 3.26 |
| LightBagel | 4.21 | 3.39 | 1.58 | 4.09 | 3.39 | 4.37 | 4.38 | 3.47 | 3.99 | 3.65 |

Table 6: Ablation study on visual tokenizer and timestep shift choices.

| Encoder | GEdit-EN | ImgEdit |
|---|---|---|
| ViT | 3.91 | 2.65 |
| VAE | 4.93 | 3.38 |
| ViT + VAE | 5.61 | 3.57 |

(a) Evaluation of different visual tokenizer choices.

| Timestep Shift | DPG-Bench | GEdit-EN | ImgEdit |
|---|---|---|---|
| 1 | 76.67 | 4.60 | 3.07 |
| 2 | 78.84 | 5.37 | 3.36 |
| 4 | 81.77 | 5.605 | 3.57 |

(b) Evaluation of different timestep shift choices.

the final VLM output is repeated as input for the remaining DiT blocks. Conversely, "100% Depth" corresponds to conditioning all DiT blocks exclusively on the final VLM output, effectively repeating it across all multimodal self-attention layers. Importantly, we keep the total number of multimodal self-attention layers fixed, ensuring identical parameter counts and a fair comparison.

As shown in Fig. 5, the "0% Depth" option—adopted in our LightBagel model—consistently outperforms shallow or early-layer fusion for both text-to-image and image-editing tasks. We attribute this advantage to the fact that the final VLM representations encode high-level semantics more suitable for next-token prediction rather than multimodal alignment.

**Visual Tokenizer Choices.** VAE and ViT encoders provide complementary visual representations: VAEs emphasize low-level details, while ViTs capture high-level semantic information. To assess which type of information is more suitable for image editing, we conduct an ablation study on visual tokenizer choice, with results reported in Tab. 6a. The findings indicate that both sources of information are essential—combining low-level details with high-level semantics leads to more consistent and accurate image editing outcomes.

**Training Timestep Shift.** Shifting timesteps during inference has been shown to improve generation quality in state-of-the-art models (Labs, 2024; Wan et al., 2025). In our preliminary experiments, we observe that the strong base DiT pathway already demonstrates robust denoising capabilities. Building on this, and following the inference-time timestep shifting strategy (Labs, 2024; Wan et al., 2025), we increase the training diffusion timestep range from 1.0 to 4.0 to achieve more noisy corrupted samples. As reported in Tab. 6b, larger timestep shifts consistently lead to better results.

## 5 CONCLUSION

In this work, we introduce LightBagel, a unified multimodal understanding and generation model for understanding and generation that achieves state-of-the-art performance in both text-to-image generation and image editing, while requiring substantially fewer training tokens and compute than prior leading UMMs. Our Double Fusion architecture enables effective cross-modal feature interactions and naturally integrates high-level semantics with low-level visual details. We believe LightBagel provides a step toward more efficient and accessible multimodal modeling, and we will release all code, models, and data to support reproducibility and accelerate progress in the community.

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

# A APPENDIX

## A.1 LLM USAGE

During the preparation of this manuscript, we used OpenAI's GPT-5 model for minor language refinement and smoothing of the writing. The AI tool was not used for generating original content, conducting data analysis, or formulating core scientific ideas. All conceptual development, experimentation, and interpretation were conducted independently without reliance on AI tools.