# OpenReview forum: "Bagel-World: Towards High-Quality Visual Question-Visual Answering"
_ICLR.cc/2026/Conference — ICLR 2026 Conference Withdrawn Submission_

### Official Review · Reviewer_kxSL · 2025-10-30

**Soundness:** 3
**Presentation:** 2
**Contribution:** 2
**Rating:** 4
**Confidence:** 3

**Summary:**

This paper studies the problem of VQVA which is a variation of VQA where the answer is not textual but visual i.e. an image is generated by the system in response to the original question (as opposed to text in original VQA). To this purpose the authors construct a dataset for model finetuning (model finetuned is from another anonymous submission). Moreover, they construct an appropriate dataset for evaluation

**Strengths:**

- The proposed dataset can be a valuable resource for the community.
- The fine-tuned model seems to significantly outperform existing SOTA models on the proposed evaluation benchmark
- From the proposed system, the retriever module constitutes a novel mechanism for constructing image pairs with non-trivial / non-obvious similarities/differences that can be used for paired training

**Weaknesses:**

- It's not clear why the proposed task is different to conditional image generation (text+image) --> image
- In many cases the answer can be produced by a 2 stage system: VQA followed by text --> image generation. Actually most results of Fig. 1 can be produced with such a pipeline.
- Lack of clarity: it's impossible to understand how the dataset is generated from the text of Section 3. There are no examples illustrating the pipeline. I didn't find figure 2 very helpful. Only by looking at the prompts in supplementary some idea can be drawn.  Moreover, there's little analysis of the contents of the proposed dataset.
- Lack of sufficient novelty for ICLR: the paper is mainly about constructing a dataset and showing that if a model is finetuned on this dataset, it outperforms previous models. I think this is a rather expected result.
- Although the authors did a good job to differentiate from a concurrent submission (lightbagel), this is not always possible: for example for standard image editing task the reported performance should be attributed to lightbagel

**Questions:**

See weaknesses above.

**Details Of Ethics Concerns:**

I think the authors do not comment on GDPR related issues

---

### Official Review · Reviewer_NMv3 · 2025-10-30

**Soundness:** 2
**Presentation:** 2
**Contribution:** 2
**Rating:** 2
**Confidence:** 3

**Summary:**

This paper proposes a data-centric solution to the problem that some open-source models struggle with poor Visual Question-Visual Answering (VQ-VA) performance due to a lack of suitable training data requiring world knowledge and reasoning capabilities. Specifically, this paper builds an agent pipeline to construct a large-scale VQ-VA dataset (BAGEL-World 1.8M) containing 1.8 million examples from web-based text-and-image documents. This dataset aims to address the knowledge and reasoning deficiencies of existing data, thereby improving the VQ-VA performance of open-source models.

**Strengths:**

1. This paper provides the open-source community with a large-scale VQ-VA training dataset (BAGEL-World 1.8M) and a dedicated benchmark (IntelligentBench).

2. By applying the BAGEL-World data to the LightBAGEL model, the model's performance on IntelligentBench improves, narrowing the gap with closed-source models.

**Weaknesses:**

1. The primary contribution is a dataset generated via an agentic pipeline. However, this pipeline appears to heavily rely on the engineering integration of existing large models (some potentially closed-source, e.g., GPT-4o), rather than introducing significant algorithmic or architectural innovations in the data generation process itself. The authors regard “weights trained with LightBAGEL” as a contribution, but it is merely fine-tuning of an existing model.

2. All experiments are confined to the LightBAGEL model, without cross-architecture validation. This lack of cross-architecture validation makes it difficult to assess the dataset's general applicability to the broader open-source community. The authors should evaluate BAGEL-World on other established open-source models to demonstrate its general applicability and mitigate concerns about potential model-specific coupling effects.

3. To properly assess the unique contribution and necessity of this new dataset, the authors should demonstrate the performance differences between the BAGEL-World dataset and existing open-source datasets (individually or in combination) on the same open-source models.

4. Given the high similarity between the training and testing sets, the authors should further clarify whether the performance gains on IntelligentBench truly reflect generalizable VQ-VA capabilities, or merely overfitting to the specific data generation pipeline.

5. On standard image editing benchmarks (GEdit-Bench-EN, ImgEdit-Bench), the model trained with BAGEL-World shows only marginal improvements over the LightBAGEL baseline, and still underperforms some existing open-source methods in certain metrics. The authors attribute this to "domain differences," but the model overall still lags behind some open-source methods. It is suggested  to further validate their performance on the BAGEL-World dataset using stronger open-source models (such as Step1X-Edit) to examine whether this dataset offers a more significant performance gain and clarify the source of the performance improvement (model architecture vs. dataset quality).

**Questions:**

See Weaknesses.

---

### Official Review · Reviewer_wbom · 2025-11-02

**Soundness:** 3
**Presentation:** 2
**Contribution:** 3
**Rating:** 6
**Confidence:** 2

**Summary:**

BAGEL-WORLD introduces a dataset, BAGEL-WORLD, and a benchmark, IntelligentBench, that captures an emerging type of question-answering: visual question, visual answering (VQ-VA). This is the phenomenon where there is an image and a text question as a prompt, and the desired response is in the form of a generated image. The paper points out a lack of recent image-to-iamge datasets outside of image generation/image editing, which they believe contribute to poor VQ-VA scores. The authors collect a large amount of image-question-answer image triplets to train LightBAGEL, a base model, on VQ-VA, forming BAGEL-WORLD. The authors also curate 360 questions to serve as their benchmark, IntelligentBench, which is evaluated using GPT-4o as the judge.

**Strengths:**

1. The core concept behind the benchmark, VQ-VA, is new and interesting.
2. The paper contributes a large open-source dataset that covers a gap in readily available data: the authors point out that many current datasets focus not on QA but on image-editing.
3. The BAGEL-WORLD dataset is effective in improving VQ-VA performance (caveat is 3. in weaknesses).

**Weaknesses:**

1. There's some confusion/lack of clarity in the text regarding which cases were manually created and which cases were filtered after unanimous agreement. The paper states that 360 cases were manually created, then all 360 were also unanimously agreed upon?
2. The size of IntelligentBench, with 360 items, is somewhat small.
3. "our results demonstrate a substantial narrowing of the gap with leading proprietary systems such as Gemini"
	* While this is true for fully open-source models discussed in the paper, the gap is from 6.81 to 45.00, compared to 81.67 in proprietary models.
	* Further, it isn't shown whether this dataset would help open-weight, but closed-data models, many of which perform much better than LightBAGEL but still perform significantly worse than proprietary models.
	* Finetuning an open-weight model with BAGEL-WORLD and evaluating on IntelligentBench would go a long way to improving the work and showcasing the effectiveness of the dataset.
4. GPT-4o is both evaluated on and is the evaluator, which is somewhat circular. If my understanding that GPT-4o uses GPT-Image-1 for image generation.

**Questions:**

1. Was there any examination as to why human-human agreement was only 82.5%? Would agreement be higher if there was more significant filtering of the data?
2. Why were Gemini-2.5-Flash and GPT-4o the only VLMs tested as LLMs as a judge?

---

### Note · Authors · 2025-11-12

I have read and agree with the venue's withdrawal policy on behalf of myself and my co-authors.